# Topology-driven protein-protein interaction network analysis detects genetic sub-networks regulating reproductive capacity

Tarun Kumar[1†], Leo Blondel[2†], Cassandra G Extavour[1,2]*

[1]Department of Organismic and Evolutionary Biology, Harvard University, Cambridge, United States; [2]Department of Molecular and Cellular Biology, Harvard University, Cambridge, United States

**Abstract** Understanding the genetic regulation of organ structure is a fundamental problem in developmental biology. Here, we use egg-producing structures of insect ovaries, called ovarioles, to deduce systems-level gene regulatory relationships from quantitative functional genetic analysis. We previously showed that Hippo signalling, a conserved regulator of animal organ size, regulates ovariole number in *Drosophila melanogaster*. To comprehensively determine how Hippo signalling interacts with other pathways in this regulation, we screened all known signalling pathway genes, and identified Hpo-dependent and Hpo-independent signalling requirements. Network analysis of known protein-protein interactions among screen results identified independent gene regulatory sub-networks regulating one or both of ovariole number and egg laying. These sub-networks predict involvement of previously uncharacterised genes with higher accuracy than the original candidate screen. This shows that network analysis combining functional genetic and large-scale interaction data can predict function of novel genes regulating development.

**\*For correspondence:**
extavour@oeb.harvard.edu

†These authors contributed equally to this work

**Competing interests:** The authors declare that no competing interests exist.

## Introduction

The final shape and size of an organ is critical to organismal function and viability. Defects in human organ morphology cause a multitude of pathologies, including cancers, organ hypertrophies and atrophies (e.g. *Yang and Xu, 2011*). It is thus critical to understand the regulatory mechanisms underlying the stereotypic shape and size of organs. To this end, assessing the genetic regulation of size is significantly facilitated by using quantifiable changes in organ size and shape.

The *Drosophila melanogaster* female reproductive system is a useful paradigm to study quantitative anatomical traits. In these organs, the effects of multiple genes and the environment combine to produce a quantitative phenotype: a species-specific average number of egg-producing ovarian tubes called ovarioles. Fruit fly ovaries can contain as few as one and as many as 50 ovarioles per ovary, depending on the species (*Kambysellis and Heed, 1971*; *King, 1970*; *Markow et al., 2009*; *Sarikaya et al., 2019*), with each ovariole capable of producing eggs. Ovariole number, therefore, may affect the reproductive fitness of *Drosophila* species by determining the potential of an adult female to produce eggs (*Klepsatel et al., 2013b*; *R' kha et al., 1997*). While ovariole number within a species can vary across temperatures (*Azevedo et al., 1996*), altitudinal and latitudinal clines (*Capy et al., 1994*; *David and Bocquet, 1975*), under constant environmental conditions ovariole number is highly stereotypic (*Capy et al., 1993*; *Klepsatel et al., 2013a*; *R'Kha et al., 1991*; *R' kha et al., 1997*). The reproducibility of ovariole number thus indicates a strong genetic component (*Sarikaya et al., 2019*). Genome wide association studies and quantitative trait locus mapping have demonstrated that the ovariole number is a highly polygenic trait (*Bergland et al., 2008*;

*Lobell et al., 2017*; *Orgogozo et al., 2006*; *Wayne et al., 2001*; *Wayne et al., 1997*; *Wayne and McIntyre, 2002*). In contrast, functional genetic studies have identified only a small number of genes whose activity regulates ovariole number (discussed below). Thus, the complexity of the genetic regulation of this important trait remains largely unknown.

The determination of ovariole number in *D. melanogaster* occurs during late larval and pupal development (*King et al., 1968*). Each ovariole in the adult fly arises from a single primordial structure called a terminal filament (TF), which forms in the late third instar larval ovary (*Godt and Laski, 1995*) by convergent extension (*Keller, 2006*) of the terminal filament cells (TFCs) (*Godt and Laski, 1995*; *Sahut-Barnola et al., 1996*). TFCs are first specified from an anterior population of somatic cells in the larval ovary by the expression of transcription factors including Bric-à-brac 1/2 (*bric-à-brac 1/2; bab1/2*) and Engrailed (*engrailed; en*) (*Godt and Laski, 1995*; *Sahut-Barnola et al., 1995*). Initially a loosely arranged group in the anterior of the larval ovary, TFCs undergo morphogenetic movements to give rise to the ordered columns of cells that are TFs. Cell intercalation during convergent extension is dependent on the actin regulators Cofilin (*twinstar; tsr*) and the large Maf factor Traffic Jam (*traffic jam; tj*), and on E-cadherin dependent adhesion (*Chen et al., 2001*; *Godt and Laski, 1995*). Regulation of ovariole number is thus largely dependent on the specification of the TFCs and their rearrangement into TFs (*Sarikaya and Extavour, 2015*).

We previously showed that the regulation of both TFC and TF number is dependent on the Hippo signalling pathway (*Sarikaya and Extavour, 2015*), a pan-metazoan regulator of organ and tissue size (*Hilman and Gat, 2011*; *Sebé-Pedrós et al., 2012*). At the core of the Hippo kinase cascade are two protein kinases, Hippo (*hippo; hpo*) and Warts (*warts; wts*), which prevent the nuclear localisation of the transcriptional co-activator Yorkie (*yorkie; yki*). Yki and the transcription factor Scalloped (*scalloped; sd*) together initiate the transcription of multiple target genes, including those that promote cell proliferation and survival. In the *D. melanogaster* larval ovary, loss of Hpo in the somatic cells causes an increase in nuclear Yki, leading to an increase in TFCs, TFs, ovariole number and egg laying in adults (*Sarikaya and Extavour, 2015*).

Production of fertile eggs from a stereotypic number of ovarioles requires a spatially and temporally coordinated interplay of signalling between the somatic and germ line cells of the ovary. Thus, signalling amongst somatic and germ line cells in the larval ovary is crucial to all stages of ovarian development (*Ables and Drummond-Barbosa, 2017*; *Gilboa, 2015*; *Green et al., 2011*; *LaFever and Drummond-Barbosa, 2005*; *LaFever et al., 2010*; *Sarikaya and Extavour, 2015*). For instance, disruptions in insulin or Tor signalling affect both somatic and germ line cell proliferation (*Gancz and Gilboa, 2013*; *Green and Extavour, 2012*; *Hsu and Drummond-Barbosa, 2009*; *LaFever and Drummond-Barbosa, 2005*; *LaFever et al., 2010*; *Sarikaya et al., 2012*). Similarly, ecdysone pulses from the prothoracic gland regulate the timely differentiation of the primordial germ cells (PGCs) and the somatic TFCs (*Gancz et al., 2011*; *Hodin and Riddiford, 1998*; *Hodin and Riddiford, 2000b*). Both Hpo and ecdysone signalling also control the proportion of germ line to somatic cells by differentially regulating proliferation of both cell types (*Gancz et al., 2011*; *Sarikaya and Extavour, 2015*).

Although it is clear that genes function together in regulatory networks (*Gonzalez and Kann, 2012*), determining how the few genes functionally verified as required for ovariole development and function work together to coordinate ovariole number and ovarian function more generally is a challenge, because most genes or pathways have been considered individually. An alternative approach that is less often applied to animal developmental genetics, is a systems biology representation of complex biological systems as networks (*Barabási and Albert, 1999*; *Watts and Strogatz, 1998*). Protein-protein interaction networks (PINs) are such an example (*Albert and Barabási, 2002*). The availability of high-throughput molecular biology datasets from, for example, yeast two-hybrid, protein CHiP and microarrays has allowed for the emergence of large scale interaction networks representing both functional and physical molecular interactions (*Barabási and Oltvai, 2004*; *Berger et al., 2007*; *Giot et al., 2003*; *Gonzalez and Kann, 2012*).

With ample evidence that signalling in the ovary can affect ovarian development, but few genes functionally verified to date, we aimed to identify novel regulators of ovariole development by functionally testing all known members of all characterized *D. melanogaster* signalling pathways. We used tissue-specific RNAi to systematically knock down 463 genes in the larval ovary and looked for modifiers of the *hpo* loss of function egg laying and ovariole number phenotypes. To analyse the results of this phenotypic analysis, we used topology-driven network analysis to identify genetic

networks regulating these phenotypes, thus generating hypotheses about the relationships between these networks. With this systems biology approach, we identify not only signalling pathway genes, but also previously untested genes that affect these reproductive traits. Functional testing showed that these novel genes affect ovariole number and/or egg laying, providing us with a novel in silico method to identify target genes that affect ovarian development and function. We use these findings to propose putative developmental regulatory networks underlying one or both of ovariole formation and egg laying.

## Results

### An RNAi modifier screen for signalling pathway involvement in ovariole number

To systematically ascertain the function of signalling pathway genes and their interactions with Hippo signalling in the development of the *D. melanogaster* ovary, we first curated a list of all known and predicted signalling genes (*Gramates et al., 2017*; *Kanehisa et al., 2010*; *Mbodj et al., 2013*). We identified 475 genes belonging to the 14 developmental signalling pathways characterised in *D. melanogaster* (*Table 1*; *Supplementary file 1*), and obtained UAS: RNAi lines for 463 of these genes from the Vienna *Drosophila* RNAi centre (VDRC) or the TRiP collections at the Bloomington *Drosophila* Stock centre (BDSC) (all *D. melanogaster* genetic lines used are listed in Materials and methods).

We previously showed that reducing the levels of *hpo* in the somatic cells of the larval ovary using *traffic jam* Gal4 (*tj:Gal4*) driving *hpo[RNAi]* increased both ovariole number and egg laying of adult female flies (*Sarikaya and Extavour, 2015*). To identify genes that modify these phenotypes, we used *tj:Gal4* to drive simultaneous *hpo[RNAi]* and *RNAi* against a signalling candidate gene, and quantified the phenotypic change (*Figure 1a–d*). We observed that on driving two copies of *hpo [RNAi]* using *tj:Gal4*, we obtained a further increase in both egg laying and ovariole number (*Figure 1e*). This indicates that ovaries have further potential to increase ovariole number and egg laying beyond the increase induced by *tj:Gal4* driving one copy of *hpo[RNAi]*, and that *tj:Gal4* can

---

**Table 1.** Number of candidate genes tested in each signalling pathway.

Candidate genes are grouped by their reported roles in one or more signalling pathways based on published literature. Genes in this list are not necessarily unique to a single pathway, but rather may function in more than one signalling pathway. The list of specific genes per pathway that were included in the screen for functional analysis (*Figure 1*) is found in the *Supplementary file 1*.

| Signalling pathway | Number of genes in screen |
| --- | --- |
| EGF | 45 |
| FGF | 25 |
| FOXO | 67 |
| Hippo | 60 |
| JAK/STAT | 31 |
| JNK | 28 |
| MAPK | 29 |
| Notch | 48 |
| SHH | 54 |
| TGF B | 52 |
| Toll | 36 |
| VEGF | 17 |
| Wnt | 125 |
| mTOR | 36 |

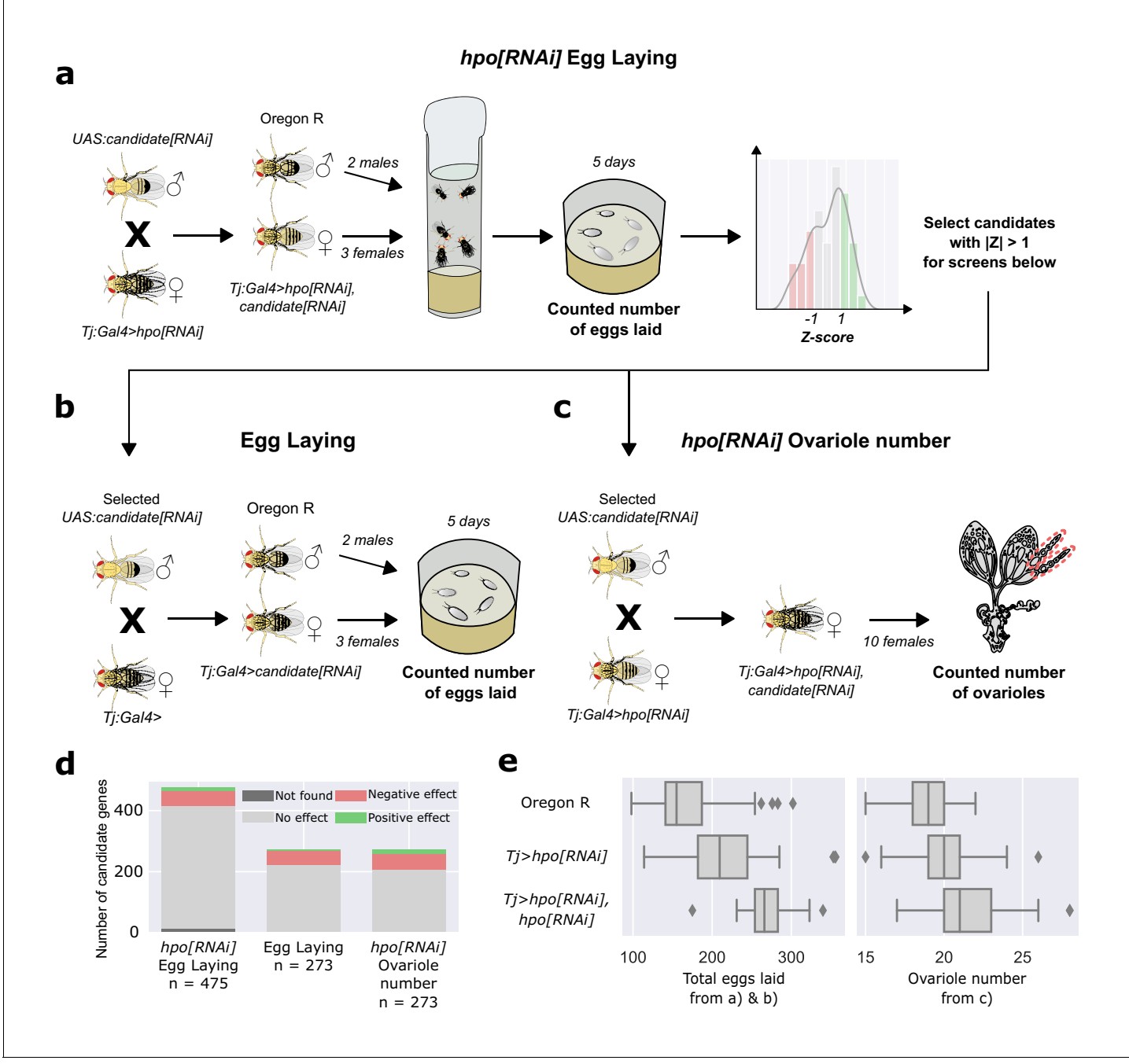

**Figure 1.** Screen methodology. (a,b,c) Diagrammatic representation of screen workflow. (d) Distributions of results of genes in the three screens. n = number of genes tested in each screen (see also **Table 2**). (e) Total eggs laid by three female flies over five days (left panel) and ovariole number (right panel) of Oregon R (top row), *tj:Gal4* driving one copy of *UAS:hpo[RNAi]* (middle row), and *tj:Gal4* driving two copies of *UAS:hpo[RNAi]* (bottom row), showing that, the previously reported *tj:Gal4 >hpo[RNAi]* ovariole number and egg laying phenotypes (**Sarikaya and Extavour, 2015**) can be modified by further UAS:RNAi-mediated gene knockdown. Distribution of egg laying and ovariole number of controls in each screen batch is illustrated in **Figure 1—figure supplement 1**.

The online version of this article includes the following figure supplement(s) for figure 1:

**Figure supplement 1.** Violin plots of egg laying and ovariole number of controls in each screen batch.

drive the expression of two RNAi constructs, indicating that our screen could identify both enhancers and suppressors of the *tj:Gal4 >hpo[RNAi]* phenotype.

We proceeded to identify modifiers of the *tj:Gal4 >hpo[RNAi]* phenotype by crossing males of each of the 463 candidate genes RNAis individually with *tj:Gal4 >hpo[RNAi]* females, and performing three phenotypic screens on the offspring. In the first screen (*Figure 1a*), we measured egg laying of three F1 female offspring (*tj:Gal4 >hpo[RNAi], signalling candidate[RNAi]*) over five days. To address batch variation (*Figure 1—figure supplement 1*), we standardised egg laying measurements by calculating the Z scores ($Z_{gene}$ = number of standard deviations from the mean) for each candidate line relative to its batch controls. 190 genes had an egg laying $|Z_{gene}|$ below 1. Previous studies have shown that the egg laying of newly eclosed adult mated females correlates with ovariole number during the first five days (*Klepsatel et al., 2013b*). We therefore eliminated these 190 genes from subsequent screening, because the change in egg laying was so modest that we considered these candidates were unlikely to show changes in ovariole number when compared to controls.

In the second screen (*Figure 1b*), we measured egg laying in a wild-type background (*tj >signalling candidate[RNAi]*) for the 273 remaining candidate genes. For the third screen (*Figure 1c*), we quantified the ovariole number of *tj:Gal4 >hpo[RNAi], signalling candidate[RNAi]* F1 adult females for the same 273 candidate genes. To choose candidates from the second and third screens for further study, we wished to account for the fact that the two screens had different effective numbers of data points. This was because egg laying data were obtained from individual vials of three females over five days, while ovariole numbers were obtained from 20 ovaries from ten females (see Materials and methods). We therefore selected the 67 genes with a $|Z_{gene}|$ above two for ovariole number (*Figure 1c and d*; *Table 2*), and the 49 genes with a more conservative $|Z_{gene}|$ above five for egg laying (*Figure 1a, b and d*; *Table 2*), for a total of 116 positive candidates for subsequent analyses.

## Ovariole number is weakly correlated with egg laying

Standardisation of the results from the three screens using Z scores allowed us to compare the effects of individual genes on one or both of egg laying and ovariole number. We performed a pairwise comparison of the $Z_{gene}$ values for all combinations of screens, and considered genes with $|Z_{gene}|$ values that were above the thresholds set for the phenotype in each screen (above two for ovariole number, above five for egg laying; green dots in *Figure 2a–c*). Across all three screens, loss of function of our positive candidates yielded reductions in ovariole number and egg laying more commonly than increases (*Figure 2a–c*). Comparing the $|Z_{gene}|$ values of egg laying and ovariole number of *tj:Gal4 >hpo[RNAi], signalling candidate[RNAi]* adult females revealed that genes that caused a change in egg laying did not always similarly affect ovariole number, and vice versa (*Figure 2a*). We therefore hypothesise that egg laying and ovariole number may be regulated by

**Table 2.** Results of the three functional genetic screens.

Number of genes tested in each screen and cumulative results. 'Negative effect' corresponds to a reduction in eggs laid or number of ovarioles below the Z score ($Z_{gene}$) threshold for each phenotype. 'Positive effect' indicates an increase above the set $Z_{gene}$ thresholds. $Z_{gene}$ thresholds for each category in each screen are indicated in brackets. The primary filter of $|Z_{gene}| < 1$ was applied only to the *hpo[RNAi]* Egg Laying screen shown in *Figure 1a*. The list of specific genes that exceeded our chosen $Z_{gene}$ thresholds for each scored phenotype (*Figure 1*) and were therefore considered to have a positive or negative effect on the phenotype, is found in the *Supplementary file 1*. The 12 genes for which RNAi stocks were unavailable at the time of testing are listed in *Table 3*.

| Egg laying screens | *Hpo[RNAi]* Egg Laying (*Figure 1a*) | Egg laying (*Figure 1b*) | Ovariole number screen | *Hpo[RNAi]* Ovariole Number (*Figure 1c*) |
|---|---|---|---|---|
| RNAi stocks unavailable | 12 | 0 | RNAi stocks unavailable | 0 |
| Primary filter ($|Z_{gene}| < 1$) | 190 | N/A | Primary filter ($|Z_{gene}| < 1$) | N/A |
| No effect ($-5 < |Z_{gene}| < 5$) | 214 | 224 | No effect ($-2 < |Z_{gene}| < 2$) | 206 |
| Negative effect ($Z_{gene} < -5$) | **48** | **44** | Negative effect ($Z_{gene} < -2$) | **54** |
| Positive effect ($Z_{gene} > 5$) | **11** | 5 | Positive effect ($Z_{gene} > 2$) | **13** |
| Total | 475 | 273 | Total | 273 |

**Table 3.** 12 signalling candidate genes with no available RNAi lines at either BDSC or VDRC at the time of this study.

| FbID | CG number | Name | Symbol |
| --- | --- | --- | --- |
| FBgn0283468 | CG3412 | supernumerary limbs | slmb |
| FBgn0267821 | CG5102 | daughterless | da |
| FBgn0266724 | CG5161 | TRAPP subunit 20 | Trs20 |
| FBgn0267378 | CG7085 | sauron | sau |
| FBgn0267487 | CG9181 | Protein tyrosine phosphatase 61F | Ptp61F |
| FBgn0267912 | CG9819 | Calcineurin A at 14F | CanA-14F |
| FBgn0086371 | CG9829 | poly | poly |
| FBgn0267350 | CG10260 | Phosphatidylinositol 4-kinase III alpha | PI4KIIIalpha |
| FBgn0267698 | CG10295 | p21-activated kinase | Pak |
| FBgn0283462 | CG18279 | Immune induced molecule prepropeptide | IMPPP |
| FBgn0267339 | CG33338 | p38c MAP kinase | p38c |
| FBgn0085506 | CG40635 | - | CG40635 |

genetically separable mechanisms. This hypothesis notwithstanding, we observed a weak but statistically significant correlation between egg laying and ovariole number (p=1e10$^{-5}$; *Figure 2d*), and this correlation was most significant in adult females that had a drastic reduction in both phenotypes (*Figure 2a*).

## No single signalling pathway dominates regulation of ovariole number or egg laying

We found that at least some genes from all tested signalling pathways could affect both egg laying and ovariole number (*Figure 3*). To determine if some pathway(s) appeared to play a more important role than others in these processes, we asked whether any of our screens were enriched for genes from a specific signalling pathway. To measure enrichment, we compared the distribution of individual pathway genes among the positive candidates in each screen, to a randomly sampled null distribution of pathway genes among a group of the same number of genes randomly selected from our curated list of 463 signalling genes (*Figure 3a*). Involvement of a pathway in the regulation of a phenotype would be reflected in a difference between the representation of pathway genes in an experimentally derived list and a randomly selected group of signalling genes. We found that rather than only one or a few pathways showing functional evidence of regulating ovariole number or egg laying, nearly all pathways affected both phenotypes (*Figure 3a*). We further tested this result by calculating the hypergeometric p-value for the enrichment of each signalling pathway, in each of the three groups of genes. Consistent with the results of the random sampling approach (*Figure 3a*), we found that most pathway members were not significantly enriched for egg laying or ovariole number phenotypes (*Figure 3b*). The absence of significant enrichment of any specific pathway is not simply attributable to the pool of genes that were screened, because our experimental manipulations of ovariole number and egg laying did cause a change in the distribution of signalling pathway members (*Figure 3—figure supplement 1*). Instead, both phenotypes appeared to be regulated by members of most or all signalling pathways (*Figure 3*). The only two exceptions to this trend were a greater than twofold enrichment of (1) genes from the Notch signalling pathway in the regulation of ovariole number (p-value<0.05, pink bar in *Figure 3a,b*), and (2) members of the Hedgehog (Hh) signalling pathway in the regulation of Hippo-dependent egg laying (p-value<0.05, brown bar in *Figure 3a and b*; *Figure 3—figure supplement 2*). In summation, our analyses of the enrichment of signalling pathways within the different screens indicated that both ovariole number and egg laying are regulated by genes from nearly all described animal signalling pathways (*Figure 3a*), rather than being dominated by any single pathway.

Comparing the results of the Egg Laying screens performed in a wild type background (*Figure 1b*) or in a *hpo[RNAi]* background (*Figure 1a*), revealed that most of the genes that met a threshold of $|Z_{gene}| > 5$ in one screen, did not meet that threshold in the other screen (*Figure 2c*).

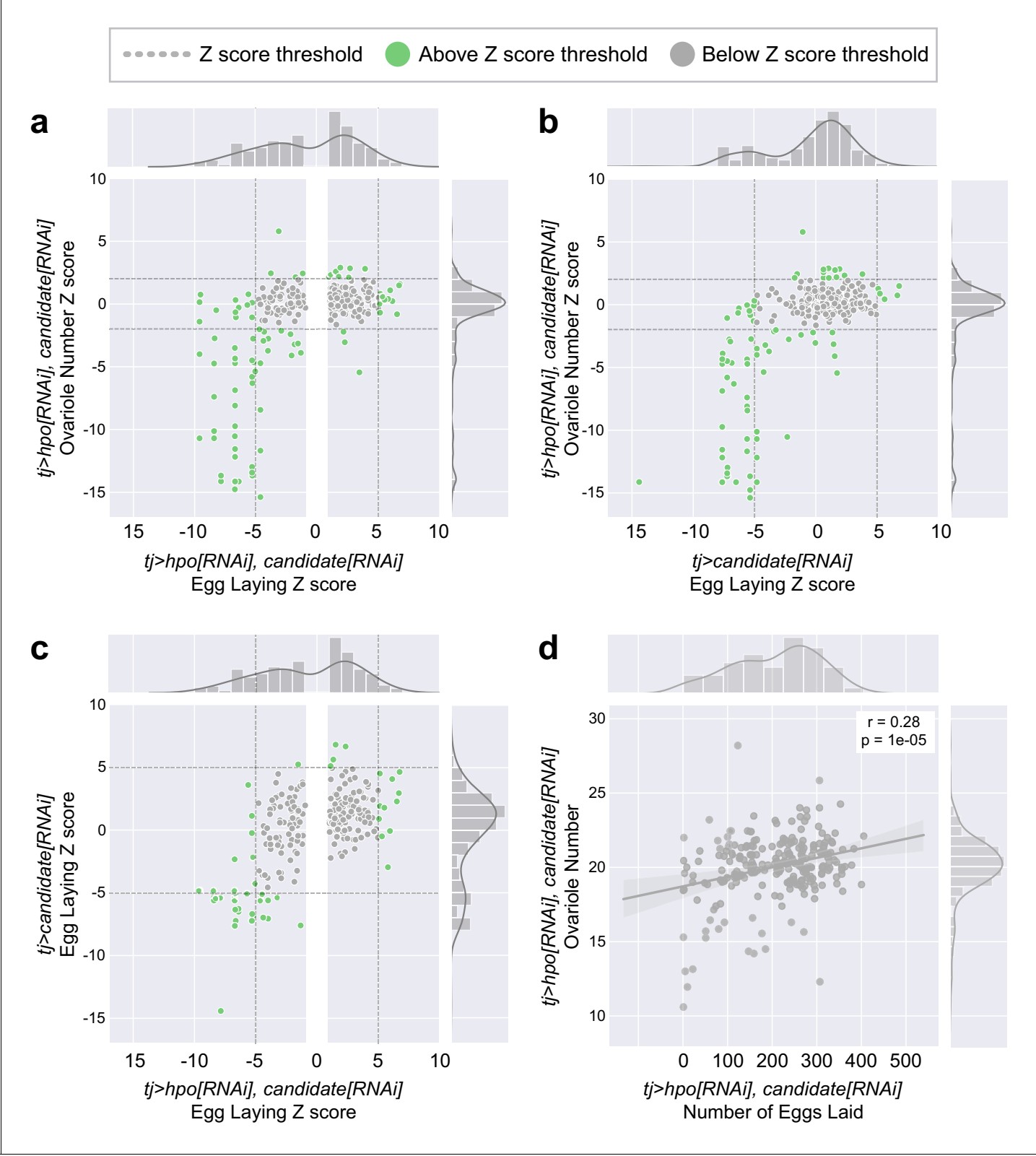

**Figure 2.** Relationship between Egg Laying and Ovariole Number phenotypes generated in the screens. (a) Scatter plots of the Z score for each gene ($Z_{gene}$) of egg laying versus the ovariole number of adult *tj >hpo[RNAi], candidate[RNAi]* females. (b) Scatter plots of the Z score for each gene ($Z_{gene}$) of egg laying of adult *tj >candidate[RNAi]* females versus the ovariole number of adult *tj >hpo[RNAi], candidate[RNAi]* females. (c) Scatter plots of the Z score for each gene ($Z_{gene}$) of egg laying of adult *tj >candidate[RNAi]* females versus egg laying of adult *tj >hpo[RNAi], candidate[RNAi]* females. In a, b

*Figure 2 continued on next page*

*Figure 2 continued*

and **c**, bar graphs on the top and right sides of each panel show the distribution of genes in each axis of the adjacent scatter plots. Green dots = genes that meet the $Z_{gene}$ threshold for the indicated phenotype. Grey dots = genes that do not meet the $Z_{gene}$ threshold for the indicated phenotype. Dark grey dotted lines = thresholds for each phenotype: $|Z_{gene}| > 5$ for Egg Laying and $|Z_{gene}| > 2$ for Ovariole Number. In **a** and **c**, the white vertical bar removes all genes in the *tj >hpo[RNAi], candidate[RNAi]* with a $|Z_{gene}| < 1$ for egg laying. These genes were not measured in the other two conditions and are therefore not represented in the scatter plots. (**d**) Correlation between non-zero Ovariole Number and Egg Laying values.

This result suggests the existence of both Hippo-dependent and Hippo-independent mechanisms of regulation of egg laying. The interpretations of separable Hippo-dependent and -independent regulation of egg laying, and of the separable regulation of ovariole number and egg laying, were supported by the results of the network analysis described in the following section.

## Centrality of genes in the ovarian protein-protein interaction networks can predict the likelihood of loss of function phenotypic effects

The finding that these reproductive traits were regulated by the genes of all signalling pathways led us to consider the broader topology of putative gene regulatory networks in the analysis of our data. Previously characterised genes in the ovary are often pleiotropic and can regulate both ovariole number and egg laying (*Gilboa, 2015*; *Sarikaya and Extavour, 2015*). As with proteins in a linear pathway, proteins in a protein-protein interaction network (PIN) are more likely to function in conjunction with genes that are connected to them within the network (e.g. *Ideker and Sharan, 2008*; *Jeong et al., 2001*). Centrality is one measure of the connectedness of a gene in the PIN and can be used to identify the most important functional centres within a protein network (*Hahn and Kern, 2005*; *Ma'ayan, 2011*). Most centrality measures use path length, which is a measure of the number of other proteins required to link any two proteins in the network. Here, we used four commonly used metrics to quantify gene centrality, each measuring slightly different properties (*Jalili et al., 2016*; *Koschützki and Schreiber, 2008*). (1) *Degree centrality* is proportional to the number of proteins that a given protein directly interacts with. (2) *Betweenness centrality* measures the number of shortest paths amongst all the shortest paths between all pairs of proteins that require passing through a particular protein. (3) *Closeness centrality* measures the average shortest path that connects a given protein to all other proteins in the network. (4) *Eigenvector centrality* is a measure of the closeness of a given protein to other highly connected proteins within the network.

We hypothesised that if the candidate genes we identified in our screen as playing roles in ovarian function worked together as a PIN, then the degree of centrality of a gene might be an indicator of function. To test this hypothesis, we first compiled a PIN consisting of all described interactions between *D. melanogaster* proteins, from the combination of publicly available protein-protein interaction (PPI) studies in the DroID database (see Materials and methods). We then calculated the four centrality measures described above for all genes within the *D. melanogaster* PIN (*Supplementary file 1*). We rank ordered only the genes tested in each screen by their score for each centrality measure, and asked whether their rank order correlated with the results of the screen, plotting these results as a receiver operating characteristic (ROC) curve. Positive correlations between centrality (a continuous variable) and phenotype (a binary variable: above or below the $|Z_{gene}|$ threshold) are reflected in an area under the curve (AUC) of more than 0.5. We found that the higher the centrality score, the greater the likelihood that a gene had $|Z_{gene}|$ values above our threshold for effects on ovariole number and egg laying (*Figure 4a*; *Figure 4—source data 1*). This supports the premise that the positive candidates identified in our screen function together as a network in the regulation of either ovariole number or egg laying. Interestingly, while the centrality of genes did predict whether a gene would affect our phenotypes of interest, it could only weakly predict the strength of that effect (p-value<0.05 in *Figure 4—figure supplement 1*).

## Genes regulating egg laying and ovariole number regulation form non-random gene interaction networks

The centrality analyses above suggested that the genes implicated in ovariole number and egg laying displayed characteristics of a functional network. PINs can often be further sorted into a collection of sub-networks. A sub-network is a smaller selection of proteins from the PIN. Examples

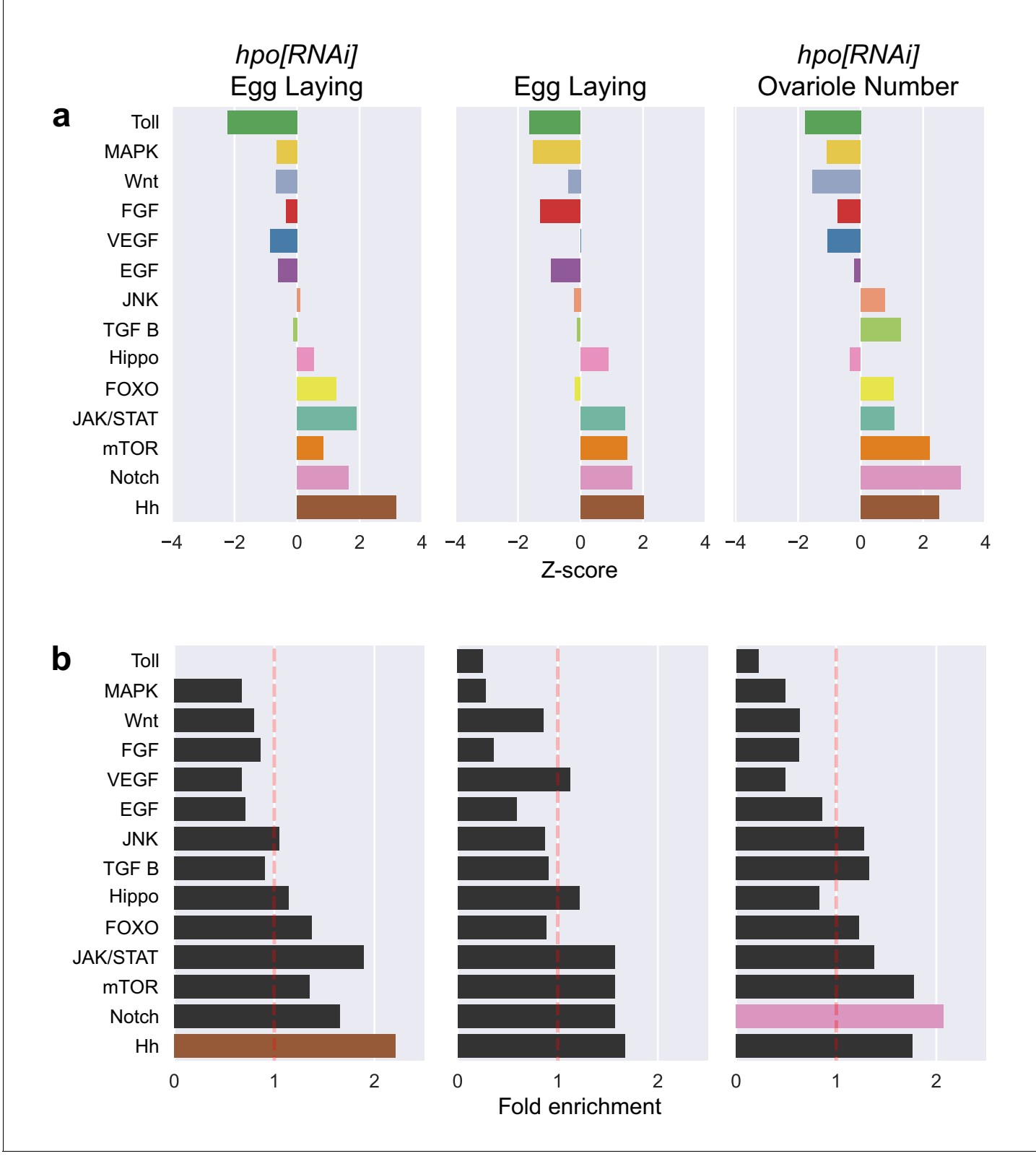

**Figure 3.** Enrichment of genes of individual signalling pathways among the experimentally obtained positive candidates of each screen. (a) Enrichment/depletion analysis to identify over- or under- represented members of individual signalling pathways among positive candidates of each screen. Positive Z scores represent an enrichment, and negative Z scores represent depletion, of genes of a pathway among those genes that experimentally affected the phenotype enrichment and depletion are defined relative to a null distribution of the expected number of members of a signalling pathway among

*Figure 3 continued on next page*

*Figure 3 continued*

a group containing the same number of randomly selected signalling genes. (**b**) Fold enrichment and hypergeometric p-value calculation to identify over- or under-representation of the genes of a pathway in each screen. Significantly enriched pathways (coloured bars: brown = Hedgehog; pink = Notch) are defined by having a hypergeometric p-value less than 0.05. Enrichment/depletion analysis of the 273 signalling pathway genes above the threshold $|Z_{gene}| > 1$ (*Figure 1a*) before screening is illustrated in *Figure 3—figure supplement 1*. *Figure 3—figure supplement 2* compares the $Z_{gene}$ of egg laying of adult females of *tj >hpo[RNAi],candidate[RNAi]* plotted against $Z_{gene}$ of egg laying of *tj >candidate[RNAi]* adult females displayed by pathway.

The online version of this article includes the following figure supplement(s) for figure 3:

**Figure supplement 1.** Enrichment/depletion analysis of the 273 signalling pathway genes above the threshold $|Z_{gene}| > 1$ (*Figure 1a*) against all signalling candidates.

**Figure supplement 2.** Comparison of egg laying candidate genes by pathway.

of such sub-networks could be proteins within the same subcellular organelle (*Foster et al., 2006*) or genes that are expressed at the same time (*Spellman et al., 1998*), thus making them likely to function together (*Srinivasan et al., 2007*). A putative module is a sub-network that can perform regulatory functions as a unit, independent of other sub-networks, and has key measurable features (*Barabási and Oltvai, 2004*; *Hartwell et al., 1999*; *Ravasz et al., 2002*; *Yook et al., 2004*). Genes and interactions between genes are not mutually exclusive to such putative modules and can be shared between putative modules. We therefore asked if our sub-networks, consisting of genes that showed similar mutant phenotypes, might display features of modularity. To determine whether genes that were implicated in regulation of ovariole number and egg laying interacted with each other in specific groups more than would be expected by chance, we created four lists of genes, called 'seed' lists, based on their individual phenotypic effects based on our screen results: (1) the core seed list, including genes positive in all three screens (*Figure 4b*); (2) the egg laying seed list, including genes positive in the wild type background egg laying screen (*Figure 1b*; *Figure 4c*); (3) the *hpo[RNAi]* egg laying seed list, including genes positive in the *hpo[RNAi]* background egg laying screen (*Figure 1a*; *Figure 4c*); and (4) the *hpo[RNAi]* ovariole seed list, including genes positive in the *hpo[RNAi]* background ovariole number screen (*Figure 1c*; *Figure 4c*). Interestingly, the core seed list, comprising genes that affected all three measured phenotypes, only consisted of genes that caused a reduction in both ovariole number and egg laying (*Figure 4b*).

We then asked whether the members of these four seed lists were more connected than would be expected by chance. In other words, we formally tested them for modularity as defined above. Meeting our criteria for modularity would suggest that the genes in these phenotypically separated seed lists might operate together as putative functional modules within the *Drosophila* PIN. We performed our modularity test using four commonly measured network metrics: (1) Largest Connected Component (LCC) (the number of proteins or nodes connected together by at least one interaction), (2) network density (the relative number of edges as compared to the theoretical maximum), (3) total number of edges, and (4) average shortest path (average of the minimum distances connecting any two proteins). We considered a sub-network to show modular features if they showed most of the following properties: higher LCC, higher network density, more edges, and shorter average shortest path length, when compared to a similarly sized, randomly sampled selection of genes from the PIN.

To determine whether these criteria would correctly identify signalling genes, which are known to function together as a module, we measured these four parameters in the original set of genes (all signalling genes) used in this study (*Table 1*). We found that the signalling genes display features of modularity when compared both to a randomly selected set of genes, as well as to a degree-controlled list of genes (*Figure 4—figure supplement 2a*). We then used this approach to test the modularity of the four phenotypic sub-networks, when compared to two different 'control sub-networks' consisting of a group of the same number of genes as contained in the sub-network, one chosen randomly from among the candidate genes from our initial screen list (*Table 1*), and the second chosen from a degree-controlled list of genes selected from the entire PIN (see Materials and methods: Building degree-controlled randomised networks). We found that the four predicted phenotypic sub-networks showed higher LCC, higher network density, and more edges (*Figure 4—figure supplement 2b*), compared to both 'control sub-networks'. This result suggests that these sub-networks display many features of modularity (although their average shortest path length is higher than

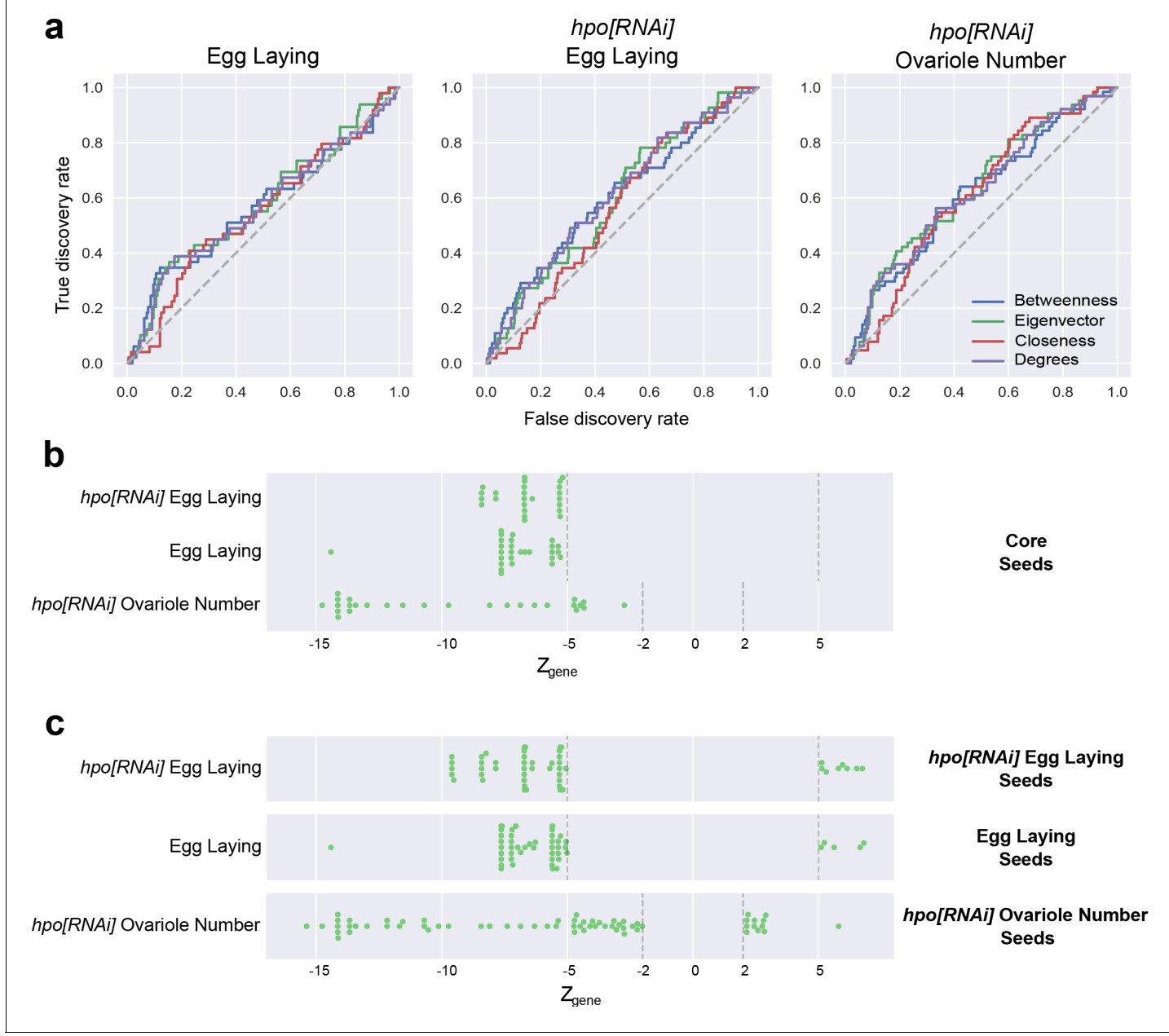

**Figure 4.** Screened genes function as a network. (**a**) Receiver operating characteristic (ROC) curves of genes ordered by rank for each of four network centrality metrics (Betweenness centrality, Eigenvector centrality, Closeness centrality and Degree centrality) versus a binary outcome (above or below Z score threshold) for each of the three screens. For each screen and metric, the Area Under the Curve (AUC) is >0.5 (***Figure 4—source data 1***). (**b**) Genes whose $|Z_{gene}|$ value was above the threshold (green dots; ***Table 2***) in all three screens were assigned to the Core seed list. (**c**) Genes whose $|Z_{gene}|$ value was above the threshold (green dots; ***Table 2***) in each screen were assigned to the corresponding seed list. ***Figure 4—source data 1*** Tabulates the AUC values for the ROC curves for each centrality measure for the three screens (a). ***Figure 4—figure supplement 1*** Compares the distribution of $Z_{gene}$ scores of the positive candidate genes for the first and fifth quintile genes sorted by their centrality metrics. ***Figure 4—figure supplement 2*** compares the network metrics of the four gene lists in b to two null distribution of genes selected from the PIN.

The online version of this article includes the following source data and figure supplement(s) for figure 4:

**Source data 1.** Area under the curve (AUC) of ROC curves.
**Figure supplement 1.** Comparisons of the $Z_{gene}$ scores of the positive candidate genes sorted by centrality metrics.
**Figure supplement 2.** Comparison of network metrics of seed lists obtained from the screen.

controls, rather than lower) and may function as putative modules within the PIN to regulate one or both of ovariole number or egg laying.

Based on published molecular interactions, in addition to the four criteria described above, further evidence for putative functional modules of genes can also be obtained by applying algorithms that use either the shortest path method (*Bromberg et al., 2008*) or the Steiner Tree approach (*Huang and Fraenkel, 2009*). Such methods identify and predict functional connections between the seed proteins, as well as additional nodes (proteins or genes) that have not been experimentally tested within the given parameters, but are known to interact with the seed genes in the PIN (*Albert and Albert, 2004*; *Yu et al., 2006*). This process can provide evidence for or against the existence of a predicted functional module, and subsequent experimental testing of this predicted module can confirm its functionality. Given its recent success in predicting gene modules, we applied the previously published Seed Connector Algorithm (SCA), a member of the Steiner Tree algorithm family (*Wang et al., 2017*; *Wang and Loscalzo, 2018*), to the groups of genes that had similar phenotypic effects in our screens (seed genes; *Figure 4b and c*). The SCA connects seed genes and previously untested novel genes (connectors) to each other using a known PIN, producing the largest possible connected putative module given the data. Using the PIN and the aforementioned four lists of seed genes, we applied a custom python implementation of the SCA (Materials and methods: 04_Seed-Connector.ipynb) to build and extract the largest possible (given our PIN) connected putative modules that regulate egg laying and ovariole number.

This SCA method yielded four putative modules, one for each seed list, which we initially referred to as the Core Module (*Figure 5b*), *hpo[RNAi]* Egg Laying module (*Figure 5—figure supplement 1*), Egg Laying Module (*Figure 5—figure supplement 2*), and *hpo[RNAi]* Ovariole Number Module (*Figure 5—figure supplement 3*) respectively. Each of these four putative modules contained seed genes, which had been functionally evaluated in our screens (green and red circles in *Figure 5*), as well as connector genes, which were genes newly predicted as regulators of these phenotypes (green and red triangles in *Figure 5*). Of the four putative modules generated by the SCA, we found that the Core module had higher centrality measures than the other three putative modules (*Figure 5—figure supplement 4*). We interpret this to mean that the genes regulating these 'Core phenotypes' are more strongly connected to each other.

We found, however, that these four groups of genes produced by the SCA did not have increased LCC values, increased network density, more edges nor decreased average shortest path (*Figure 5—figure supplement 5*), compared to our 'control sub-networks'. This result shows that the SCA in this instance does not provide evidence for putative functional modules from the four phenotypic sub-networks in our system, above and beyond the evidence provided by the application of the four network metrics discussed above. To be conservative in our description of these results, we therefore henceforth refer to these four groups of genes united by phenotype and with strong predicted interactions, as sub-networks rather than as modules. We noted that each of these four sub-networks contains genes from most, if not all, known signalling pathways, rather than only genes from a single pathway (*Figure 5—figure supplement 6*).

## Low edge densities between sub-networks suggest genetically separable mechanisms of ovariole number and egg laying

Our network analysis identified four highly connected sub-networks of genes that regulate two distinct developmental processes, together with or independently of Hippo signalling activity: ovariole number determination, which occurs primarily during larval development, and egg laying, which takes place in adult life (*Figure 5*). We wished to assess the degree to which there might be any shared genetic components between these four phenotypic sub-networks, and whether the addition of connector genes by the SCA had any impact on this. To understand potential interactions between the phenotypic sub-networks in the regulation of both ovariole number and egg laying, we constructed a composite network of all genes in each of the four phenotypic sub-networks (*Figure 5b*; *Figure 5—figure supplement 1*; *Figure 5—figure supplement 2*; *Figure 5—figure supplement 3*), which we refer to as the 'meta network' (*Figure 6a*). We then grouped the genes of the meta network into seven bins based on their phenotypic effects as measured in the three screens, resulting in sub-groups I through VII shown in *Figure 6a*. To ask whether the genes in these phenotypic groupings showed any notable interaction patterns, we compared the connectivity between genes assigned to the same phenotypic group, to the connectivity of a group of the same size

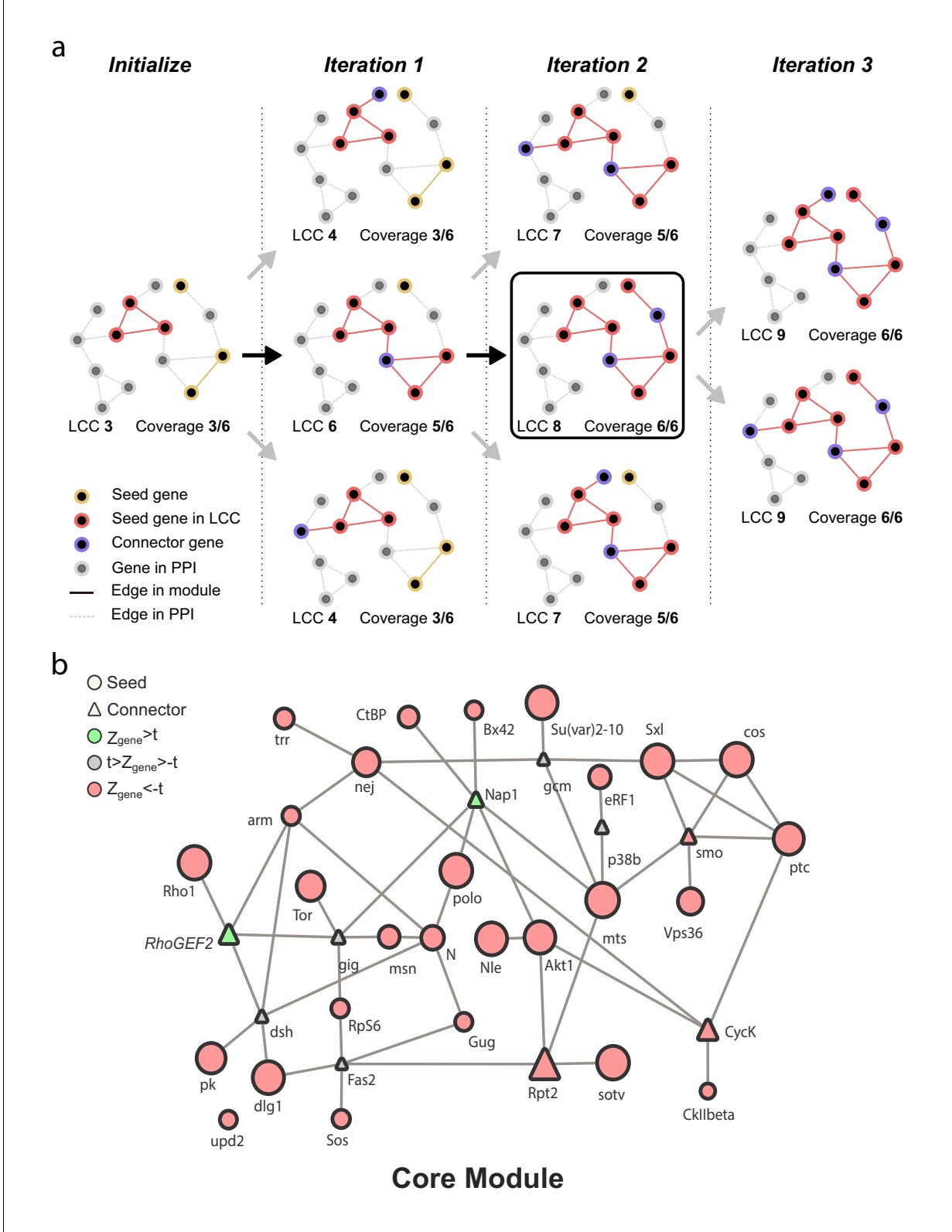

**Figure 5.** Representation of Seed Connector Algorithm (SCA) and output. (a) Schematic representation of the SCA. The algorithm initialises by creating a sub-network of seed genes from the PIN and computes the Largest Connected Component (LCC) and coverage (number of genes from the seed set in the LCC). At each iteration, genes in the direct neighbourhood of the LCC (distance = 1) are added one at a time to the seed set, and the coverage and LCC are recomputed. This process is repeated for each gene in the direct neighbourhood, each time restarting from the seed set of the preceding

*Figure 5 continued on next page*

*Figure 5 continued*

iteration. If any gene outside the seed set but in the direct neighbourhood is found to maximise coverage while minimising the LCC, it is added to the seed set as a connector gene. Black arrows indicate the path taken by the algorithm for which the criteria of maximal coverage and minimal LCC are met; such a path would be used to proceed to the subsequent iteration. Grey arrows indicate paths that fail to meet these criteria; such paths would be disregarded. The iteration repeats until the coverage cannot be increased; in this schematic example, this state is achieved in iteration 3. (**b**) The Core sub-network generated by the SCA based on the results of the genetic screens (*Figure 1a–c*). The size and colour of the shapes indicate the relative $Z_{gene}$ score of ovariole number of adult *tj >hpo[RNAi], candidate[RNAi]* females. Circles indicate seed genes (functionally tested in the screen; *Table 2*; *Supplementary file 1*) while triangles are connector genes (novel predicted genes; *Supplementary file 1*, *Figure 5—figure supplement 1*, *Figure 5— figure supplement 2*, *Figure 5—figure supplement 3*). Green = genes with a positive $Z_{gene}$ score above the threshold; red = genes with a negative $Z_{gene}$ score above the threshold; grey = genes with $Z_{gene}$ values below the threshold. *hpo[RNAi]* Egg Laying, Egg laying and *hpo[RNAi]* Ovariole Number sub-networks generated by the SCA are illustrated in *Figure 5—figure supplement 1*; *Figure 5—figure supplement 2*; *Figure 5—figure supplement 3*, respectively. *Figure 5—figure supplement 4* shows the distribution of the four centrality measures calculated for the genes in each of the four phenotypic sub-networks obtained from the SCA (*Figure 5* and *Figure 5—figure supplements 1*, *2* and *3*). *Figure 5—figure supplement 5* compares the network metrics after application of the SCA. *Figure 5—figure supplement 6* shows the enrichment/depletion of signalling pathway genes in each of the four sub-networks obtained from the SCA.

The online version of this article includes the following figure supplement(s) for figure 5:

**Figure supplement 1.** *Hpo[RNAi]* Egg Laying Sub-Network generated by the Seed Connector Algorithm (SCA).
**Figure supplement 2.** Egg Laying Sub-Network generated by the Seed Connector Algorithm (SCA).
**Figure supplement 3.** *Hpo[RNAi]* Ovariole Number Sub-Network generated by the Seed Connector Algorithm (SCA).
**Figure supplement 4.** Box plots of the four centrality measures calculated for the genes in each of the four phenotypic putative modules generated by the SCA, shown in *Figure 5* and *Figure 5—figure supplements 1*, *2* and *3*.
**Figure supplement 5.** Comparison of network metrics after application of the seed connector algorithm (SCA).
**Figure supplement 6.** Signalling pathway enrichment/depletion analysis.

randomly assembled from the genes of the meta network (*Figure 6—figure supplement 1*). As a measure of connectivity, we used an edge density map, which reflects the number of interactions between the genes within a group and between groups. We quantified the deviation between the edge density of each of groups I through VII, and their corresponding randomly assigned groups of the same size, by computing their respective Z score. When we included only seed genes in each of groups I through VII, we found that the edge density values of these groups were somewhat lower (Z score $<-1.5$) than those of the randomised groups (*Figure 6—figure supplement 1a*). The single exception to this was group IV, whose members shared more edges with each other than did the members of its randomised comparison group of the same size (*Figure 6—figure supplement 1a*). In other words, these groups of phenotypically binned seed genes were not notably more connected to each other then we would expect by chance.

In contrast, expanding each of the seven phenotypic sub-groups to include both seed genes and the connector genes predicted by the SCA changed the edge densities of these groups relative to their randomised control groups. Specifically, edge densities were much lower between groups I, II and III (Z score $< -3$), and much higher within group IV (Z score $>3$) (*Figure 6—figure supplement 1*). This shows that applying the SCA to these phenotypically binned groups increased the non-random differences in connectivity between them that were already present within the seed genes (*Figure 6—figure supplement 1a*), thus clarifying the internal structure of the meta network.

We then asked if these seven sub-groups were as connected to each other, as were the genes within each of the sub-groups, again using the edge density assessment as described above (*Figure 6b*). This analysis yielded three principal findings. First, edge densities between the three groups corresponding to the three scored phenotypes (I, II and III in *Figure 6a*) were very low (*Figure 6b*).This implies that the genes in each of the groups that regulate only one phenotype (I, II and III in *Figure 6a*) share more interactions with themselves than with genes in the other two groups, suggesting that each of these initially scored phenotype can be largely regulated by an independent, non-interacting set of genes. Second, the core group (IV in *Figure 6a*) displayed a higher edge density with the other three groups (I, II and III in *Figure 6a*) than any of those three groups did with each other (*Figure 6b*). Consistent with the definition of core genes as regulating all three scored reproductive phenotypes, this result suggests that the core genes, in contrast to those from the other three groups, may share substantial functional interactions with genes of the other groups. Finally, three small additional groups emerged from this analysis (V, VI and VII in *Figure 6a*),

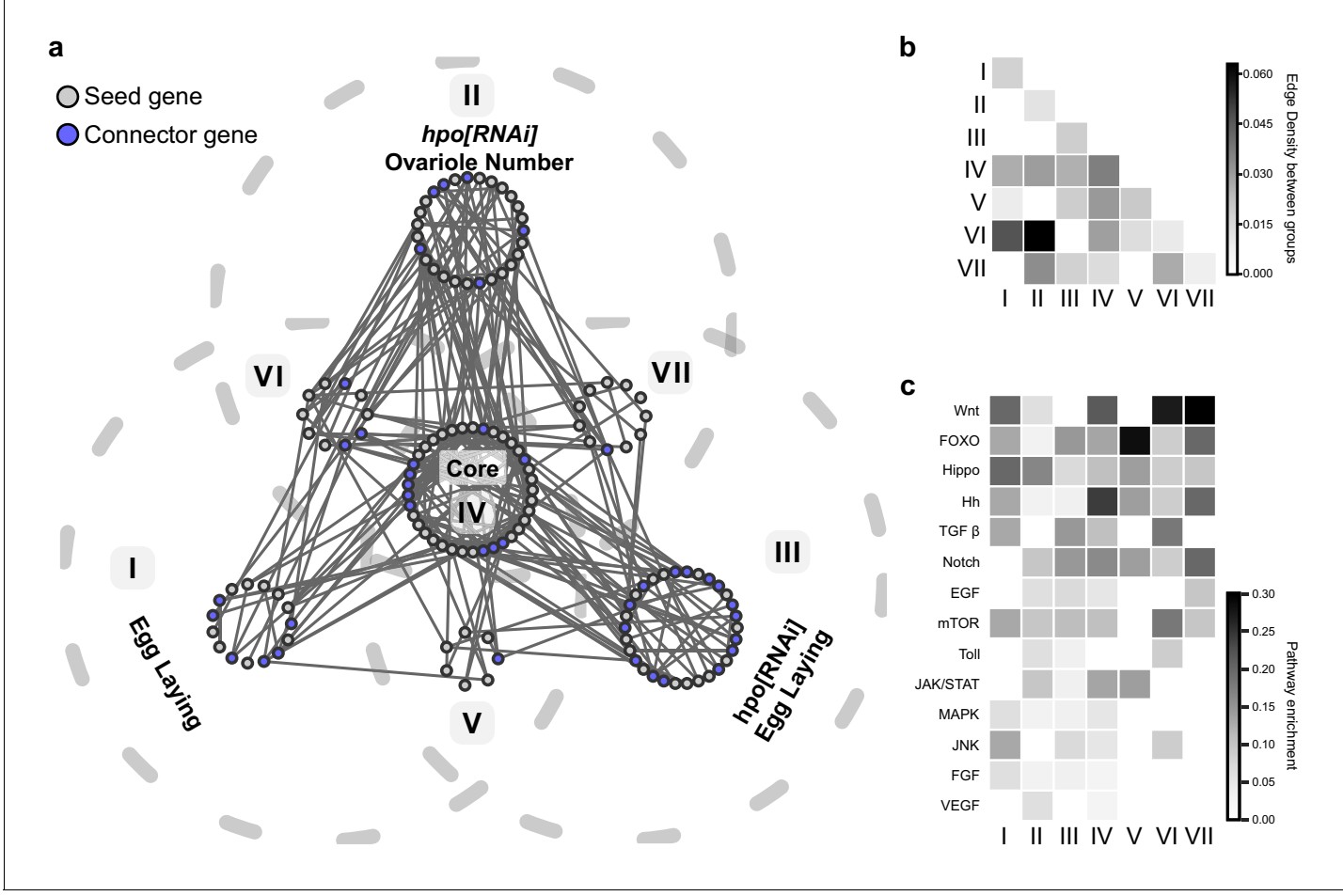

**Figure 6.** Phenotypically separable sub-networks formed by analysis of the combined genes from all sub-networks. The meta network is generated by the union of the genes in the four phenotypic sub-networks: *hpo[RNAi]* Egg Laying (*Figure 5—figure supplement 1*), Egg Laying (*Figure 5—figure supplement 2*), *hpo[RNAi]* Ovariole Number (*Figure 5—figure supplement 3*) and Core (*Figure 5b*). (a) The meta network is represented as a Venn diagram, in which each grey dotted outline represents the screen in which a given gene was identified as affecting the scored phenotype. Within each sub-network, grey circles indicate seed genes, and blue circles indicate connector genes. Solid grey lines indicate interactions between genes in the meta network from the PIN. (b) Edge densities between the seven sub-networks of the meta network. (c) Relative enrichment of screened members of the 14 tested developmental signalling pathways within the seven sub-networks of the meta network. *Figure 6—figure supplement 1* compares the edge density of the seven sub-networks of the meta network to the edge densities of a random assignment of the positive candidates in the screen to a seven similarly sized sub-networks.

The online version of this article includes the following figure supplement(s) for figure 6:

**Figure supplement 1.** Comparison of edge densities between the seven sub-networks of the meta network, and to a randomly assigned grouping of genes in the meta network.

suggesting small networks of genes that might work together to regulate two of the three scored phenotypes. In sum, this meta network analysis supports the hypothesis of three potentially largely non-interacting genetic networks that regulate Hippo-dependent ovariole number, Hippo-dependent egg laying, and Hippo-independent egg laying, respectively. The presence of smaller sub-networks (V, VI and VII in *Figure 6a*) that interact with each other further supports the observation that the putative modules predicted by the SCA – which we refer to as sub-networks – could include genes that function within more than one such sub-network (*Figure 5—figure supplement 4*). Moreover, each of these genetically separable sub-networks included genes in multiple signalling pathways (*Figure 6c*).

## Network analysis predicts novel genes involved in egg laying and ovariole number

The four predicted phenotypic sub-networks produced by the SCA approach included connector genes that were not included in our original screen, and thus had not been tested for possible effects on our phenotypes of interest (triangles in *Figure 5b*; *Figure 5—figure supplements 1–3*). Given that prior work in human disease models showed that predicted disease modules can correctly identify gene involvement in the relevant diseases (*Chen et al., 2006*; *Gonzalez et al., 2007*; *Wang et al., 2017*; *Wang and Loscalzo, 2018*), we asked whether our deployment of the SCA had likewise successfully predicted novel, functionally important genes in each sub-network. To do this, we used *UAS:RNAi* lines for each connector, driven by *tj:Gal4* to measure the effects of knocking down each of the connector genes (triangles in *Figure 5b* and *Figure 5—figure supplement 1*; *Figure 5—figure supplement 2*; *Figure 5—figure supplement 3*) both on phenotypes within the sub-network where they were predicted (*Figure 7a–b*, *Figure 7—source data 1*), and on either of the other two tested phenotypes (*Figure 7c*, *Figure 7—source data 2*).

Of the ten predicted novel connectors within the Core sub-network, loss of function of several of these had significant effects on at least one of the three scored phenotypes. Five affected ovariole number, two affected Hpo-dependent egg laying and one affected Hpo-independent egg laying. However, only one of them significantly altered all three scored phenotypes (*Figure 7a*; *Figure 7—source data 1*).

The predicted connector genes from two of the other three phenotypic sub-networks showed high positive prediction rates for novel genes within the sub-networks. RNAi against seven out of 18 of the *hpo[RNAi]* Egg Laying connectors, three out of 11 of the *hpo[RNAi]* Ovariole Number connectors, and none of the 11 Egg Laying connectors, significantly affected the sub-network phenotype (*Figure 7—source data 1*). Thus, although the Egg Laying connectors failed to impact this phenotype in our assay, 41.1% and 27.2% of the connectors from the other two sub-networks were correctly predicted (*Figure 7b*; *Figure 7—source data 1*).

In sum, taken across all sub-networks, this methodology correctly identified genes regulating at least one of the scored reproductive phenotypes, at significantly higher rates than those obtained in the original screen of 463 members of all known signalling pathways (*Figure 7c*; *Figure 7—source data 2*). By this measure, testing network-predicted novel genes derived from experimentally obtained data was even more successful than testing signalling pathways as a means of identifying novel genes that regulate ovariole number and egg laying.

## Discussion

In this study, we have identified many novel genes that regulate one or both of egg laying and ovariole number. Though the development of the insect ovary has been studied for over 100 years, our understanding of the genetic mechanisms that regulate the development of the ovary is sparse. The female reproductive system and its ability to produce eggs are one of the key determinants for the survival of a species in an ecological niche. The genes we have uncovered here are possible targets for the regulation of the construction and function of the reproductive system in *D. melanogaster*, and potentially in other species of insects as well. Understanding the gene regulatory networks that regulate egg laying and ovariole development could provide a framework to understand the key regulatory steps during this process that may be modified over evolutionary time, to yield the wide diversity of ovariole numbers and fecundities displayed by extant insects. We suggest that, given our success in applying a network approach to the results of a traditional forward genetic screen, the field of developmental genetics should find it fruitful to apply network analyses to the interpretation of large scale transcriptomic and proteomic data.

### Identification of regulatory sub-networks for ovariole development and egg laying

The *D. melanogaster* ovary is a commonly studied model for organogenesis (*Chen et al., 2001*; *Godt and Laski, 1995*; *Lobell et al., 2017*; *Sarikaya and Extavour, 2015*), stem cell maintenance (*Gilboa, 2015*) and interactions of development and ecology (*Cohet and David, 1978*; *Hodin and Riddiford, 2000a*; *Klepsatel et al., 2013a*; *Sarikaya et al., 2019*). Nevertheless, our understanding

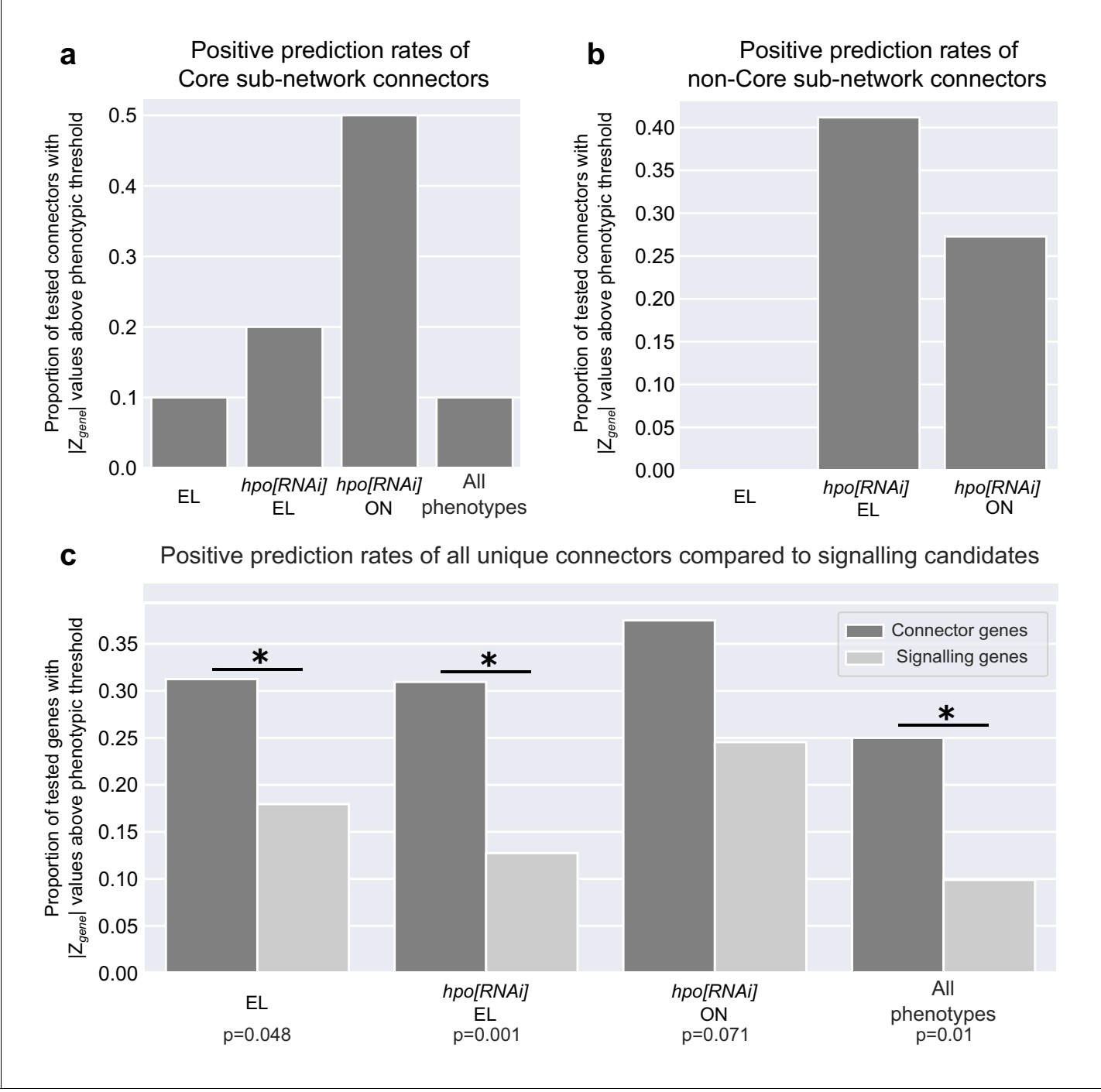

**Figure 7.** Positive prediction rates of the connector genes in each of the four sub-networks. (a) Proportion of Core sub-network connector genes with $|Z_{gene}|$ above the threshold in each of the three screens. The 'All phenotypes' category includes the genes with $|Z_{gene}|$ above the threshold in all three screens. (b) Proportion of tested connector genes in each of the three sub-networks with $|Z_{gene}|$ above the threshold within their corresponding screen. (c) Proportion of all unique connector genes (dark grey bars) predicted by all four sub-networks compared to the proportion of signalling candidate genes (light grey bars) with $|Z_{gene}|$ above the respective threshold in any of the three phenotypic screens (*Figure 1a, b and c*). Positive connector and signalling candidates that were above the $|Z_{gene}|$ threshold in all three phenotypic screens (*Figure 1a, b and c*) are indicated in an 'All phenotypes' column. Statistical significance was computed using the binomial test, comparing the probability of a positive candidate amongst the connectors to the probability of a positive candidate amongst the signalling candidates (p-value is found below each bar). *Figure 7—source data 1* tabulates the distribution of seed genes and connectors in each sub-network used in *Figure 7a and b*. *Figure 7—source data 2* tabulates the unique connector and signalling genes above $|Z_{gene}|$ threshold for the three phenotypic measurements plotted in *Figure 7c*. The connector genes are listed in the

*Figure 7 continued on next page*

*Figure 7 continued*

*Supplementary file 1* and can be identified under the column header [SubNetworkName]_Connector. The raw data for egg laying and ovariole number for each of the connector genes can be found within the *Supplementary file 1*.

The online version of this article includes the following source data for figure 7:

**Source data 1.** Distribution of seed genes and connectors in each sub-network.
**Source data 2.** Number of unique connector and signalling genes above |Z_gene| threshold for the three phenotypes measured in this screen (*Figure 1a, b and c*).

of the genetic mechanisms that regulate these processes remains fragmentary. In this paper, we have identified four distinct protein interaction sub-networks that regulate ovariole number and egg laying in the *D. melanogaster* ovary. These sub-networks consist of both novel and previously characterised genes that regulate either ovariole number or egg laying or both, thus enhancing our understanding of the genetic underpinnings of this reproductive system.

Of the four sub-networks, the Core sub-network affects both ovariole number and egg laying. The Core sub-network contains numerous housekeeping genes, including regulators of transcription, translation and cell division such as *polo* (*Llamazares et al., 1991*), *cyclin K* (*Edwards et al., 1998*), *nucleosome assembly protein 1* (*Ito et al., 1996*) and *eukaryotic translation release factor 1* (*Chao et al., 2003*). While *polo* and *eukaryotic translation release factor one* are members of signalling pathways, *cyclin K* and n*ucleosome assembly protein one* are genes predicted by the SCA. Given that the Core sub-network largely consists of genes whose loss of function decreases both of these parameters, we hypothesise that these are essential genes for the basic structure and function of the ovaries. Essential genes are more interconnected in a PIN with higher centrality measures (*Jeong et al., 2001*; but see *Yu et al., 2008*) and interestingly, we find that the genes in the Core sub-network also have higher connectivity than those in the other three sub-networks (*Figure 5—figure supplement 4*).

In addition to genes that regulate basic cellular processes, the Core sub-network is enriched for the central components of the Hh signalling cascade, namely *patched* (*ptc*), *smoothened* (*smo*) and *costa* (*cos*) (*Lee et al., 2016*). However, we find that the loss of Hh ligand, which is expressed in the TF cells in the developing larval ovary (*Lai et al., 2017*), does not significantly affect either ovariole number or egg laying. Though surprising, ligand-independent activation of Hedgehog signalling has been observed before. For example, in the *Drosophila* eye, loss of either *ptc* or *cos* in clones leads to non-cell autonomous proliferation in wild type cells, as well as growth disadvantages in the mutant tissue (*Christiansen et al., 2012*). In another example, sufficient intracellular *smo* levels can also activate downstream transcription of Hh pathway targets, showing that Hh itself is not always required to activate the cascade (*Jiang et al., 2018*).

## Development of the larval ovary

The *hpo[RNAi]* Ovariole Number sub-network is composed of genes that affect the Hippo signalling activity-dependent determination of ovariole number during development. Establishment of ovariole number occurs largely during the third instar stage of larval development in *D. melanogaster* (*Godt and Laski, 1995*; *Hodin and Riddiford, 1998*; *King et al., 1968*; *Sahut-Barnola et al., 1996*). During this period, the TFCs are specified in the anterior of the ovary and undergo rearrangement into stacks of cells called TFs, each of which gives rise to an ovariole (*Godt and Laski, 1995*; *Sahut-Barnola et al., 1995*). TF specification requires the expression of *engrailed* (En) (*Bolívar et al., 2006*) and the transcription factors Bab1 and Bab2, encoded by the *bric-à-brac* locus (*Couderc et al., 2002*; *Godt et al., 1993*). A third transcription factor, Lmx1a, was recently found to be necessary for the specification of the TFCs (*Allbee et al., 2018*). Our *hpo[RNAi]* Ovariole Number sub-network identifies numerous additional novel transcription factors including *bunched* (*bun*) and *retinoblastoma-family protein* (*rbf*), which we hypothesise could also be involved in the specification of ovariole number. *bun* and *rbf* have been implicated in the migration (*Dobens et al., 2005*) and endoreplication (*Cayirlioglu et al., 2003*) of the follicle cells during oogenesis, but have not, to our knowledge, been previously identified as playing a role in the context of larval ovary development.

The TFCs specified in the larval ovary undergo a process of convergent extension to form TFs. This process of convergent extension requires cell intercalation, and the actin depolymerising factor Cofilin, encoded by the gene *twinstar*, is essential to this process (*Chen et al., 2001*). During intercalation, the cells also dynamically modify their actin cytoskeleton and their expression of E-cadherin (*Godt and Laski, 1995*). Our *hpo[RNAi]* Ovariole Number sub-network further identifies *Rho1* (*Barrett et al., 1997*) and *Rho kinase* (*Rok*) (*Mizuno et al., 1999*) as necessary for correct ovariole number. During the extension of the *D. melanogaster* embryonic germ band, a commonly studied model of convergent extension, the localised activation of the actin-myosin network facilitated by *Rho1* and *Rok* is necessary for cell intercalation (*Kasza et al., 2014*). Given the known roles of *Rho1* and *Rok* as regulators of the actin cytoskeleton (*Ridley, 2006*), we propose that TF assembly in the ovary requires both these proteins for correct cell intercalation. A third actin cytoskeleton regulator, *misshapen* (*msn*), was also identified by our *hpo[RNAi]* Ovariole Number sub-network. *msn* encodes a MAP kinase previously shown to regulate the polarisation of the actin cytoskeleton during oogenesis (*Lewellyn et al., 2013*), but has not, to our knowledge, been studied to date in the context of larval ovarian development.

We propose that the polarity of the somatic cells in the ovary is also necessary for correct larval ovary development, given the presence of the lateral membrane proteins *discs large 1* (*dlg1*) and *prickle* (*pk*) in the ovariole sub-network. During the maturation of the TFs during larval development, the TFCs undergo significant cell shape changes, coincident with localised expression of beta-catenin and actin to the lateral edges of the TFCs (*Godt and Laski, 1995*). Restriction of the E-cadherin domain in epithelia requires establishment of the basolateral domain (*Harris and Peifer, 2004*) and we propose that testing a similar requirement for *dlg1* and *pk* in the larval ovary would be a fruitful avenue for future studies.

## Network analysis as a tool in developmental biology

Using a systems biology approach to analyse RNAi screening data has proven fruitful, providing us with new insights into the development and function of the *D. melanogaster* ovary by identifying novel and previously understudied genes that regulate this process. Systematic analysis of the function of single genes in development has been a historical convention and has provided valuable and precise genetic interaction information (*Jansen et al., 2002*; *von Mering et al., 2002*). With the advent of genome-wide analysis, however, we can use data from a larger number of genes to predict the identity of additional functionally significant genes with relative ease (*Yu et al., 2006*). We note that the novel gene prediction rate within each individual sub-network ranged from as high as 41.1% from the *hpo[RNAi]* Ovariole Number sub-network to as low as 0% from the Egg Laying sub-network (*Figure 7b*; *Figure 7—source data 1*). We suggest that this may be due to multiple factors. Firstly, the possible incompleteness of the PIN is expected to lead to some areas of the network being sparse or non-existent (*von Mering et al., 2002*). If the sub-network of interest happens to fall in such regions of the PIN, prediction algorithms will fail. Secondly, the initial restriction of tested genes to signalling pathway members might have provided a seed list too sparse to usefully predict functional connectors. Finally, it could be the case that 'Egg Laying' is such a complex phenotype that its gene regulation cannot be adequately captured within a highly connected network of the type suited for identification by the analyses we have used here.

Ovariole number in *D. melanogaster* is the outcome of a discrete developmental process with a clear beginning and end, comprising a specific series of cellular behaviours that take place in the confines of one organ (*Godt and Laski, 1995*; *Hodin and Riddiford, 2000a*; *Sahut-Barnola et al., 1996*). Once established during larval life, ovariole number in *Drosophila* remains unaltered through to and during adulthood, even if oogenesis within those ovarioles suffers congenital or age-related defects (*King, 1970*). Because previous work suggested that ovariole number in *Drosophila* could have at least some predictive relation to egg laying (*Cohet and David, 1978*; *Klepsatel et al., 2013b*; *Sarikaya and Extavour, 2015*), we reasoned that scoring the latter phenotype in a primary screen (*Figure 1a*) could be an effective way to uncover ovariole number regulators (*Figure 1c*). While our results showed that this was true in many cases, it was also clear that these two traits can vary independently (*Figure 2*), highlighting the fact that ovariole number is not the only determinant of egg laying. Egg-laying dynamics, even during the limited five-day assay used in our study, are likely influenced not just by a single anatomical parameter such as ovariole number, but rather by many biological, biomechanical, hormonal and behavioural processes. Consequently, the sub-

network we were able to extract from the results of this screen (*Figure 1a and b*) might be too coarse to extract novel genes that participate in the potentially complex gene interactions regulating egg laying. Furthermore, genes predicted within each of the sub-networks are unlikely to function exclusively within just one sub-network. This conclusion is supported by our observation that genes predicted to function in any of the sub-networks, also function in at least one of the four sub-networks at a higher rate than genes selected for screening by their presence in a signalling pathway (*Figure 7c*). We also observe that although substantial regions of the meta network do not share interactions with genes in the other sub-networks (*Figure 6b*), we do find smaller sub-networks where there is some overlap, further indicating pleiotropy of some genes in both egg laying and ovariole number regulation.

The predictive rates of the approach we have used here, although encouraging, are likely limited by the degree of noise in the high-throughput data used to generate the PIN (*Li et al., 2010*), the sparseness of the PIN, and the degree of misidentification of protein interactions (*Zhang et al., 2017*). Addressing one or more of these parameters could improve the outcomes of future network predictions from developmental genetics data. For example, the problem of sparseness, which is a paucity of high confidence detectable interactions relative to all biologically relevant interactions, has been addressed in other studies by using an 'Interolog PIN' (*Matthews et al., 2001*) in place of an organism-specific PIN. An Interolog PIN combines known interactions from multiple organisms and has been used successfully to identify, for example, gene modules relevant in squamous carcinoma, based on a starting dataset of microarray data on differentially expressed genes between cancer cells and the surrounding tissue (*Wachi et al., 2005*). Future studies applying such an Interolog PIN to the outcomes of genetic screens for developmental processes of interest could potentially overcome the problem of sparseness, as well as the biases towards proteins that are more heavily studied and thus better represented in organism-specific PINs.

# Materials and methods

**Key resources table**

| Reagent type (species) or resource | Designation | Source or reference | Identifiers | Additional information |
|---|---|---|---|---|
| Chemical compound, drug | Hoechst 33321 | ThermoFisher | H1399 | 1 ug/ml final concentration |
| Software, algorithm | matplotlib | https://matplotlib.org/ | 3.0.0 | |
| Software, algorithm | networkx | http://networkx.github.io/ | 2.3 | |
| Software, algorithm | numpy | https://www.numpy.org/ | 1.11.3 | |
| Software, algorithm | pandas | https://pandas.pydata.org / | 0.20.3 | |
| Software, algorithm | progressbar | https://github.com/niltonvolpato/python-progressbar | 3.38.0 | |
| Software, algorithm | scipy | https://www.scipy.org/ | 1.1.0 | |
| Software, algorithm | seaborn | https://seaborn.pydata.org/ | 0.9.0 | |
| Software, algorithm | Cytoscape | https://cytoscape.org/ | 3.4.0 | |
| Software, algorithm | Inkscape | https://inkscape.org/ | 0.92.3 | |
| Software, algorithm | Python3 | https://www.python.org/ | 3.7 | |

*Continued on next page*

*Continued*

| Reagent type (species) or resource | Designation | Source or reference | Identifiers | Additional information |
|---|---|---|---|---|
| Genetic reagent (*D. melanogaster*) | *D. melanogaster.* Expresses dsRNA for RNAi of CanA1 (FBgn0010015) under UAS control in the VALIUM10 vector.: y[1] v[1]; P{y[+t7.7] v[+t1.8]=TRiP.JF01871}attP2 | Bloomington *Drosophila* Stock Center | BDSC:25850; FlyBase: FBst0025850; | |
| Genetic reagent (*D. melanogaster*) | *D. melanogaster.* Expresses dsRNA for RNAi of fng (FBgn0011591) under UAS control in the VALIUM10 vector.: y[1] v[1]; P{y[+t7.7] v[+t1.8]=TRiP.JF01967}attP2 | Bloomington *Drosophila* Stock Center | BDSC:25947; FlyBase: FBst0025947; | |
| Genetic reagent (*D. melanogaster*) | *D. melanogaster.* Expresses dsRNA for RNAi of Aplip1 (FBgn0040281) under UAS control in the VALIUM10 vector.: y[1] v[1]; P{y[+t7.7] v[+t1.8]=TRiP.JF02049}attP2 | Bloomington *Drosophila* Stock Center | BDSC:26024; FlyBase: FBst0026024; | |
| Genetic reagent (*D. melanogaster*) | *D. melanogaster.* Expresses dsRNA for RNAi of E(spl)mdelta-HLH (FBgn0002734) under UAS control in the VALIUM10 vector.: y[1] v[1]; P{y[+t7.7] v[+t1.8]=TRiP.JF02101}attP2/TM3, Sb[1] | Bloomington *Drosophila* Stock Center | BDSC:26203; FlyBase: FBst0026203; | |
| Genetic reagent (*D. melanogaster*) | *D. melanogaster.* Expresses dsRNA for RNAi of sima (FBgn0266411) under UAS control in the VALIUM10 vector.: y[1] v[1]; P{y[+t7.7] v[+t1.8]=TRiP.JF02105}attP2 | Bloomington *Drosophila* Stock Center | BDSC:26207; FlyBase: FBst0026207; | |
| Genetic reagent (*D. melanogaster*) | *D. melanogaster.* Expresses dsRNA for RNAi of E(spl)m8-HLH (FBgn0000591) under UAS control in the VALIUM10 vector.: y[1] v[1]; P{y[+t7.7] v[+t1.8]=TRiP.JF02096}attP2 | Bloomington *Drosophila* Stock Center | BDSC:26322; FlyBase: FBst0026322; | |
| Genetic reagent (*D. melanogaster*) | *D. melanogaster.* Expresses dsRNA for RNAi of pan (FBgn0085432) under UAS control in the VALIUM10 vector.: y[1] v[1]; P{y[+t7.7] v[+t1.8]=TRiP.JF02306}attP2 | Bloomington *Drosophila* Stock Center | BDSC:26743; FlyBase: FBst0026743; | |
| Genetic reagent (*D. melanogaster*) | *D. melanogaster.* Expresses dsRNA for RNAi of CanB (FBgn0010014) under UAS control in the VALIUM10 vector.: y[1] v[1]; P{y[+t7.7] v[+t1.8]=TRiP.JF02616}attP2 | Bloomington *Drosophila* Stock Center | BDSC:27307; FlyBase: FBst0027307; | |
| Genetic reagent (*D. melanogaster*) | *D. melanogaster.* Expresses dsRNA for RNAi of mib1 (FBgn0263601) under UAS control in the VALIUM10 vector.: y[1] v[1]; P{y[+t7.7] v[+t1.8]=TRiP.JF02629}attP2 | Bloomington *Drosophila* Stock Center | BDSC:27320; FlyBase: FBst0027320; | |
| Genetic reagent (*D. melanogaster*) | *D. melanogaster.* Expresses dsRNA for RNAi of Nct (FBgn0039234) under UAS control in the VALIUM10 vector.: y[1] v[1]; P{y[+t7.7] v[+t1.8]=TRiP.JF02648}attP2 | Bloomington *Drosophila* Stock Center | BDSC:27498; FlyBase: FBst0027498; | |

*Continued*

| Reagent type (species) or resource | Designation | Source or reference | Identifiers | Additional information |
|---|---|---|---|---|
| Genetic reagent (*D. melanogaster*) | *D. melanogaster.* Expresses dsRNA for RNAi of Cbl (FBgn0020224) under UAS control in the VALIUM10 vector.: y[1] v[1]; P{y[+t7.7] v[+t1.8]=TRiP.JF02650}attP2 | Bloomington *Drosophila* Stock Center | BDSC:27500; FlyBase: FBst0027500; | |
| Genetic reagent (*D. melanogaster*) | *D. melanogaster.* Expresses dsRNA for RNAi of Atg12 (FBgn0036255) under UAS control in the VALIUM10 vector.: y[1] v[1]; P{y[+t7.7] v[+t1.8]=TRiP.JF02704}attP2 | Bloomington *Drosophila* Stock Center | BDSC:27552; FlyBase: FBst0027552; | |
| Genetic reagent (*D. melanogaster*) | *D. melanogaster.* Expresses dsRNA for RNAi of fz2 (FBgn0016797) under UAS control in the VALIUM10 vector.: y[1] v[1]; P{y[+t7.7] v[+t1.8]=TRiP.JF02722}attP2 | Bloomington *Drosophila* Stock Center | BDSC:27568; FlyBase: FBst0027568; | |
| Genetic reagent (*D. melanogaster*) | *D. melanogaster.* Expresses dsRNA for RNAi of Psn (FBgn0284421) under UAS control in the VALIUM10 vector.: y[1] v[1]; P{y[+t7.7] v[+t1.8]=TRiP.JF02760}attP2/TM3, Sb[1] | Bloomington *Drosophila* Stock Center | BDSC:27681; FlyBase: FBst0027681; | |
| Genetic reagent (*D. melanogaster*) | *D. melanogaster.* Expresses dsRNA for RNAi of Pi3K92E (FBgn0015279) under UAS control in the VALIUM10 vector.: y[1] v[1]; P{y[+t7.7] v[+t1.8]=TRiP.JF02770}attP2/TM3, Sb[1] | Bloomington *Drosophila* Stock Center | BDSC:27690; FlyBase: FBst0027690; | |
| Genetic reagent (*D. melanogaster*) | *D. melanogaster.* Expresses dsRNA for RNAi of Cdk4 (FBgn0016131) under UAS control in the VALIUM10 vector.: y[1] v[1]; P{y[+t7.7] v[+t1.8]=TRiP.JF02795}attP2 | Bloomington *Drosophila* Stock Center | BDSC:27714; FlyBase: FBst0027714; | |
| Genetic reagent (*D. melanogaster*) | *D. melanogaster.* Expresses dsRNA for RNAi of mts (FBgn0004177) under UAS control in the VALIUM10 vector.: y[1] v[1]; P{y[+t7.7] v[+t1.8]=TRiP.JF02805}attP2 | Bloomington *Drosophila* Stock Center | BDSC:27723; FlyBase: FBst0027723; | |
| Genetic reagent (*D. melanogaster*) | *D. melanogaster.* Expresses dsRNA for RNAi of Pdk1 (FBgn0020386) under UAS control in the VALIUM10 vector.: y[1] v[1]; P{y[+t7.7] v[+t1.8]=TRiP.JF02807}attP2 | Bloomington *Drosophila* Stock Center | BDSC:27725; FlyBase: FBst0027725; | |
| Genetic reagent (*D. melanogaster*) | *D. melanogaster.* Expresses dsRNA for RNAi of ds (FBgn0284247) under UAS control in the VALIUM10 vector.: y[1] v[1]; P{y[+t7.7] v[+t1.8]=TRiP.JF02842}attP2 | Bloomington *Drosophila* Stock Center | BDSC:28008; FlyBase: FBst0028008; | |
| Genetic reagent (*D. melanogaster*) | *D. melanogaster.* Expresses dsRNA for RNAi of Hrs (FBgn0031450) under UAS control in the VALIUM10 vector.: y[1] v[1]; P{y[+t7.7] v[+t1.8]=TRiP.JF02860}attP2 | Bloomington *Drosophila* Stock Center | BDSC:28026; FlyBase: FBst0028026; | |

*Continued on next page*

*Continued*

| Reagent type (species) or resource | Designation | Source or reference | Identifiers | Additional information |
|---|---|---|---|---|
| Genetic reagent (*D. melanogaster*) | *D. melanogaster.* Expresses dsRNA for RNAi of mam (FBgn0002643) under UAS control in the VALIUM10 vector.: y[1] v[1]; P{y[+t7.7] v[+t1.8]=TRiP.JF02881}attP2 | Bloomington *Drosophila* Stock Center | BDSC:28046; FlyBase: FBst0028046; | |
| Genetic reagent (*D. melanogaster*) | *D. melanogaster.* Expresses dsRNA for RNAi of bun (FBgn0259176) under UAS control in the VALIUM10 vector.: y[1] v[1]; P{y[+t7.7] v[+t1.8]=TRiP.JF02954}attP2 | Bloomington *Drosophila* Stock Center | BDSC:28322; FlyBase: FBst0028322; | |
| Genetic reagent (*D. melanogaster*) | *D. melanogaster.* Expresses dsRNA for RNAi of aos (FBgn0004569) under UAS control in the VALIUM10 vector.: y[1] v[1]; P{y[+t7.7] v[+t1.8]=TRiP.JF03020}attP2 | Bloomington *Drosophila* Stock Center | BDSC:28383; FlyBase: FBst0028383; | |
| Genetic reagent (*D. melanogaster*) | *D. melanogaster.* Expresses dsRNA for RNAi of pcx (FBgn0003048) under UAS control in the VALIUM10 vector.: y[1] v[1]; P{y[+t7.7] v[+t1.8]=TRiP.HM05038}attP2/TM3, Sb[1] | Bloomington *Drosophila* Stock Center | BDSC:28552; FlyBase: FBst0028552; | |
| Genetic reagent (*D. melanogaster*) | *D. melanogaster.* Expresses dsRNA for RNAi of Su(fu) (FBgn0005355) under UAS control in the VALIUM10 vector.: y[1] v[1]; P{y[+t7.7] v[+t1.8]=TRiP.HM05045}attP2 | Bloomington *Drosophila* Stock Center | BDSC:28559; FlyBase: FBst0028559; | |
| Genetic reagent (*D. melanogaster*) | *D. melanogaster.* Expresses dsRNA for RNAi of Apc2 (FBgn0026598) under UAS control in the VALIUM10 vector.: y[1] v[1]; P{y[+t7.7] v[+t1.8]=TRiP.HM05073}attP2 | Bloomington *Drosophila* Stock Center | BDSC:28585; FlyBase: FBst0028585; | |
| Genetic reagent (*D. melanogaster*) | *D. melanogaster.* Expresses dsRNA for RNAi of l(2)tid (FBgn0002174) under UAS control in the VALIUM10 vector.: y[1] v[1]; P{y[+t7.7] v[+t1.8]=TRiP.HM05082}attP2 | Bloomington *Drosophila* Stock Center | BDSC:28594; FlyBase: FBst0028594; | |
| Genetic reagent (*D. melanogaster*) | *D. melanogaster.* Expresses dsRNA for RNAi of Su(H) (FBgn0004837) under UAS control in the VALIUM10 vector.: y[1] v[1]; P{y[+t7.7] v[+t1.8]=TRiP.HM05110}attP2/TM3, Sb[1] | Bloomington *Drosophila* Stock Center | BDSC:28900; FlyBase: FBst0028900; | |
| Genetic reagent (*D. melanogaster*) | *D. melanogaster.* Expresses dsRNA for RNAi of PDZ-GEF (FBgn0265778) under UAS control in the VALIUM10 vector.: y[1] v[1]; P{y[+t7.7] v[+t1.8]=TRiP.HM05139}attP2 | Bloomington *Drosophila* Stock Center | BDSC:28928; FlyBase: FBst0028928; | |
| Genetic reagent (*D. melanogaster*) | *D. melanogaster.* Expresses dsRNA for RNAi of wntD (FBgn0038134) under UAS control in the VALIUM10 vector.: y[1] v[1]; P{y[+t7.7] v[+t1.8]=TRiP.HM05158}attP2 | Bloomington *Drosophila* Stock Center | BDSC:28947; FlyBase: FBst0028947; | |

*Continued*

| Reagent type (species) or resource | Designation | Source or reference | Identifiers | Additional information |
|---|---|---|---|---|
| Genetic reagent (*D. melanogaster*) | *D. melanogaster.* Expresses dsRNA for RNAi of Cdk2 (FBgn0004107) under UAS control in the VALIUM10 vector.: y[1] v[1]; P{y[+t7.7] v[+t1.8]=TRiP.HM05163}attP2 | Bloomington *Drosophila* Stock Center | BDSC:28952; FlyBase: FBst0028952; | |
| Genetic reagent (*D. melanogaster*) | *D. melanogaster.* Expresses dsRNA for RNAi of ci (FBgn0004859) under UAS control in the VALIUM10 vector.: y[1] v[1]; P{y[+t7.7] v[+t1.8]=TRiP.JF01715}attP2 | Bloomington *Drosophila* Stock Center | BDSC:28984; FlyBase: FBst0028984; | |
| Genetic reagent (*D. melanogaster*) | *D. melanogaster.* Expresses dsRNA for RNAi of mgl (FBgn0261260) under UAS control in the VALIUM10 vector.: y[1] v[1]; P{y[+t7.7] v[+t1.8]=TRiP.JF02485}attP2 | Bloomington *Drosophila* Stock Center | BDSC:29324; FlyBase: FBst0029324; | |
| Genetic reagent (*D. melanogaster*) | *D. melanogaster.* Expresses dsRNA for RNAi of E(spl)m4-BFM (FBgn0002629) under UAS control in the VALIUM10 vector.: y[1] v[1]; P{y[+t7.7] v[+t1.8]=TRiP.JF03310}attP2 | Bloomington *Drosophila* Stock Center | BDSC:29378; FlyBase: FBst0029378; | |
| Genetic reagent (*D. melanogaster*) | *D. melanogaster.* Expresses dsRNA for RNAi of Wnt4 (FBgn0010453) under UAS control in the VALIUM10 vector.: y[1] v[1]; P{y[+t7.7] v[+t1.8]=TRiP.JF03378}attP2 | Bloomington *Drosophila* Stock Center | BDSC:29442; FlyBase: FBst0029442; | |
| Genetic reagent (*D. melanogaster*) | *D. melanogaster.* Expresses dsRNA for RNAi of Dif (FBgn0011274) under UAS control in the VALIUM10 vector.: y[1] sc[*] v[1] sev[21]; P{y[+t7.7] v[+t1.8]=TRiP.HM05257}attP2 | Bloomington *Drosophila* Stock Center | BDSC:30513; FlyBase: FBst0030513; | |
| Genetic reagent (*D. melanogaster*) | *D. melanogaster.* Expresses dsRNA for RNAi of InR (FBgn0283499) under UAS control in the VALIUM1 vector.: y[1] v[1]; P{y[+t7.7] v[+t1.8]=TRiP.JF01482}attP2 | Bloomington *Drosophila* Stock Center | BDSC:31037; FlyBase: FBst0031037; | |
| Genetic reagent (*D. melanogaster*) | *D. melanogaster.* Expresses dsRNA for RNAi of Cdc5 (FBgn0265574) and Roc1b (FBgn0040291) under UAS control in the VALIUM1 vector.: y[1] v[1]; P{y[+t7.7] v[+t1.8]=TRiP.JF01517}attP2 | Bloomington *Drosophila* Stock Center | BDSC:31067; FlyBase: FBst0031067; | |
| Genetic reagent (*D. melanogaster*) | *D. melanogaster.* Expresses dsRNA for RNAi of Egfr (FBgn0003731) under UAS control in the VALIUM1 vector.: y[1] v[1]; P{y[+t7.7] v[+t1.8]=TRiP.JF01696}attP2 | Bloomington *Drosophila* Stock Center | BDSC:31183; FlyBase: FBst0031183; | |
| Genetic reagent (*D. melanogaster*) | *D. melanogaster.* Expresses dsRNA for RNAi of norpA (FBgn0262738) under UAS control in the VALIUM1 vector.: y[1] v[1]; P{y[+t7.7] v[+t1.8]=TRiP.JF01713}attP2 | Bloomington *Drosophila* Stock Center | BDSC:31197; FlyBase: FBst0031197; | |

*Continued on next page*

*Continued*

| Reagent type (species) or resource | Designation | Source or reference | Identifiers | Additional information |
|---|---|---|---|---|
| Genetic reagent (*D. melanogaster*) | *D. melanogaster.* Expresses dsRNA for RNAi of CenG1A (FBgn0028509) under UAS control in the VALIUM1 vector.: y[1] v[1]; P{y[+t7.7] v[+t1.8]=TRiP.JF01807}attP2 | Bloomington *Drosophila* Stock Center | BDSC:31228; FlyBase: FBst0031228; | |
| Genetic reagent (*D. melanogaster*) | *D. melanogaster.* Expresses dsRNA for RNAi of dsh (FBgn0000499) under UAS control in the VALIUM1 vector.: y[1] v[1]; P{y[+t7.7] v[+t1.8]=TRiP.JF01254}attP2 | Bloomington *Drosophila* Stock Center | BDSC:31307; FlyBase: FBst0031307; | |
| Genetic reagent (*D. melanogaster*) | *D. melanogaster.* Expresses dsRNA for RNAi of Tl (FBgn0262473) under UAS control in the VALIUM1 vector.: y[1] v[1]; P{y[+t7.7] v[+t1.8]=TRiP.JF01276}attP2 | Bloomington *Drosophila* Stock Center | BDSC:31477; FlyBase: FBst0031477; | |
| Genetic reagent (*D. melanogaster*) | *D. melanogaster.* Expresses dsRNA for RNAi of l(2)gl (FBgn0002121) under UAS control in the VALIUM1 vector.: y[1] v[1]; P{y[+t7.7] v[+t1.8]=TRiP.JF01073}attP2 | Bloomington *Drosophila* Stock Center | BDSC:31517; FlyBase: FBst0031517; | |
| Genetic reagent (*D. melanogaster*) | *D. melanogaster.* Expresses dsRNA for RNAi of Jra (FBgn0001291) under UAS control in the VALIUM1 vector.: y[1] v[1]; P{y[+t7.7] v[+t1.8]=TRiP.JF01184}attP2 | Bloomington *Drosophila* Stock Center | BDSC:31595; FlyBase: FBst0031595; | |
| Genetic reagent (*D. melanogaster*) | *D. melanogaster.* Expresses dsRNA for RNAi of Axn (FBgn0026597) under UAS control in the VALIUM1 vector.: y[1] v[1]; P{y[+t7.7] v[+t1.8]=TRiP.HM04012}attP2 | Bloomington *Drosophila* Stock Center | BDSC:31705; FlyBase: FBst0031705; | |
| Genetic reagent (*D. melanogaster*) | *D. melanogaster.* Expresses dsRNA for RNAi of gig (FBgn0005198) under UAS control in the VALIUM1 vector.: y[1] v[1]; P{y[+t7.7] v[+t1.8]=TRiP.HM04083}attP2 | Bloomington *Drosophila* Stock Center | BDSC:31770; FlyBase: FBst0031770; | |
| Genetic reagent (*D. melanogaster*) | *D. melanogaster.* Expresses dsRNA for RNAi of Med (FBgn0011655) under UAS control in the VALIUM10 vector.: y[1] v[1]; P{y[+t7.7] v[+t1.8]=TRiP.JF02218}attP2 | Bloomington *Drosophila* Stock Center | BDSC:31928; FlyBase: FBst0031928; | |
| Genetic reagent (*D. melanogaster*) | *D. melanogaster.* Expresses dsRNA for RNAi of RagC-D (FBgn0033272) under UAS control in the VALIUM20 vector.: y[1] sc[*] v[1] sev[21]; P{y[+t7.7] v[+t1.8]=TRiP.HMS00333}attP2 | Bloomington *Drosophila* Stock Center | BDSC:32342; FlyBase: FBst0032342; | |
| Genetic reagent (*D. melanogaster*) | *D. melanogaster.* Expresses dsRNA for RNAi of AMPKalpha (FBgn0023169) under UAS control in the VALIUM20 vector.: y[1] sc[*] v[1] sev[21]; P{y[+t7.7] v[+t1.8]=TRiP.HMS00362}attP2 | Bloomington *Drosophila* Stock Center | BDSC:32371; FlyBase: FBst0032371; | |

*Continued on next page*

*Continued*

| Reagent type (species) or resource | Designation | Source or reference | Identifiers | Additional information |
|---|---|---|---|---|
| Genetic reagent (*D. melanogaster*) | *D. melanogaster.* Expresses dsRNA for RNAi of Rho1 (FBgn0014020) under UAS control in the VALIUM20 vector.: y[1] sc[*] v[1] sev[21]; P{y[+t7.7] v[+t1.8]=TRiP.HMS00375}attP2/TM3, Sb[1] | Bloomington *Drosophila* Stock Center | BDSC:32383; FlyBase: FBst0032383; | |
| Genetic reagent (*D. melanogaster*) | *D. melanogaster.* Expresses dsRNA for RNAi of pk (FBgn0003090) under UAS control in the VALIUM20 vector.: y[1] sc[*] v[1] sev[21]; P{y[+t7.7] v[+t1.8]=TRiP.HMS00408}attP2 | Bloomington *Drosophila* Stock Center | BDSC:32413; FlyBase: FBst0032413; | |
| Genetic reagent (*D. melanogaster*) | *D. melanogaster.* Expresses dsRNA for RNAi of RpS6 (FBgn0261592) under UAS control in the VALIUM20 vector.: y[1] sc[*] v[1] sev[21]; P{y[+t7.7] v[+t1.8]=TRiP.HMS00413}attP2 | Bloomington *Drosophila* Stock Center | BDSC:32418; FlyBase: FBst0032418; | |
| Genetic reagent (*D. melanogaster*) | *D. melanogaster.* Expresses dsRNA for RNAi of foxo (FBgn0038197) under UAS control in the VALIUM20 vector.: y[1] sc[*] v[1] sev[21]; P{y[+t7.7] v[+t1.8]=TRiP.HMS00422}attP2 | Bloomington *Drosophila* Stock Center | BDSC:32427; FlyBase: FBst0032427; | |
| Genetic reagent (*D. melanogaster*) | *D. melanogaster.* Expresses dsRNA for RNAi of Plc21C (FBgn0004611) under UAS control in the VALIUM20 vector.: y[1] sc[*] v[1] sev[21]; P{y[+t7.7] v[+t1.8]=TRiP.HMS00436}attP2/TM3, Sb[1] | Bloomington *Drosophila* Stock Center | BDSC:32438; FlyBase: FBst0032438; | |
| Genetic reagent (*D. melanogaster*) | *D. melanogaster.* Expresses dsRNA for RNAi of Ilp2 (FBgn0036046) under UAS control in the VALIUM20 vector.: y[1] sc[*] v[1] sev[21]; P{y[+t7.7] v[+t1.8]=TRiP.HMS00476}attP2/TM3, Sb[1] | Bloomington *Drosophila* Stock Center | BDSC:32475; FlyBase: FBst0032475; | |
| Genetic reagent (*D. melanogaster*) | *D. melanogaster.* Expresses dsRNA for RNAi of hh (FBgn0004644) under UAS control in the VALIUM20 vector.: y[1] sc[*] v[1] sev[21]; P{y[+t7.7] v[+t1.8]=TRiP.HMS00492}attP2/TM3, Sb[1] | Bloomington *Drosophila* Stock Center | BDSC:32489; FlyBase: FBst0032489; | |
| Genetic reagent (*D. melanogaster*) | *D. melanogaster.* Expresses dsRNA for RNAi of Spred (FBgn0020767) under UAS control in the VALIUM20 vector.: y[1] sc[*] v[1] sev[21]; P{y[+t7.7] v[+t1.8]=TRiP.HMS00637}attP2 | Bloomington *Drosophila* Stock Center | BDSC:32852; FlyBase: FBst0032852; | |
| Genetic reagent (*D. melanogaster*) | *D. melanogaster.* Expresses dsRNA for RNAi of upd3 (FBgn0053542) under UAS control in the VALIUM20 vector.: y[1] sc[*] v[1] sev[21]; P{y[+t7.7] v[+t1.8]=TRiP.HMS00646}attP2 | Bloomington *Drosophila* Stock Center | BDSC:32859; FlyBase: FBst0032859; | |

*Continued on next page*

*Continued*

| Reagent type (species) or resource | Designation | Source or reference | Identifiers | Additional information |
|---|---|---|---|---|
| Genetic reagent (*D. melanogaster*) | *D. melanogaster*. Expresses dsRNA for RNAi of Ilp1 (FBgn0044051) under UAS control in the VALIUM20 vector.: y[1] sc[*] v[1] sev[21]; P{y[+t7.7] v[+t1.8]=TRiP.HMS00648}attP2 | Bloomington *Drosophila* Stock Center | BDSC:32861; FlyBase: FBst0032861; | |
| Genetic reagent (*D. melanogaster*) | *D. melanogaster*. Expresses dsRNA for RNAi of Ilp7 (FBgn0044046) under UAS control in the VALIUM20 vector.: y[1] sc[*] v[1] sev[21]; P{y[+t7.7] v[+t1.8]=TRiP.HMS00649}attP2 | Bloomington *Drosophila* Stock Center | BDSC:32862; FlyBase: FBst0032862; | |
| Genetic reagent (*D. melanogaster*) | *D. melanogaster*. Expresses dsRNA for RNAi of SkpA (FBgn0025637) under UAS control in the VALIUM20 vector.: y[1] sc[*] v[1] sev[21]; P{y[+t7.7] v[+t1.8]=TRiP.HMS00657}attP2 | Bloomington *Drosophila* Stock Center | BDSC:32870; FlyBase: FBst0032870; | |
| Genetic reagent (*D. melanogaster*) | *D. melanogaster*. Expresses dsRNA for RNAi of CG3226 (FBgn0029882) under UAS control in the VALIUM20 vector.: y[1] sc[*] v[1] sev[21]; P{y[+t7.7] v[+t1.8]=TRiP.HMS00662}attP2 | Bloomington *Drosophila* Stock Center | BDSC:32875; FlyBase: FBst0032875; | |
| Genetic reagent (*D. melanogaster*) | *D. melanogaster*. Expresses dsRNA for RNAi of CtBP (FBgn0020496) under UAS control in the VALIUM20 vector.: y[1] sc[*] v[1] sev[21]; P{y[+t7.7] v[+t1.8]=TRiP.HMS00677}attP2 | Bloomington *Drosophila* Stock Center | BDSC:32889; FlyBase: FBst0032889; | |
| Genetic reagent (*D. melanogaster*) | *D. melanogaster*. Expresses dsRNA for RNAi of Hel89B (FBgn0022787) under UAS control in the VALIUM20 vector.: y[1] sc[*] v[1] sev[21]; P{y[+t7.7] v[+t1.8]=TRiP.HMS00684}attP2 | Bloomington *Drosophila* Stock Center | BDSC:32895; FlyBase: FBst0032895; | |
| Genetic reagent (*D. melanogaster*) | *D. melanogaster*. Expresses dsRNA for RNAi of Tctp (FBgn0037874) under UAS control in the VALIUM20 vector.: y[1] sc[*] v[1] sev[21]; P{y[+t7.7] v[+t1.8]=TRiP. HMS00701}attP2/TM3, Sb[1] | Bloomington *Drosophila* Stock Center | BDSC:32911; FlyBase: FBst0032911; | |
| Genetic reagent (*D. melanogaster*) | *D. melanogaster*. Expresses dsRNA for RNAi of jub (FBgn0030530) under UAS control in the VALIUM20 vector.: y[1] sc[*] v[1] sev[21]; P{y[+t7.7] v[+t1.8]=TRiP. HMS00714}attP2 | Bloomington *Drosophila* Stock Center | BDSC:32923; FlyBase: FBst0032923; | |
| Genetic reagent (*D. melanogaster*) | *D. melanogaster*. Expresses dsRNA for RNAi of Gug (FBgn0010825) under UAS control in the VALIUM20 vector.: y[1] sc[*] v[1] sev[21]; P{y[+t7.7] v[+t1.8]=TRiP.HMS00756}attP2 | Bloomington *Drosophila* Stock Center | BDSC:32961; FlyBase: FBst0032961; | |
| Genetic reagent (*D. melanogaster*) | *D. melanogaster*. Expresses dsRNA for RNAi of sav (FBgn0053193) under UAS control in the VALIUM20 vector.: y[1] sc[*] v[1] sev[21]; P{y[+t7.7] v[+t1.8]=TRiP.HMS00760}attP2 | Bloomington *Drosophila* Stock Center | BDSC:32965; FlyBase: FBst0032965; | |

*Continued on next page*

*Continued*

| Reagent type (species) or resource | Designation | Source or reference | Identifiers | Additional information |
|---|---|---|---|---|
| Genetic reagent (*D. melanogaster*) | *D. melanogaster*. Expresses dsRNA for RNAi of bsk (FBgn0000229) under UAS control in the VALIUM20 vector.: y[1] sc[*] v[1] sev[21]; P{y[+t7.7] v[+t1.8]=TRiP.HMS00777}attP2 | Bloomington *Drosophila* Stock Center | BDSC:32977; FlyBase: FBst0032977; | |
| Genetic reagent (*D. melanogaster*) | *D. melanogaster*. Expresses dsRNA for RNAi of Pgcl (FBgn0011822) under UAS control in the VALIUM20 vector.: y[1] sc[*] v[1] sev[21]; P{y[+t7.7] v[+t1.8]=TRiP.HMS00792}attP2/TM3, Sb[1] | Bloomington *Drosophila* Stock Center | BDSC:32992; FlyBase: FBst0032992; | |
| Genetic reagent (*D. melanogaster*) | *D. melanogaster*. Expresses dsRNA for RNAi of polo (FBgn0003124) under UAS control in the VALIUM20 vector.: y[1] sc[*] v[1] sev[21]; P{y[+t7.7] v[+t1.8]=TRiP. HMS00530}attP2 | Bloomington *Drosophila* Stock Center | BDSC:33042; FlyBase: FBst0033042; | |
| Genetic reagent (*D. melanogaster*) | *D. melanogaster*. Expresses dsRNA for RNAi of cnk (FBgn0286070) under UAS control in the VALIUM20 vector.: y[1] sc[*] v[1] sev[21]; P{y[+t7.7] v[+t1.8]=TRiP.HMS00238}attP2 | Bloomington *Drosophila* Stock Center | BDSC:33366; FlyBase: FBst0033366; | |
| Genetic reagent (*D. melanogaster*) | *D. melanogaster*. Expresses dsRNA for RNAi of kay (FBgn0001297) under UAS control in the VALIUM20 vector.: y[1] sc[*] v[1] sev[21]; P{y[+t7.7] v[+t1.8]=TRiP.HMS00254}attP2 | Bloomington *Drosophila* Stock Center | BDSC:33379; FlyBase: FBst0033379; | |
| Genetic reagent (*D. melanogaster*) | *D. melanogaster*. Expresses dsRNA for RNAi of apolpp (FBgn0087002) under UAS control in the VALIUM20 vector.: y[1] sc[*] v[1] sev[21]; P{y[+t7.7] v[+t1.8]=TRiP.HMS00265}attP2/TM3, Sb[1] | Bloomington *Drosophila* Stock Center | BDSC:33388; FlyBase: FBst0033388; | |
| Genetic reagent (*D. melanogaster*) | *D. melanogaster*. Expresses dsRNA for RNAi of Su(var)205 (FBgn0003607) under UAS control in the VALIUM20 vector.: y[1] sc[*] v[1] sev[21]; P{y[+t7.7] v[+t1.8]=TRiP.HMS00278}attP2 | Bloomington *Drosophila* Stock Center | BDSC:33400; FlyBase: FBst0033400; | |
| Genetic reagent (*D. melanogaster*) | *D. melanogaster*. Expresses dsRNA for RNAi of Tak1 (FBgn0026323) under UAS control in the VALIUM20 vector.: y[1] sc[*] v[1] sev[21]; P{y[+t7.7] v[+t1.8]=TRiP.HMS00282}attP2 | Bloomington *Drosophila* Stock Center | BDSC:33404; FlyBase: FBst0033404; | |
| Genetic reagent (*D. melanogaster*) | *D. melanogaster*. Expresses dsRNA for RNAi of hpo (FBgn0261456) under UAS control in the VALIUM20 vector.: y[1] v[1]; P{y[+t7.7] v[+t1.8]=TRiP.7.7 v[+t1.8]=TRiP.HMS00006}attP2 | Bloomington *Drosophila* Stock Center | BDSC:33614; FlyBase: FBst0033614; | |

*Continued on next page*

*Continued*

| Reagent type (species) or resource | Designation | Source or reference | Identifiers | Additional information |
|---|---|---|---|---|
| Genetic reagent (*D. melanogaster*) | *D. melanogaster.* Expresses dsRNA for RNAi of Akt1 (FBgn0010379) under UAS control in the VALIUM20 vector.: y[1] v[1]; P{y[+t7.7] v[+t1.8]=TRiP.7.7 v[+t1.8]=TRiP.HMS00007}attP2 | Bloomington *Drosophila* Stock Center | BDSC:33615; FlyBase: FBst0033615; | |
| Genetic reagent (*D. melanogaster*) | *D. melanogaster.* Expresses dsRNA for RNAi of N (FBgn0004647) under UAS control in the VALIUM20 vector.: y[1] v[1]; P{y[+t7.7] v[+t1.8]=TRiP.7.7 v[+t1.8]=TRiP.HMS00009}attP2 | Bloomington *Drosophila* Stock Center | BDSC:33616; FlyBase: FBst0033616; | |
| Genetic reagent (*D. melanogaster*) | *D. melanogaster.* Expresses dsRNA for RNAi of csw (FBgn0000382) under UAS control in the VALIUM20 vector.: y[1] v[1]; P{y[+t7.7] v[+t1.8]=TRiP.7.7 v[+t1.8]=TRiP.HMS00012}attP2 | Bloomington *Drosophila* Stock Center | BDSC:33619; FlyBase: FBst0033619; | |
| Genetic reagent (*D. melanogaster*) | *D. melanogaster.* Expresses dsRNA for RNAi of tor (FBgn0003733) under UAS control in the VALIUM20 vector.: y[1] v[1]; P{y[+t7.7] v[+t1.8]=TRiP.7.7 v[+t1.8]=TRiP.HMS00021}attP2 | Bloomington *Drosophila* Stock Center | BDSC:33627; FlyBase: FBst0033627; | |
| Genetic reagent (*D. melanogaster*) | *D. melanogaster.* Expresses dsRNA for RNAi of Stat92E (FBgn0016917) under UAS control in the VALIUM20 vector.: y[1] v[1]; P{y[+t7.7] v[+t1.8]=TRiP.7.7 v[+t1.8]=TRiP.HMS00035}attP2 | Bloomington *Drosophila* Stock Center | BDSC:33637; FlyBase: FBst0033637; | |
| Genetic reagent (*D. melanogaster*) | *D. melanogaster.* Expresses dsRNA for RNAi of Dsor1 (FBgn0010269) under UAS control in the VALIUM20 vector.: y[1] v[1]; P{y[+t7.7] v[+t1.8]=TRiP.7.7 v[+t1.8]=TRiP.HMS00037}attP2 | Bloomington *Drosophila* Stock Center | BDSC:33639; FlyBase: FBst0033639; | |
| Genetic reagent (*D. melanogaster*) | *D. melanogaster.* Expresses dsRNA for RNAi of Pten (FBgn0026379) under UAS control in the VALIUM20 vector.: y[1] v[1]; P{y[+t7.7] v[+t1.8]=TRiP.7.7 v[+t1.8]=TRiP.HMS00044}attP2 | Bloomington *Drosophila* Stock Center | BDSC:33643; FlyBase: FBst0033643; | |
| Genetic reagent (*D. melanogaster*) | *D. melanogaster.* Expresses dsRNA for RNAi of CycD (FBgn0010315) under UAS control in the VALIUM20 vector.: y[1] sc[*] v[1] sev[21]; P{y[+t7.7] v[+t1.8]=TRiP.HMS00059}attP2 | Bloomington *Drosophila* Stock Center | BDSC:33653; FlyBase: FBst0033653; | |
| Genetic reagent (*D. melanogaster*) | *D. melanogaster.* Expresses dsRNA for RNAi of Rel (FBgn0014018) under UAS control in the VALIUM20 vector.: y[1] sc[*] v[1] sev[21]; P{y[+t7.7] v[+t1.8]=TRiP.HMS00070}attP2 | Bloomington *Drosophila* Stock Center | BDSC:33661; FlyBase: FBst0033661; | |

*Continued on next page*

*Continued*

| Reagent type (species) or resource | Designation | Source or reference | Identifiers | Additional information |
|---|---|---|---|---|
| Genetic reagent (*D. melanogaster*) | *D. melanogaster.* Expresses dsRNA for RNAi of upd1 (FBgn0004956) under UAS control in the VALIUM20 vector.: y[1] sc[*] v[1] sev[21]; P{y[+t7.7] v[+t1.8]=TRiP.HMS00545}attP2 | Bloomington *Drosophila* Stock Center | BDSC:33680; FlyBase: FBst0033680; | |
| Genetic reagent (*D. melanogaster*) | *D. melanogaster.* Expresses dsRNA for RNAi of Ilp3 (FBgn0044050) under UAS control in the VALIUM20 vector.: y[1] sc[*] v[1] sev[21]; P{y[+t7.7] v[+t1.8]=TRiP.HMS00546}attP2 | Bloomington *Drosophila* Stock Center | BDSC:33681; FlyBase: FBst0033681; | |
| Genetic reagent (*D. melanogaster*) | *D. melanogaster.* Expresses dsRNA for RNAi of Ilp4 (FBgn0044049) under UAS control in the VALIUM20 vector.: y[1] sc[*] v[1] sev[21]; P{y[+t7.7] v[+t1.8]=TRiP.HMS00547}attP2 | Bloomington *Drosophila* Stock Center | BDSC:33682; FlyBase: FBst0033682; | |
| Genetic reagent (*D. melanogaster*) | *D. melanogaster.* Expresses dsRNA for RNAi of Ilp5 (FBgn0044048) under UAS control in the VALIUM20 vector.: y[1] sc[*] v[1] sev[21]; P{y[+t7.7] v[+t1.8]=TRiP.HMS00548}attP2 | Bloomington *Drosophila* Stock Center | BDSC:33683; FlyBase: FBst0033683; | |
| Genetic reagent (*D. melanogaster*) | *D. melanogaster.* Expresses dsRNA for RNAi of Ilp6 (FBgn0044047) under UAS control in the VALIUM20 vector.: y[1] sc[*] v[1] sev[21]; P{y[+t7.7] v[+t1.8]=TRiP.HMS00549}attP2 | Bloomington *Drosophila* Stock Center | BDSC:33684; FlyBase: FBst0033684; | |
| Genetic reagent (*D. melanogaster*) | *D. melanogaster.* Expresses dsRNA for RNAi of d (FBgn0262029) under UAS control in the VALIUM20 vector.: y[1] sc[*] v[1] sev[21]; P{y[+t7.7] v[+t1.8]=TRiP.HMS01096}attP2 | Bloomington *Drosophila* Stock Center | BDSC:33754; FlyBase: FBst0033754; | |
| Genetic reagent (*D. melanogaster*) | *D. melanogaster.* Expresses dsRNA for RNAi of Dad (FBgn0020493) under UAS control in the VALIUM20 vector.: y[1] sc[*] v[1] sev[21]; P{y[+t7.7] v[+t1.8]=TRiP.HMS01102}attP2 | Bloomington *Drosophila* Stock Center | BDSC:33759; FlyBase: FBst0033759; | |
| Genetic reagent (*D. melanogaster*) | *D. melanogaster.* Expresses dsRNA for RNAi of inaC (FBgn0004784) under UAS control in the VALIUM20 vector.: y[1] v[1]; P{y[+t7.7] v[+t1.8]=TRiP.7.7 v[+t1.8]=TRiP.JF02958}attP2/TM3, Sb[1] | Bloomington *Drosophila* Stock Center | BDSC:33768; FlyBase: FBst0033768; | |
| Genetic reagent (*D. melanogaster*) | *D. melanogaster.* Expresses dsRNA for RNAi of wg (FBgn0284084) under UAS control in the VALIUM20 vector.: y[1] sc[*] v[1] sev[21]; P{y[+t7.7] v[+t1.8]=TRiP.HMS00844}attP2 | Bloomington *Drosophila* Stock Center | BDSC:33902; FlyBase: FBst0033902; | |
| Genetic reagent (*D. melanogaster*) | *D. melanogaster.* Expresses dsRNA for RNAi of srl (FBgn0037248) under UAS control in the VALIUM20 vector.: y[1] sc[*] v[1] sev[21]; P{y[+t7.7] v[+t1.8]=TRiP.HMS00858}attP2 | Bloomington *Drosophila* Stock Center | BDSC:33915; FlyBase: FBst0033915; | |

*Continued on next page*

*Continued*

| Reagent type (species) or resource | Designation | Source or reference | Identifiers | Additional information |
|---|---|---|---|---|
| Genetic reagent (*D. melanogaster*) | *D. melanogaster.* Expresses dsRNA for RNAi of SkpC (FBgn0026175) under UAS control in the VALIUM20 vector.: y[1] sc[*] v[1] sev[21]; P{y[+t7.7] v[+t1.8]=TRiP.HMS00871}attP2 | Bloomington *Drosophila* Stock Center | BDSC:33925; FlyBase: FBst0033925; | |
| Genetic reagent (*D. melanogaster*) | *D. melanogaster.* Expresses dsRNA for RNAi of SPE (FBgn0039102) under UAS control in the VALIUM20 vector.: y[1] sc[*] v[1] sev[21]; P{y[+t7.7] v[+t1.8]=TRiP.HMS00873}attP2 | Bloomington *Drosophila* Stock Center | BDSC:33926; FlyBase: FBst0033926; | |
| Genetic reagent (*D. melanogaster*) | *D. melanogaster.* Expresses dsRNA for RNAi of Traf6 (FBgn0265464) under UAS control in the VALIUM20 vector.: y[1] sc[*] v[1] sev[21]; P{y[+t7.7] v[+t1.8]=TRiP.HMS00880}attP2 | Bloomington *Drosophila* Stock Center | BDSC:33931; FlyBase: FBst0033931; | |
| Genetic reagent (*D. melanogaster*) | *D. melanogaster.* Expresses dsRNA for RNAi of upd2 (FBgn0030904) under UAS control in the VALIUM20 vector.: y[1] sc[*] v[1] sev[21]; P{y[+t7.7] v[+t1.8]=TRiP.HMS00901}attP2 | Bloomington *Drosophila* Stock Center | BDSC:33949; FlyBase: FBst0033949; | |
| Genetic reagent (*D. melanogaster*) | *D. melanogaster.* Expresses dsRNA for RNAi of Tor (FBgn0021796) under UAS control in the VALIUM20 vector.: y[1] sc[*] v[1] sev[21]; P{y[+t7.7] v[+t1.8]=TRiP.HMS00904}attP2 | Bloomington *Drosophila* Stock Center | BDSC:33951; FlyBase: FBst0033951; | |
| Genetic reagent (*D. melanogaster*) | *D. melanogaster.* Expresses dsRNA for RNAi of dally (FBgn0263930) under UAS control in the VALIUM20 vector.: y[1] sc[*] v[1] sev[21]; P{y[+t7.7] v[+t1.8]=TRiP.HMS00905}attP2 | Bloomington *Drosophila* Stock Center | BDSC:33952; FlyBase: FBst0033952; | |
| Genetic reagent (*D. melanogaster*) | *D. melanogaster.* Expresses dsRNA for RNAi of Rheb (FBgn0041191) under UAS control in the VALIUM20 vector.: y[1] sc[*] v[1] sev[21]; P{y[+t7.7] v[+t1.8]=TRiP.HMS00923}attP2 | Bloomington *Drosophila* Stock Center | BDSC:33966; FlyBase: FBst0033966; | |
| Genetic reagent (*D. melanogaster*) | *D. melanogaster.* Expresses dsRNA for RNAi of Cat (FBgn0000261) under UAS control in the VALIUM20 vector.: y[1] sc[*] v[1] sev[21]; P{y[+t7.7] v[+t1.8]=TRiP.HMS00990}attP2 | Bloomington *Drosophila* Stock Center | BDSC:34020; FlyBase: FBst0034020; | |
| Genetic reagent (*D. melanogaster*) | *D. melanogaster.* Expresses dsRNA for RNAi of CycK (FBgn0025674) under UAS control in the VALIUM20 vector.: y[1] sc[*] v[1] sev[21]; P{y[+t7.7] v[+t1.8]=TRiP.HMS01003}attP2 | Bloomington *Drosophila* Stock Center | BDSC:34032; FlyBase: FBst0034032; | |
| Genetic reagent (*D. melanogaster*) | *D. melanogaster.* Expresses dsRNA for RNAi of wts (FBgn0011739) under UAS control in the VALIUM20 vector.: y[1] v[1]; P{y[+t7.7] v[+t1.8]=TRiP.7.7 v[+t1.8]=TRiP.HMS00026}attP2 | Bloomington *Drosophila* Stock Center | BDSC:34064; FlyBase: FBst0034064; | |

*Continued on next page*

*Continued*

| Reagent type (species) or resource | Designation | Source or reference | Identifiers | Additional information |
|---|---|---|---|---|
| Genetic reagent (*D. melanogaster*) | *D. melanogaster.* Expresses dsRNA for RNAi of yki (FBgn0034970) under UAS control in the VALIUM20 vector.: y[1] v[1]; P{y[+t7.7] v[+t1.8]=TRiP.7.7 v[+t1.8]=TRiP.HMS00041}attP2 | Bloomington *Drosophila* Stock Center | BDSC:34067; FlyBase: FBst0034067; | |
| Genetic reagent (*D. melanogaster*) | *D. melanogaster.* Expresses dsRNA for RNAi of srp (FBgn0003507) under UAS control in the VALIUM20 vector.: y[1] sc[*] v[1] sev[21]; P{y[+t7.7] v[+t1.8]=TRiP.HMS01083}attP2 | Bloomington *Drosophila* Stock Center | BDSC:34080; FlyBase: FBst0034080; | |
| Genetic reagent (*D. melanogaster*) | *D. melanogaster.* Expresses dsRNA for RNAi of Fas2 (FBgn0000635) under UAS control in the VALIUM20 vector.: y[1] sc[*] v[1] sev[21]; P{y[+t7.7] v[+t1.8]=TRiP.HMS01098}attP2 | Bloomington *Drosophila* Stock Center | BDSC:34084; FlyBase: FBst0034084; | |
| Genetic reagent (*D. melanogaster*) | *D. melanogaster.* Expresses dsRNA for RNAi of eIF4E1 (FBgn0015218) under UAS control in the VALIUM20 vector.: y[1] sc[*] v[1] sev[21]; P{y[+t7.7] v[+t1.8]=TRiP.HMS00969}attP2 | Bloomington *Drosophila* Stock Center | BDSC:34096; FlyBase: FBst0034096; | |
| Genetic reagent (*D. melanogaster*) | *D. melanogaster.* Expresses dsRNA for RNAi of Sema1a (FBgn0011259) under UAS control in the VALIUM20 vector.: y[1] sc[*] v[1] sev[21]; P{y[+t7.7] v[+t1.8]=TRiP.HMS01307}attP2 | Bloomington *Drosophila* Stock Center | BDSC:34320; FlyBase: FBst0034320; | |
| Genetic reagent (*D. melanogaster*) | *D. melanogaster.* Expresses dsRNA for RNAi of fz (FBgn0001085) under UAS control in the VALIUM20 vector.: y[1] sc[*] v[1] sev[21]; P{y[+t7.7] v[+t1.8]=TRiP.HMS01308}attP2 | Bloomington *Drosophila* Stock Center | BDSC:34321; FlyBase: FBst0034321; | |
| Genetic reagent (*D. melanogaster*) | *D. melanogaster.* Expresses dsRNA for RNAi of fj (FBgn0000658) under UAS control in the VALIUM20 vector.: y[1] sc[*] v[1] sev[21]; P{y[+t7.7] v[+t1.8]=TRiP.HMS01310}attP2 | Bloomington *Drosophila* Stock Center | BDSC:34323; FlyBase: FBst0034323; | |
| Genetic reagent (*D. melanogaster*) | *D. melanogaster.* Expresses dsRNA for RNAi of eve (FBgn0000606) under UAS control in the VALIUM20 vector.: y[1] sc[*] v[1] sev[21]; P{y[+t7.7] v[+t1.8]=TRiP.HMS01312}attP2 | Bloomington *Drosophila* Stock Center | BDSC:34325; FlyBase: FBst0034325; | |
| Genetic reagent (*D. melanogaster*) | *D. melanogaster.* Expresses dsRNA for RNAi of tll (FBgn0003720) under UAS control in the VALIUM20 vector.: y[1] sc[*] v[1] sev[21]; P{y[+t7.7] v[+t1.8]=TRiP.HMS01316}attP2 | Bloomington *Drosophila* Stock Center | BDSC:34329; FlyBase: FBst0034329; | |
| Genetic reagent (*D. melanogaster*) | *D. melanogaster.* Expresses dsRNA for RNAi of aPKC (FBgn0261854) under UAS control in the VALIUM20 vector.: y[1] sc[*] v[1] sev[21]; P{y[+t7.7] v[+t1.8]=TRiP.HMS01320}attP2 | Bloomington *Drosophila* Stock Center | BDSC:34332; FlyBase: FBst0034332; | |

*Continued on next page*

*Continued*

| Reagent type (species) or resource | Designation | Source or reference | Identifiers | Additional information |
|---|---|---|---|---|
| Genetic reagent (*D. melanogaster*) | *D. melanogaster.* Expresses dsRNA for RNAi of Vang (FBgn0015838) under UAS control in the VALIUM20 vector.: y[1] sc[*] v[1] sev[21]; P{y[+t7.7] v[+t1.8]=TRiP.HMS01343}attP2 | Bloomington *Drosophila* Stock Center | BDSC:34354; FlyBase: FBst0034354; | |
| Genetic reagent (*D. melanogaster*) | *D. melanogaster.* Expresses dsRNA for RNAi of Lst8 (FBgn0264691) under UAS control in the VALIUM20 vector.: y[1] sc[*] v[1] sev[21]; P{y[+t7.7] v[+t1.8]=TRiP.HMS01350}attP2 | Bloomington *Drosophila* Stock Center | BDSC:34361; FlyBase: FBst0034361; | |
| Genetic reagent (*D. melanogaster*) | *D. melanogaster.* Expresses dsRNA for RNAi of Lkb1 (FBgn0038167) under UAS control in the VALIUM20 vector.: y[1] sc[*] v[1] sev[21]; P{y[+t7.7] v[+t1.8]=TRiP.HMS01351}attP2 | Bloomington *Drosophila* Stock Center | BDSC:34362; FlyBase: FBst0034362; | |
| Genetic reagent (*D. melanogaster*) | *D. melanogaster.* Expresses dsRNA for RNAi of puc (FBgn0243512) under UAS control in the VALIUM20 vector.: y[1] sc[*] v[1] sev[21]; P{y[+t7.7] v[+t1.8]=TRiP. HMS01386}attP2/TM3, Sb[1] | Bloomington *Drosophila* Stock Center | BDSC:34392; FlyBase: FBst0034392; | |
| Genetic reagent (*D. melanogaster*) | *D. melanogaster.* Expresses dsRNA for RNAi of Sxl (FBgn0264270) under UAS control in the VALIUM20 vector.: y[1] sc[*] v[1] sev[21]; P{y[+t7.7] v[+t1.8]=TRiP. HMS00609}attP2 | Bloomington *Drosophila* Stock Center | BDSC:34393; FlyBase: FBst0034393; | |
| Genetic reagent (*D. melanogaster*) | *D. melanogaster.* Expresses dsRNA for RNAi of Cka (FBgn0044323) under UAS control in the VALIUM20 vector.: y[1] sc[*] v[1] sev[21]; P{y[+t7.7] v[+t1.8]=TRiP. HMS00081}attP2 | Bloomington *Drosophila* Stock Center | BDSC:34522; FlyBase: FBst0034522; | |
| Genetic reagent (*D. melanogaster*) | *D. melanogaster.* Expresses dsRNA for RNAi of bnl (FBgn0014135) under UAS control in the VALIUM20 vector.: y[1] sc[*] v[1] sev[21]; P{y[+t7.7] v[+t1.8]=TRiP.HMS01046}attP2 | Bloomington *Drosophila* Stock Center | BDSC:34572; FlyBase: FBst0034572; | |
| Genetic reagent (*D. melanogaster*) | *D. melanogaster.* Expresses dsRNA for RNAi of RagA-B (FBgn0037647) under UAS control in the VALIUM20 vector.: y[1] sc[*] v[1] sev[21]; P{y[+t7.7] v[+t1.8]=TRiP.HMS01064}attP2 | Bloomington *Drosophila* Stock Center | BDSC:34590; FlyBase: FBst0034590; | |
| Genetic reagent (*D. melanogaster*) | *D. melanogaster.* Expresses dsRNA for RNAi of Wbp2 (FBgn0036318) under UAS control in the VALIUM20 vector.: y[1] sc[*] v[1] sev[21]; P{y[+t7.7] v[+t1.8]=TRiP.HMS00563}attP2 | Bloomington *Drosophila* Stock Center | BDSC:34603; FlyBase: FBst0034603; | |
| Genetic reagent (*D. melanogaster*) | *D. melanogaster.* Expresses dsRNA for RNAi of pip (FBgn0003089) under UAS control in the VALIUM20 vector.: y[1] sc[*] v[1] sev[21]; P{y[+t7.7] v[+t1.8]=TRiP.HMS01288}attP2 | Bloomington *Drosophila* Stock Center | BDSC:34613; FlyBase: FBst0034613; | |

*Continued on next page*

*Continued*

| Reagent type (species) or resource | Designation | Source or reference | Identifiers | Additional information |
|---|---|---|---|---|
| Genetic reagent (*D. melanogaster*) | *D. melanogaster*. Expresses dsRNA for RNAi of dome (FBgn0043903) under UAS control in the VALIUM20 vector.: y[1] sc[*] v[1] sev[21]; P{y[+t7.7] v[+t1.8]=TRiP.HMS01293}attP2 | Bloomington *Drosophila* Stock Center | BDSC:34618; FlyBase: FBst0034618; | |
| Genetic reagent (*D. melanogaster*) | *D. melanogaster*. Expresses dsRNA for RNAi of Stlk (FBgn0046692) under UAS control in the VALIUM20 vector.: y[1] sc[*] v[1] sev[21]; P{y[+t7.7] v[+t1.8]=TRiP. HMS01295}attP2/TM3, Sb[1] | Bloomington *Drosophila* Stock Center | BDSC:34620; FlyBase: FBst0034620; | |
| Genetic reagent (*D. melanogaster*) | *D. melanogaster*. Expresses dsRNA for RNAi of slp1 (FBgn0003430) under UAS control in the VALIUM20 vector.: y[1] sc[*] v[1] sev[21]; P{y[+t7.7] v[+t1.8]=TRiP. HMS01107}attP2 | Bloomington *Drosophila* Stock Center | BDSC:34633; FlyBase: FBst0034633; | |
| Genetic reagent (*D. melanogaster*) | *D. melanogaster*. Expresses dsRNA for RNAi of slp2 (FBgn0004567) under UAS control in the VALIUM20 vector.: y[1] sc[*] v[1] sev[21]; P{y[+t7.7] v[+t1.8]=TRiP.HMS01108}attP2 | Bloomington *Drosophila* Stock Center | BDSC:34634; FlyBase: FBst0034634; | |
| Genetic reagent (*D. melanogaster*) | *D. melanogaster*. Expresses dsRNA for RNAi of hth (FBgn0001235) under UAS control in the VALIUM20 vector.: y[1] sc[*] v[1] sev[21]; P{y[+t7.7] v[+t1.8]=TRiP.HMS01112}attP2 | Bloomington *Drosophila* Stock Center | BDSC:34637; FlyBase: FBst0034637; | |
| Genetic reagent (*D. melanogaster*) | *D. melanogaster*. Expresses dsRNA for RNAi of RhoGEF2 (FBgn0023172) under UAS control in the VALIUM20 vector.: y[1] sc[*] v[1] sev[21]; P{y[+t7.7] v[+t1.8]=TRiP.HMS01118}attP2 | Bloomington *Drosophila* Stock Center | BDSC:34643; FlyBase: FBst0034643; | |
| Genetic reagent (*D. melanogaster*) | *D. melanogaster*. Expresses dsRNA for RNAi of Wnt5 (FBgn0010194) under UAS control in the VALIUM20 vector.: y[1] sc[*] v[1] sev[21]; P{y[+t7.7] v[+t1.8]=TRiP.HMS01119}attP2 | Bloomington *Drosophila* Stock Center | BDSC:34644; FlyBase: FBst0034644; | |
| Genetic reagent (*D. melanogaster*) | *D. melanogaster*. Expresses dsRNA for RNAi of spi (FBgn0005672) under UAS control in the VALIUM20 vector.: y[1] sc[*] v[1] sev[21]; P{y[+t7.7] v[+t1.8]=TRiP.HMS01120}attP2 | Bloomington *Drosophila* Stock Center | BDSC:34645; FlyBase: FBst0034645; | |
| Genetic reagent (*D. melanogaster*) | *D. melanogaster*. Expresses dsRNA for RNAi of Galphao (FBgn0001122) under UAS control in the VALIUM20 vector.: y[1] sc[*] v[1] sev[21]; P{y[+t7.7] v[+t1.8]=TRiP.HMS01129}attP2 | Bloomington *Drosophila* Stock Center | BDSC:34653; FlyBase: FBst0034653; | |
| Genetic reagent (*D. melanogaster*) | *D. melanogaster*. Expresses dsRNA for RNAi of shn (FBgn0003396) under UAS control in the VALIUM20 vector.: y[1] sc[*] v[1] sev[21]; P{y[+t7.7] v[+t1.8]=TRiP. HMS01167}attP2/TM3, Sb[1] | Bloomington *Drosophila* Stock Center | BDSC:34689; FlyBase: FBst0034689; | |

*Continued on next page*

*Continued*

| Reagent type (species) or resource | Designation | Source or reference | Identifiers | Additional information |
|---|---|---|---|---|
| Genetic reagent (*D. melanogaster*) | *D. melanogaster.* Expresses dsRNA for RNAi of Ser (FBgn0004197) under UAS control in the VALIUM20 vector.: y[1] sc[*] v[1] sev[21]; P{y[+t7.7] v[+t1.8]=TRiP.HMS01179}attP2 | Bloomington *Drosophila* Stock Center | BDSC:34700; FlyBase: FBst0034700; | |
| Genetic reagent (*D. melanogaster*) | *D. melanogaster.* Expresses dsRNA for RNAi of H (FBgn0001169) under UAS control in the VALIUM20 vector.: y[1] sc[*] v[1] sev[21]; P{y[+t7.7] v[+t1.8]=TRiP.HMS01182}attP2 | Bloomington *Drosophila* Stock Center | BDSC:34703; FlyBase: FBst0034703; | |
| Genetic reagent (*D. melanogaster*) | *D. melanogaster.* Expresses dsRNA for RNAi of opa (FBgn0003002) under UAS control in the VALIUM20 vector.: y[1] sc[*] v[1] sev[21]; P{y[+t7.7] v[+t1.8]=TRiP. HMS01185}attP2/TM3, Sb[1] | Bloomington *Drosophila* Stock Center | BDSC:34706; FlyBase: FBst0034706; | |
| Genetic reagent (*D. melanogaster*) | *D. melanogaster.* Expresses dsRNA for RNAi of Pkc53E (FBgn0003091) under UAS control in the VALIUM20 vector.: y[1] sc[*] v[1] sev[21]; P{y[+t7.7] v[+t1.8]=TRiP.HMS01195}attP2 | Bloomington *Drosophila* Stock Center | BDSC:34716; FlyBase: FBst0034716; | |
| Genetic reagent (*D. melanogaster*) | *D. melanogaster.* Expresses dsRNA for RNAi of hkb (FBgn0261434) under UAS control in the VALIUM20 vector.: y[1] sc[*] v[1] sev[21]; P{y[+t7.7] v[+t1.8]=TRiP. HMS01216}attP2/TM3, Sb[1] | Bloomington *Drosophila* Stock Center | BDSC:34736; FlyBase: FBst0034736; | |
| Genetic reagent (*D. melanogaster*) | *D. melanogaster.* Expresses dsRNA for RNAi of gig (FBgn0005198) under UAS control in the VALIUM20 vector.: y[1] sc[*] v[1] sev[21]; P{y[+t7.7] v[+t1.8]=TRiP. HMS01217}attP2/TM3, Sb[1] | Bloomington *Drosophila* Stock Center | BDSC:34737; FlyBase: FBst0034737; | |
| Genetic reagent (*D. melanogaster*) | *D. melanogaster.* Expresses dsRNA for RNAi of ken (FBgn0011236) under UAS control in the VALIUM20 vector.: y[1] sc[*] v[1] sev[21]; P{y[+t7.7] v[+t1.8]=TRiP.HMS01219}attP2 | Bloomington *Drosophila* Stock Center | BDSC:34739; FlyBase: FBst0034739; | |
| Genetic reagent (*D. melanogaster*) | *D. melanogaster.* Expresses dsRNA for RNAi of cact (FBgn0000250) under UAS control in the VALIUM20 vector.: y[1] sc[*] v[1] sev[21]; P{y[+t7.7] v[+t1.8]=TRiP.HMS00084}attP2 | Bloomington *Drosophila* Stock Center | BDSC:34775; FlyBase: FBst0034775; | |
| Genetic reagent (*D. melanogaster*) | *D. melanogaster.* Expresses dsRNA for RNAi of Bx42 (FBgn0004856) under UAS control in the VALIUM20 vector.: y[1] sc[*] v[1] sev[21]; P{y[+t7.7] v[+t1.8]=TRiP.HMS00086}attP2 | Bloomington *Drosophila* Stock Center | BDSC:34777; FlyBase: FBst0034777; | |
| Genetic reagent (*D. melanogaster*) | *D. melanogaster.* Expresses dsRNA for RNAi of Rpt2 (FBgn0015282) under UAS control in the VALIUM20 vector.: y[1] sc[*] v[1] sev[21]; P{y[+t7.7] v[+t1.8]=TRiP.HMS00104}attP2 | Bloomington *Drosophila* Stock Center | BDSC:34795; FlyBase: FBst0034795; | |

*Continued on next page*

*Continued*

| Reagent type (species) or resource | Designation | Source or reference | Identifiers | Additional information |
|---|---|---|---|---|
| Genetic reagent (*D. melanogaster*) | *D. melanogaster.* Expresses dsRNA for RNAi of Skp2 (FBgn0037236) under UAS control in the VALIUM20 vector.: y[1] sc[*] v[1] sev[21]; P{y[+t7.7] v[+t1.8]=TRiP. HMS00116}attP2/TM3, Sb[1] | Bloomington *Drosophila* Stock Center | BDSC:34807; FlyBase: FBst0034807; | |
| Genetic reagent (*D. melanogaster*) | *D. melanogaster.* Expresses dsRNA for RNAi of raptor (FBgn0029840) under UAS control in the VALIUM20 vector.: y[1] sc[*] v[1] sev[21]; P{y[+t7.7] v[+t1.8]=TRiP. HMS00124}attP2 | Bloomington *Drosophila* Stock Center | BDSC:34814; FlyBase: FBst0034814; | |
| Genetic reagent (*D. melanogaster*) | *D. melanogaster.* Expresses dsRNA for RNAi of Sos (FBgn0001965) under UAS control in the VALIUM20 vector.: y[1] sc[*] v[1] sev[21]; P{y[+t7.7] v[+t1.8]=TRiP.HMS00149}attP2 | Bloomington *Drosophila* Stock Center | BDSC:34833; FlyBase: FBst0034833; | |
| Genetic reagent (*D. melanogaster*) | *D. melanogaster.* Expresses dsRNA for RNAi of Gprk2 (FBgn0261988) under UAS control in the VALIUM20 vector.: y[1] sc[*] v[1] sev[21]; P{y[+t7.7] v[+t1.8]=TRiP.HMS00161}attP2 | Bloomington *Drosophila* Stock Center | BDSC:34843; FlyBase: FBst0034843; | |
| Genetic reagent (*D. melanogaster*) | *D. melanogaster.* Expresses dsRNA for RNAi of rl (FBgn0003256) under UAS control in the VALIUM20 vector.: y[1] sc[*] v[1] sev[21]; P{y[+t7.7] v[+t1.8]=TRiP.HMS00173}attP2 | Bloomington *Drosophila* Stock Center | BDSC:34855; FlyBase: FBst0034855; | |
| Genetic reagent (*D. melanogaster*) | *D. melanogaster.* Expresses dsRNA for RNAi of mwh (FBgn0264272) under UAS control in the VALIUM20 vector.: y[1] sc[*] v[1] sev[21]; P{y[+t7.7] v[+t1.8]=TRiP.HMS00180}attP2 | Bloomington *Drosophila* Stock Center | BDSC:34862; FlyBase: FBst0034862; | |
| Genetic reagent (*D. melanogaster*) | *D. melanogaster.* Expresses dsRNA for RNAi of Apc (FBgn0015589) under UAS control in the VALIUM20 vector.: y[1] sc[*] v[1] sev[21]; P{y[+t7.7] v[+t1.8]=TRiP.HMS00188}attP2 | Bloomington *Drosophila* Stock Center | BDSC:34869; FlyBase: FBst0034869; | |
| Genetic reagent (*D. melanogaster*) | *D. melanogaster.* Expresses dsRNA for RNAi of Tao (FBgn0031030) under UAS control in the VALIUM20 vector.: y[1] sc[*] v[1] sev[21]; P{y[+t7.7] v[+t1.8]=TRiP.HMS01226}attP2 | Bloomington *Drosophila* Stock Center | BDSC:34881; FlyBase: FBst0034881; | |
| Genetic reagent (*D. melanogaster*) | *D. melanogaster.* Expresses dsRNA for RNAi of 14-3-3epsilon (FBgn0020238) under UAS control in the VALIUM20 vector.: y[1] sc[*] v[1] sev[21]; P{y[+t7.7] v[+t1.8]=TRiP.HMS01229}attP2 | Bloomington *Drosophila* Stock Center | BDSC:34884; FlyBase: FBst0034884; | |
| Genetic reagent (*D. melanogaster*) | *D. melanogaster.* Expresses dsRNA for RNAi of Sara (FBgn0026369) under UAS control in the VALIUM20 vector.: y[1] sc[*] v[1] sev[21]; P{y[+t7.7] v[+t1.8]=TRiP.HMS01239}attP2 | Bloomington *Drosophila* Stock Center | BDSC:34894; FlyBase: FBst0034894; | |

*Continued on next page*

*Continued*

| Reagent type (species) or resource | Designation | Source or reference | Identifiers | Additional information |
|---|---|---|---|---|
| Genetic reagent (*D. melanogaster*) | *D. melanogaster*. Expresses dsRNA for RNAi of gbb (FBgn0024234) under UAS control in the VALIUM20 vector.: y[1] sc[*] v[1] sev[21]; P{y[+t7.7] v[+t1.8]=TRiP.HMS01243}attP2 | Bloomington *Drosophila* Stock Center | BDSC:34898; FlyBase: FBst0034898; | |
| Genetic reagent (*D. melanogaster*) | *D. melanogaster*. Expresses dsRNA for RNAi of Atg8b (FBgn0038539) under UAS control in the VALIUM20 vector.: y[1] sc[*] v[1] sev[21]; P{y[+t7.7] v[+t1.8]=TRiP.HMS01245}attP2 | Bloomington *Drosophila* Stock Center | BDSC:34900; FlyBase: FBst0034900; | |
| Genetic reagent (*D. melanogaster*) | *D. melanogaster*. Expresses dsRNA for RNAi of Rac1 (FBgn0010333) under UAS control in the VALIUM20 vector.: y[1] sc[*] v[1] sev[21]; P{y[+t7.7] v[+t1.8]=TRiP.HMS01258}attP2 | Bloomington *Drosophila* Stock Center | BDSC:34910; FlyBase: FBst0034910; | |
| Genetic reagent (*D. melanogaster*) | *D. melanogaster*. Expresses dsRNA for RNAi of ato (FBgn0010433) under UAS control in the VALIUM20 vector.: y[1] sc[*] v[1] sev[21]; P{y[+t7.7] v[+t1.8]=TRiP.HMS01278}attP2 | Bloomington *Drosophila* Stock Center | BDSC:34929; FlyBase: FBst0034929; | |
| Genetic reagent (*D. melanogaster*) | *D. melanogaster*. Expresses dsRNA for RNAi of dl (FBgn0260632) under UAS control in the VALIUM20 vector.: y[1] sc[*] v[1] sev[21]; P{y[+t7.7] v[+t1.8]=TRiP.HMS00028}attP2 | Bloomington *Drosophila* Stock Center | BDSC:34938; FlyBase: FBst0034938; | |
| Genetic reagent (*D. melanogaster*) | *D. melanogaster*. Expresses dsRNA for RNAi of Mer (FBgn0086384) under UAS control in the VALIUM20 vector.: y[1] sc[*] v[1] sev[21]; P{y[+t7.7] v[+t1.8]=TRiP.HMS00459}attP2 | Bloomington *Drosophila* Stock Center | BDSC:34958; FlyBase: FBst0034958; | |
| Genetic reagent (*D. melanogaster*) | *D. melanogaster*. Expresses dsRNA for RNAi of mats (FBgn0038965) under UAS control in the VALIUM20 vector.: y[1] sc[*] v[1] sev[21]; P{y[+t7.7] v[+t1.8]=TRiP.HMS00475}attP2 | Bloomington *Drosophila* Stock Center | BDSC:34959; FlyBase: FBst0034959; | |
| Genetic reagent (*D. melanogaster*) | *D. melanogaster*. Expresses dsRNA for RNAi of ex (FBgn0004583) under UAS control in the VALIUM20 vector.: y[1] sc[*] v[1] sev[21]; P{y[+t7.7] v[+t1.8]=TRiP.HMS00874}attP2 | Bloomington *Drosophila* Stock Center | BDSC:34968; FlyBase: FBst0034968; | |
| Genetic reagent (*D. melanogaster*) | *D. melanogaster*. Expresses dsRNA for RNAi of ft (FBgn0001075) under UAS control in the VALIUM20 vector.: y[1] sc[*] v[1] sev[21]; P{y[+t7.7] v[+t1.8]=TRiP.HMS00932}attP2 | Bloomington *Drosophila* Stock Center | BDSC:34970; FlyBase: FBst0034970; | |
| Genetic reagent (*D. melanogaster*) | *D. melanogaster*. Expresses dsRNA for RNAi of msk (FBgn0026252) under UAS control in the VALIUM20 vector.: y[1] sc[*] v[1] sev[21]; P{y[+t7.7] v[+t1.8]=TRiP.HMS01408}attP2 | Bloomington *Drosophila* Stock Center | BDSC:34998; FlyBase: FBst0034998; | |

*Continued on next page*

*Continued*

| Reagent type (species) or resource | Designation | Source or reference | Identifiers | Additional information |
|---|---|---|---|---|
| Genetic reagent (*D. melanogaster*) | *D. melanogaster.* Expresses dsRNA for RNAi of baz (FBgn0000163) under UAS control in the VALIUM20 vector.: y[1] sc[*] v[1] sev[21]; P{y[+t7.7] v[+t1.8]=TRiP. HMS01412}attP2/TM3, Sb[1] | Bloomington *Drosophila* Stock Center | BDSC:35002; FlyBase: FBst0035002; | |
| Genetic reagent (*D. melanogaster*) | *D. melanogaster.* Expresses dsRNA for RNAi of arm (FBgn0000117) under UAS control in the VALIUM20 vector.: y[1] sc[*] v[1] sev[21]; P{y[+t7.7] v[+t1.8]=TRiP.HMS01414}attP2 | Bloomington *Drosophila* Stock Center | BDSC:35004; FlyBase: FBst0035004; | |
| Genetic reagent (*D. melanogaster*) | *D. melanogaster.* Expresses dsRNA for RNAi of Stam (FBgn0027363) under UAS control in the VALIUM20 vector.: y[1] sc[*] v[1] sev[21]; P{y[+t7.7] v[+t1.8]=TRiP.HMS01429}attP2 | Bloomington *Drosophila* Stock Center | BDSC:35016; FlyBase: FBst0035016; | |
| Genetic reagent (*D. melanogaster*) | *D. melanogaster.* Expresses dsRNA for RNAi of Gadd45 (FBgn0033153) under UAS control in the VALIUM20 vector.: y[1] sc[*] v[1] sev[21]; P{y[+t7.7] v[+t1.8]=TRiP.HMS01436}attP2 | Bloomington *Drosophila* Stock Center | BDSC:35023; FlyBase: FBst0035023; | |
| Genetic reagent (*D. melanogaster*) | *D. melanogaster.* Expresses dsRNA for RNAi of htl (FBgn0010389) under UAS control in the VALIUM20 vector.: y[1] sc[*] v[1] sev[21]; P{y[+t7.7] v[+t1.8]=TRiP.HMS01437}attP2 | Bloomington *Drosophila* Stock Center | BDSC:35024; FlyBase: FBst0035024; | |
| Genetic reagent (*D. melanogaster*) | *D. melanogaster.* Expresses dsRNA for RNAi of tsh (FBgn0003866) under UAS control in the VALIUM20 vector.: y[1] sc[*] v[1] sev[21]; P{y[+t7.7] v[+t1.8]=TRiP.HMS01443}attP2 | Bloomington *Drosophila* Stock Center | BDSC:35030; FlyBase: FBst0035030; | |
| Genetic reagent (*D. melanogaster*) | *D. melanogaster.* Expresses dsRNA for RNAi of Socs36E (FBgn0041184) under UAS control in the VALIUM20 vector.: y[1] sc[*] v[1] sev[21]; P{y[+t7.7] v[+t1.8]=TRiP.HMS01450}attP2 | Bloomington *Drosophila* Stock Center | BDSC:35036; FlyBase: FBst0035036; | |
| Genetic reagent (*D. melanogaster*) | *D. melanogaster.* Expresses dsRNA for RNAi of Cad99C (FBgn0039709) under UAS control in the VALIUM20 vector.: y[1] sc[*] v[1] sev[21]; P{y[+t7.7] v[+t1.8]=TRiP.HMS01451}attP2 | Bloomington *Drosophila* Stock Center | BDSC:35037; FlyBase: FBst0035037; | |
| Genetic reagent (*D. melanogaster*) | *D. melanogaster.* Expresses dsRNA for RNAi of pnt (FBgn0003118) under UAS control in the VALIUM20 vector.: y[1] sc[*] v[1] sev[21]; P{y[+t7.7] v[+t1.8]=TRiP.HMS01452}attP2 | Bloomington *Drosophila* Stock Center | BDSC:35038; FlyBase: FBst0035038; | |
| Genetic reagent (*D. melanogaster*) | *D. melanogaster.* Expresses dsRNA for RNAi of numb (FBgn0002973) under UAS control in the VALIUM20 vector.: y[1] sc[*] v[1] sev[21]; P{y[+t7.7] v[+t1.8]=TRiP.HMS01459}attP2 | Bloomington *Drosophila* Stock Center | BDSC:35045; FlyBase: FBst0035045; | |

*Continued*

| Reagent type (species) or resource | Designation | Source or reference | Identifiers | Additional information |
|---|---|---|---|---|
| Genetic reagent (*D. melanogaster*) | *D. melanogaster.* Expresses dsRNA for RNAi of stan (FBgn0024836) under UAS control in the VALIUM20 vector.: y[1] sc[*] v[1] sev[21]; P{y[+t7.7] v[+t1.8]=TRiP.HMS01464}attP2 | Bloomington *Drosophila* Stock Center | BDSC:35050; FlyBase: FBst0035050; | |
| Genetic reagent (*D. melanogaster*) | *D. melanogaster.* Expresses dsRNA for RNAi of dco (FBgn0002413) under UAS control in the VALIUM22 vector.: y[1] sc[*] v[1] sev[21]; P{y[+t7.7] v[+t1.8]=TRiP .GL00001}attP2/TM3, Sb[1] | Bloomington *Drosophila* Stock Center | BDSC:35134; FlyBase: FBst0035134; | |
| Genetic reagent (*D. melanogaster*) | *D. melanogaster.* Expresses dsRNA for RNAi of CkIIalpha (FBgn0264492) under UAS control in the VALIUM22 vector.: y[1] sc[*] v[1] sev[21]; P{y[+t7.7] v[+t1.8]=TRiP .GL00003}attP2 | Bloomington *Drosophila* Stock Center | BDSC:35136; FlyBase: FBst0035136; | |
| Genetic reagent (*D. melanogaster*) | *D. melanogaster.* Expresses dsRNA for RNAi of gish (FBgn0250823) under UAS control in the VALIUM22 vector.: y[1] sc[*] v[1] sev[21]; P{y[+t7.7] v[+t1.8]=TRiP .GL00005}attP2 | Bloomington *Drosophila* Stock Center | BDSC:35138; FlyBase: FBst0035138; | |
| Genetic reagent (*D. melanogaster*) | *D. melanogaster.* Expresses dsRNA for RNAi of Mkk4 (FBgn0024326) under UAS control in the VALIUM22 vector.: y[1] sc[*] v[1] sev[21]; P{y[+t7.7] v[+t1.8]=TRiP .GL00010}attP2 | Bloomington *Drosophila* Stock Center | BDSC:35143; FlyBase: FBst0035143; | |
| Genetic reagent (*D. melanogaster*) | *D. melanogaster.* Expresses dsRNA for RNAi of CkIalpha (FBgn0015024) under UAS control in the VALIUM22 vector.: y[1] sc[*] v[1] sev[21]; P{y[+t7.7] v[+t1.8]=TRiP .GL00021}attP2 | Bloomington *Drosophila* Stock Center | BDSC:35153; FlyBase: FBst0035153; | |
| Genetic reagent (*D. melanogaster*) | *D. melanogaster.* Expresses dsRNA for RNAi of lic (FBgn0261524) under UAS control in the VALIUM22 vector.: y[1] sc[*] v[1] sev[21]; P{y[+t7.7] v[+t1.8]=TRiP .GL00022}attP2/TM3, Sb[1] | Bloomington *Drosophila* Stock Center | BDSC:35154; FlyBase: FBst0035154; | |
| Genetic reagent (*D. melanogaster*) | *D. melanogaster.* Expresses dsRNA for RNAi of Btk29A (FBgn0003502) under UAS control in the VALIUM22 vector.: y[1] sc[*] v[1] sev[21]; P{y[+t7.7] v[+t1.8]=TRiP .GL00027}attP2 | Bloomington *Drosophila* Stock Center | BDSC:35159; FlyBase: FBst0035159; | |
| Genetic reagent (*D. melanogaster*) | *D. melanogaster.* Expresses dsRNA for RNAi of alc (FBgn0260972) under UAS control in the VALIUM22 vector.: y[1] sc[*] v[1] sev[21]; P{y[+t7.7] v[+t1.8]=TRiP .GL00029}attP2/TM3, Sb[1] | Bloomington *Drosophila* Stock Center | BDSC:35161; FlyBase: FBst0035161; | |
| Genetic reagent (*D. melanogaster*) | *D. melanogaster.* Expresses dsRNA for RNAi of Pka-C1 (FBgn0000273) under UAS control in the VALIUM22 vector.: y[1] sc[*] v[1] sev[21]; P{y[+t7.7] v[+t1.8]=TRiP .GL00038}attP2 | Bloomington *Drosophila* Stock Center | BDSC:35169; FlyBase: FBst0035169; | |

*Continued on next page*

*Continued*

| Reagent type (species) or resource | Designation | Source or reference | Identifiers | Additional information |
|---|---|---|---|---|
| Genetic reagent (*D. melanogaster*) | *D. melanogaster.* Expresses dsRNA for RNAi of hpo (FBgn0261456) under UAS control in the VALIUM22 vector.: y[1] sc[*] v[1] sev[21]; P{y[+t7.7] v[+t1.8]=TRiP .GL00046}attP2 | Bloomington *Drosophila* Stock Center | BDSC:35176; FlyBase: FBst0035176; | |
| Genetic reagent (*D. melanogaster*) | *D. melanogaster.* Expresses dsRNA for RNAi of IKKbeta (FBgn0024222) under UAS control in the VALIUM22 vector.: y[1] sc[*] v[1] sev[21]; P{y[+t7.7] v[+t1.8]=TRiP .GL00058}attP2 | Bloomington *Drosophila* Stock Center | BDSC:35186; FlyBase: FBst0035186; | |
| Genetic reagent (*D. melanogaster*) | *D. melanogaster.* Expresses dsRNA for RNAi of Aduk (FBgn0037679) under UAS control in the VALIUM22 vector.: y[1] sc[*] v[1] sev[21]; P{y[+t7.7] v[+t1.8]=TRiP .GL00059}attP2 | Bloomington *Drosophila* Stock Center | BDSC:35187; FlyBase: FBst0035187; | |
| Genetic reagent (*D. melanogaster*) | *D. melanogaster.* Expresses dsRNA for RNAi of put (FBgn0003169) under UAS control in the VALIUM22 vector.: y[1] sc[*] v[1] sev[21]; P{y[+t7.7] v[+t1.8]=TRiP .GL00069}attP2 | Bloomington *Drosophila* Stock Center | BDSC:35195; FlyBase: FBst0035195; | |
| Genetic reagent (*D. melanogaster*) | *D. melanogaster.* Expresses dsRNA for RNAi of hep (FBgn0010303) under UAS control in the VALIUM22 vector.: y[1] sc[*] v[1] sev[21]; P{y[+t7.7] v[+t1.8]=TRiP .GL00089}attP2 | Bloomington *Drosophila* Stock Center | BDSC:35210; FlyBase: FBst0035210; | |
| Genetic reagent (*D. melanogaster*) | *D. melanogaster.* Expresses dsRNA for RNAi of brm (FBgn0000212) under UAS control in the VALIUM22 vector.: y[1] sc[*] v[1] sev[21]; P{y[+t7.7] v[+t1.8]=TRiP .GL00090}attP2 | Bloomington *Drosophila* Stock Center | BDSC:35211; FlyBase: FBst0035211; | |
| Genetic reagent (*D. melanogaster*) | *D. melanogaster.* Expresses dsRNA for RNAi of pyd (FBgn0262614) under UAS control in the VALIUM22 vector.: y[1] sc[*] v[1] sev[21]; P{y[+t7.7] v[+t1.8]=TRiP .GL00109}attP2 | Bloomington *Drosophila* Stock Center | BDSC:35225; FlyBase: FBst0035225; | |
| Genetic reagent (*D. melanogaster*) | *D. melanogaster.* Expresses dsRNA for RNAi of hyx (FBgn0037657) under UAS control in the VALIUM22 vector.: y[1] sc[*] v[1] sev[21]; P{y[+t7.7] v[+t1.8]=TRiP .GL00123}attP2 | Bloomington *Drosophila* Stock Center | BDSC:35238; FlyBase: FBst0035238; | |
| Genetic reagent (*D. melanogaster*) | *D. melanogaster.* Expresses dsRNA for RNAi of p38a (FBgn0015765) under UAS control in the VALIUM22 vector.: y[1] sc[*] v[1] sev[21]; P{y[+t7.7] v[+t1.8]=TRiP .GL00131}attP2 | Bloomington *Drosophila* Stock Center | BDSC:35244; FlyBase: FBst0035244; | |
| Genetic reagent (*D. melanogaster*) | *D. melanogaster.* Expresses dsRNA for RNAi of p38b (FBgn0024846) under UAS control in the VALIUM22 vector.: y[1] sc[*] v[1] sev[21]; P{y[+t7.7] v[+t1.8]=TRiP .GL00140}attP2 | Bloomington *Drosophila* Stock Center | BDSC:35252; FlyBase: FBst0035252; | |

*Continued on next page*

*Continued*

| Reagent type (species) or resource | Designation | Source or reference | Identifiers | Additional information |
|---|---|---|---|---|
| Genetic reagent (*D. melanogaster*) | *D. melanogaster*. Expresses dsRNA for RNAi of app (FBgn0260941) under UAS control in the VALIUM22 vector.: y[1] sc[*] v[1] sev[21]; P{y[+t7.7] v[+t1.8]=TRiP .GL00181}attP2 | Bloomington *Drosophila* Stock Center | BDSC:35280; FlyBase: FBst0035280; | |
| Genetic reagent (*D. melanogaster*) | *D. melanogaster*. Expresses dsRNA for RNAi of sdt (FBgn0261873) under UAS control in the VALIUM22 vector.: y[1] sc[*] v[1] sev[21]; P{y[+t7.7] v[+t1.8]=TRiP .GL00193}attP2 | Bloomington *Drosophila* Stock Center | BDSC:35291; FlyBase: FBst0035291; | |
| Genetic reagent (*D. melanogaster*) | *D. melanogaster*. Expresses dsRNA for RNAi of aux (FBgn0037218) under UAS control in the VALIUM22 vector.: y[1] sc[*] v[1] sev[21]; P{y[+t7.7] v[+t1.8]=TRiP .GL00213}attP2 | Bloomington *Drosophila* Stock Center | BDSC:35310; FlyBase: FBst0035310; | |
| Genetic reagent (*D. melanogaster*) | *D. melanogaster*. Expresses dsRNA for RNAi of CaMKII (FBgn0264607) under UAS control in the VALIUM22 vector.: y[1] sc[*] v[1] sev[21]; P{y[+t7.7] v[+t1.8]=TRiP .GL00237}attP2/TM3, Sb[1] | Bloomington *Drosophila* Stock Center | BDSC:35330; FlyBase: FBst0035330; | |
| Genetic reagent (*D. melanogaster*) | *D. melanogaster*. Expresses dsRNA for RNAi of Ask1 (FBgn0014006) under UAS control in the VALIUM22 vector.: y[1] sc[*] v[1] sev[21]; P{y[+t7.7] v[+t1.8]=TRiP .GL00238}attP2 | Bloomington *Drosophila* Stock Center | BDSC:35331; FlyBase: FBst0035331; | |
| Genetic reagent (*D. melanogaster*) | *D. melanogaster*. Expresses dsRNA for RNAi of SAK (FBgn0026371) under UAS control in the VALIUM22 vector.: y[1] sc[*] v[1] sev[21]; P{y[+t7.7] v[+t1.8]=TRiP .GL00244}attP2/TM3, Sb[1] | Bloomington *Drosophila* Stock Center | BDSC:35335; FlyBase: FBst0035335; | |
| Genetic reagent (*D. melanogaster*) | *D. melanogaster*. Expresses dsRNA for RNAi of sgg (FBgn0003371) under UAS control in the VALIUM22 vector.: y[1] sc[*] v[1] sev[21]; P{y[+t7.7] v[+t1.8]=TRiP .GL00277}attP2 | Bloomington *Drosophila* Stock Center | BDSC:35364; FlyBase: FBst0035364; | |
| Genetic reagent (*D. melanogaster*) | *D. melanogaster*. Expresses dsRNA for RNAi of hop (FBgn0004864) under UAS control in the VALIUM22 vector.: y[1] sc[*] v[1] sev[21]; P{y[+t7.7] v[+t1.8]=TRiP .GL00305}attP2 | Bloomington *Drosophila* Stock Center | BDSC:35386; FlyBase: FBst0035386; | |
| Genetic reagent (*D. melanogaster*) | *D. melanogaster*. Expresses dsRNA for RNAi of Mekk1 (FBgn0024329) under UAS control in the VALIUM22 vector.: y[1] sc[*] v[1] sev[21]; P{y[+t7.7] v[+t1.8]=TRiP .GL00322}attP2 | Bloomington *Drosophila* Stock Center | BDSC:35402; FlyBase: FBst0035402; | |
| Genetic reagent (*D. melanogaster*) | *D. melanogaster*. Expresses dsRNA for RNAi of aop (FBgn0000097) under UAS control in the VALIUM22 vector.: y[1] sc[*] v[1] sev[21]; P{y[+t7.7] v[+t1.8]=TRiP .GL00324}attP2 | Bloomington *Drosophila* Stock Center | BDSC:35404; FlyBase: FBst0035404; | |

*Continued*

| Reagent type (species) or resource | Designation | Source or reference | Identifiers | Additional information |
|---|---|---|---|---|
| Genetic reagent (*D. melanogaster*) | *D. melanogaster.* Expresses dsRNA for RNAi of Ras85D (FBgn0003205) under UAS control in the VALIUM22 vector.: y[1] sc[*] v[1] sev[21]; P{y[+t7.7] v[+t1.8]=TRiP .GL00336}attP2/TM3, Sb[1] | Bloomington *Drosophila* Stock Center | BDSC:35414; FlyBase: FBst0035414; | |
| Genetic reagent (*D. melanogaster*) | *D. melanogaster.* Expresses dsRNA for RNAi of Nap1 (FBgn0015268) under UAS control in the VALIUM22 vector.: y[1] sc[*] v[1] sev[21]; P{y[+t7.7] v[+t1.8]=TRiP .GL00370}attP2 | Bloomington *Drosophila* Stock Center | BDSC:35445; FlyBase: FBst0035445; | |
| Genetic reagent (*D. melanogaster*) | *D. melanogaster.* Expresses dsRNA for RNAi of osa (FBgn0261885) under UAS control in the VALIUM22 vector.: y[1] sc[*] v[1] sev[21]; P{y[+t7.7] v[+t1.8]=TRiP .GL00372}attP2 | Bloomington *Drosophila* Stock Center | BDSC:35447; FlyBase: FBst0035447; | |
| Genetic reagent (*D. melanogaster*) | *D. melanogaster.* Expresses dsRNA for RNAi of key (FBgn0041205) under UAS control in the VALIUM22 vector.: y[1] sc[*] v[1] sev[21]; P{y[+t7.7] v[+t1.8]=TRiP .GL00088}attP2 | Bloomington *Drosophila* Stock Center | BDSC:35572; FlyBase: FBst0035572; | |
| Genetic reagent (*D. melanogaster*) | *D. melanogaster.* Expresses dsRNA for RNAi of Stat92E (FBgn0016917) under UAS control in the VALIUM22 vector.: y[1] sc[*] v[1] sev[21]; P{y[+t7.7] v[+t1.8]=TRiP .GL00437}attP40/CyO | Bloomington *Drosophila* Stock Center | BDSC:35600; FlyBase: FBst0035600; | |
| Genetic reagent (*D. melanogaster*) | *D. melanogaster.* Expresses dsRNA for RNAi of Gcn5 (FBgn0020388) under UAS control in the VALIUM22 vector.: y[1] sc[*] v[1] sev[21]; P{y[+t7.7] v[+t1.8]=TRiP .GL00439}attP40 | Bloomington *Drosophila* Stock Center | BDSC:35601; FlyBase: FBst0035601; | |
| Genetic reagent (*D. melanogaster*) | *D. melanogaster.* Expresses dsRNA for RNAi of TER94 (FBgn0261014) under UAS control in the VALIUM22 vector.: y[1] sc[*] v[1] sev[21]; P{y[+t7.7] v[+t1.8]=TRiP .GL00448}attP2 | Bloomington *Drosophila* Stock Center | BDSC:35608; FlyBase: FBst0035608; | |
| Genetic reagent (*D. melanogaster*) | *D. melanogaster.* Expresses dsRNA for RNAi of wek (FBgn0001990) under UAS control in the VALIUM21 vector.: y[1] sc[*] v[1] sev[21]; P{y[+t7.7] v[+t1.8]=TRiP.GLV21045}attP2 | Bloomington *Drosophila* Stock Center | BDSC:35680; FlyBase: FBst0035680; | |
| Genetic reagent (*D. melanogaster*) | *D. melanogaster.* Expresses dsRNA for RNAi of bab2 (FBgn0025525) under UAS control in the VALIUM21 vector.: y[1] sc[*] v[1] sev[21]; P{y[+t7.7] v[+t1.8]=TRiP.GLV21085}attP2 | Bloomington *Drosophila* Stock Center | BDSC:35720; FlyBase: FBst0035720; | |
| Genetic reagent (*D. melanogaster*) | *D. melanogaster.* Expresses dsRNA for RNAi of Patj (FBgn0067864) under UAS control in the VALIUM20 vector.: y[1] sc[*] v[1] sev[21]; P{y[+t7.7] v[+t1.8]=TRiP.HMS01489}attP2 | Bloomington *Drosophila* Stock Center | BDSC:35747; FlyBase: FBst0035747; | |

*Continued*

| Reagent type (species) or resource | Designation | Source or reference | Identifiers | Additional information |
|---|---|---|---|---|
| Genetic reagent (*D. melanogaster*) | *D. melanogaster.* Expresses dsRNA for RNAi of Mtl (FBgn0039532) under UAS control in the VALIUM20 vector.: y[1] sc[*] v[1] sev[21]; P{y[+t7.7] v[+t1.8]=TRiP.HMS01500}attP2 | Bloomington *Drosophila* Stock Center | BDSC:35754; FlyBase: FBst0035754; | |
| Genetic reagent (*D. melanogaster*) | *D. melanogaster.* Expresses dsRNA for RNAi of gro (FBgn0001139) under UAS control in the VALIUM20 vector.: y[1] sc[*] v[1] sev[21]; P{y[+t7.7] v[+t1.8]=TRiP.HMS01506}attP2 | Bloomington *Drosophila* Stock Center | BDSC:35759; FlyBase: FBst0035759; | |
| Genetic reagent (*D. melanogaster*) | *D. melanogaster.* Expresses dsRNA for RNAi of Wnt6 (FBgn0031902) under UAS control in the VALIUM22 vector.: y[1] sc[*] v[1] sev[21]; P{y[+t7.7] v[+t1.8]=TRiP .GL00457}attP2 | Bloomington *Drosophila* Stock Center | BDSC:35808; FlyBase: FBst0035808; | |
| Genetic reagent (*D. melanogaster*) | *D. melanogaster.* Expresses dsRNA for RNAi of CycE (FBgn0010382) under UAS control in the VALIUM22 vector.: y[1] sc[*] v[1] sev[21]; P{y[+t7.7] v[+t1.8]=TRiP .GL00511}attP40 | Bloomington *Drosophila* Stock Center | BDSC:36092; FlyBase: FBst0036092; | |
| Genetic reagent (*D. melanogaster*) | *D. melanogaster.* Expresses dsRNA for RNAi of Myd88 (FBgn0033402) under UAS control in the VALIUM20 vector.: y[1] sc[*] v[1] sev[21]; P{y[+t7.7] v[+t1.8]=TRiP.HMS00183}attP2 | Bloomington *Drosophila* Stock Center | BDSC:36107; FlyBase: FBst0036107; | |
| Genetic reagent (*D. melanogaster*) | *D. melanogaster.* Expresses dsRNA for RNAi of Cul1 (FBgn0015509) under UAS control in the VALIUM22 vector.: y[1] sc[*] v[1] sev[21]; P{y[+t7.7] v[+t1.8]=TRiP .GL00561}attP2 | Bloomington *Drosophila* Stock Center | BDSC:36601; FlyBase: FBst0036601; | |
| Genetic reagent (*D. melanogaster*) | *D. melanogaster.* Expresses dsRNA for RNAi of chico (FBgn0024248) under UAS control in the VALIUM20 vector.: y[1] sc[*] v[1] sev[21]; P{y[+t7.7] v[+t1.8]=TRiP. HMS01553}attP2/TM3, Sb[1] | Bloomington *Drosophila* Stock Center | BDSC:36665; FlyBase: FBst0036665; | |
| Genetic reagent (*D. melanogaster*) | *D. melanogaster.* Expresses dsRNA for RNAi of Rbf2 (FBgn0038390) under UAS control in the VALIUM20 vector.: y[1] sc[*] v[1] sev[21]; P{y[+t7.7] v[+t1.8]=TRiP.HMS01586}attP2 | Bloomington *Drosophila* Stock Center | BDSC:36697; FlyBase: FBst0036697; | |
| Genetic reagent (*D. melanogaster*) | *D. melanogaster.* Expresses dsRNA for RNAi of rictor (FBgn0031006) under UAS control in the VALIUM20 vector.: y[1] sc[*] v[1] sev[21]; P{y[+t7.7] v[+t1.8]=TRiP.HMS01588}attP2 | Bloomington *Drosophila* Stock Center | BDSC:36699; FlyBase: FBst0036699; | |
| Genetic reagent (*D. melanogaster*) | *D. melanogaster.* Expresses dsRNA for RNAi of sty (FBgn0014388) under UAS control in the VALIUM20 vector.: y[1] sc[*] v[1] sev[21]; P{y[+t7.7] v[+t1.8]=TRiP.HMS01599}attP2 | Bloomington *Drosophila* Stock Center | BDSC:36709; FlyBase: FBst0036709; | |

*Continued on next page*

Continued

| Reagent type (species) or resource | Designation | Source or reference | Identifiers | Additional information |
|---|---|---|---|---|
| Genetic reagent (*D. melanogaster*) | *D. melanogaster*. Expresses dsRNA for RNAi of Zyx (FBgn0011642) under UAS control in the VALIUM20 vector.: y[1] sc[*] v[1] sev[21]; P{y[+t7.7] v[+t1.8]=TRiP.HMS01606}attP40 | Bloomington *Drosophila* Stock Center | BDSC:36716; FlyBase: FBst0036716; | |
| Genetic reagent (*D. melanogaster*) | *D. melanogaster*. Expresses dsRNA for RNAi of Wnt2 (FBgn0004360) under UAS control in the VALIUM20 vector.: y[1] sc[*] v[1] sev[21]; P{y[+t7.7] v[+t1.8]=TRiP.HMS01613}attP2 | Bloomington *Drosophila* Stock Center | BDSC:36722; FlyBase: FBst0036722; | |
| Genetic reagent (*D. melanogaster*) | *D. melanogaster*. Expresses dsRNA for RNAi of Rbf (FBgn0015799) under UAS control in the VALIUM20 vector.: y[1] sc[*] v[1] sev[21]; P{y[+t7.7] v[+t1.8]=TRiP. HMS03004}attP2/TM3, Sb[1] | Bloomington *Drosophila* Stock Center | BDSC:36744; FlyBase: FBst0036744; | |
| Genetic reagent (*D. melanogaster*) | *D. melanogaster*. Expresses dsRNA for RNAi of dlg1 (FBgn0001624) under UAS control in the VALIUM20 vector.: y[1] v[1]; P{y[+t7.7] v[+t1.8]=TRiP.7.7 v[+t1.8]=TRiP. JF02287}attP2 | Bloomington *Drosophila* Stock Center | BDSC:36771; FlyBase: FBst0036771; | |
| Genetic reagent (*D. melanogaster*) | *D. melanogaster*. Expresses dsRNA for RNAi of dpp (FBgn0000490) under UAS control in the VALIUM20 vector.: y[1] v[1]; P{y[+t7.7] v[+t1.8]=TRiP.7.7 v[+t1.8]=TRiP.JF02455}attP2 | Bloomington *Drosophila* Stock Center | BDSC:36779; FlyBase: FBst0036779; | |
| Genetic reagent (*D. melanogaster*) | *D. melanogaster*. Expresses dsRNA for RNAi of Dl (FBgn0000463) under UAS control in the VALIUM22 vector.: y[1] sc[*] v[1] sev[21]; P{y[+t7.7] v[+t1.8]=TRiP .GL00520}attP40 | Bloomington *Drosophila* Stock Center | BDSC:36784; FlyBase: FBst0036784; | |
| Genetic reagent (*D. melanogaster*) | *D. melanogaster*. Expresses dsRNA for RNAi of HDAC1 (FBgn0015805) under UAS control in the VALIUM22 vector.: y[1] sc[*] v[1] sev[21]; P{y[+t7.7] v[+t1.8]=TRiP .GL01005}attP40 | Bloomington *Drosophila* Stock Center | BDSC:36800; FlyBase: FBst0036800; | |
| Genetic reagent (*D. melanogaster*) | *D. melanogaster*. Expresses dsRNA for RNAi of Su(dx) (FBgn0003557) under UAS control in the VALIUM22 vector.: y[1] sc[*] v[1] sev[21]; P{y[+t7.7] v[+t1.8]=TRiP .GL01077}attP2 | Bloomington *Drosophila* Stock Center | BDSC:36836; FlyBase: FBst0036836; | |
| Genetic reagent (*D. melanogaster*) | *D. melanogaster*. Expresses dsRNA for RNAi of pbl (FBgn0003041) under UAS control in the VALIUM22 vector.: y[1] sc[*] v[1] sev[21]; P{y[+t7.7] v[+t1.8]=TRiP .GL01092}attP2 | Bloomington *Drosophila* Stock Center | BDSC:36841; FlyBase: FBst0036841; | |
| Genetic reagent (*D. melanogaster*) | *D. melanogaster*. Expresses dsRNA for RNAi of Sod2 (FBgn0010213) under UAS control in the VALIUM22 vector.: y[1] sc[*] v[1] sev[21]; P{y[+t7.7] v[+t1.8]=TRiP .GL01015}attP40 | Bloomington *Drosophila* Stock Center | BDSC:36871; FlyBase: FBst0036871; | |

*Continued*

| Reagent type (species) or resource | Designation | Source or reference | Identifiers | Additional information |
|---|---|---|---|---|
| Genetic reagent (*D. melanogaster*) | *D. melanogaster.* Expresses dsRNA for RNAi of Art1 (FBgn0037834) under UAS control in the VALIUM22 vector.: y[1] sc[*] v[1] sev[21]; P{y[+t7.7] v[+t1.8]=TRiP .GL01072}attP2/TM3, Sb[1] | Bloomington *Drosophila* Stock Center | BDSC:36891; FlyBase: FBst0036891; | |
| Genetic reagent (*D. melanogaster*) | *D. melanogaster.* Expresses dsRNA for RNAi of CG10924 (FBgn0034356) under UAS control in the VALIUM20 vector.: y[1] sc[*] v[1] sev[21]; P{y[+t7.7] v[+t1.8]=TRiP.HMS00200}attP2 | Bloomington *Drosophila* Stock Center | BDSC:36915; FlyBase: FBst0036915; | |
| Genetic reagent (*D. melanogaster*) | *D. melanogaster.* Expresses dsRNA for RNAi of trr (FBgn0023518) under UAS control in the VALIUM20 vector.: y[1] sc[*] v[1] sev[21]; P{y[+t7.7] v[+t1.8]=TRiP.HMS01019}attP2 | Bloomington *Drosophila* Stock Center | BDSC:36916; FlyBase: FBst0036916; | |
| Genetic reagent (*D. melanogaster*) | *D. melanogaster.* Expresses dsRNA for RNAi of Cdc42 (FBgn0010341) under UAS control in the VALIUM22 vector.: y[1] sc[*] v[1] sev[21]; P{y[+t7.7] v[+t1.8]=TRiP .GL00620}attP40 | Bloomington *Drosophila* Stock Center | BDSC:37477; FlyBase: FBst0037477; | |
| Genetic reagent (*D. melanogaster*) | *D. melanogaster.* Expresses dsRNA for RNAi of nej (FBgn0261617) under UAS control in the VALIUM20 vector.: y[1] sc[*] v[1] sev[21]; P{y[+t7.7] v[+t1.8]=TRiP.HMS01507}attP2 | Bloomington *Drosophila* Stock Center | BDSC:37489; FlyBase: FBst0037489; | |
| Genetic reagent (*D. melanogaster*) | *D. melanogaster.* Expresses dsRNA for RNAi of Pvr (FBgn0032006) under UAS control in the VALIUM20 vector.: y[1] sc[*] v[1] sev[21]; P{y[+t7.7] v[+t1.8]=TRiP.HMS01662}attP40 | Bloomington *Drosophila* Stock Center | BDSC:37520; FlyBase: FBst0037520; | |
| Genetic reagent (*D. melanogaster*) | *D. melanogaster.* Expresses dsRNA for RNAi of cno (FBgn0259212) under UAS control in the VALIUM22 vector.: y[1] sc[*] v[1] sev[21]; P{y[+t7.7] v[+t1.8]=TRiP .GL00633}attP40 | Bloomington *Drosophila* Stock Center | BDSC:38194; FlyBase: FBst0038194; | |
| Genetic reagent (*D. melanogaster*) | *D. melanogaster.* Expresses dsRNA for RNAi of pygo (FBgn0043900) under UAS c ontrol in the VALIUM22 vector.: y[1] sc[*] v[1] sev[21]; P{y[+t7.7] v[+t1.8]=TRiP .GL00647}attP40 | Bloomington *Drosophila* Stock Center | BDSC:38208; FlyBase: FBst0038208; | |
| Genetic reagent (*D. melanogaster*) | *D. melanogaster.* Expresses dsRNA for RNAi of ed (FBgn0000547) under UAS control in the VALIUM22 vector.: y[1] sc[*] v[1] sev[21]; P{y[+t7.7] v[+t1.8]=TRiP .GL00648}attP40 | Bloomington *Drosophila* Stock Center | BDSC:38209; FlyBase: FBst0038209; | |
| Genetic reagent (*D. melanogaster*) | *D. melanogaster.* Expresses dsRNA for RNAi of Nrg (FBgn0264975) under UAS control in the VALIUM22 vector.: y[1] sc[*] v[1] sev[21]; P{y[+t7.7] v[+t1.8]=TRiP .GL00656}attP40 | Bloomington *Drosophila* Stock Center | BDSC:38215; FlyBase: FBst0038215; | |

*Continued on next page*

*Continued*

| Reagent type (species) or resource | Designation | Source or reference | Identifiers | Additional information |
|---|---|---|---|---|
| Genetic reagent (*D. melanogaster*) | *D. melanogaster.* Expresses dsRNA for RNAi of aph-1 (FBgn0031458) under UAS control in the VALIUM20 vector.: y[1] sc[*] v[1] sev[21]; P{y[+t7.7] v[+t1.8]=TRiP.HMS01693}attP40 | Bloomington *Drosophila* Stock Center | BDSC:38249; FlyBase: FBst0038249; | |
| Genetic reagent (*D. melanogaster*) | *D. melanogaster.* Expresses dsRNA for RNAi of cbt (FBgn0043364) under UAS control in the VALIUM20 vector.: y[1] sc[*] v[1] sev[21]; P{y[+t7.7] v[+t1.8]=TRiP.HMS01726}attP40 | Bloomington *Drosophila* Stock Center | BDSC:38276; FlyBase: FBst0038276; | |
| Genetic reagent (*D. melanogaster*) | *D. melanogaster.* Expresses dsRNA for RNAi of Vps36 (FBgn0086785) under UAS control in the VALIUM20 vector.: y[1] sc[*] v[1] sev[21]; P{y[+t7.7] v[+t1.8]=TRiP. HMS01739}attP40 | Bloomington *Drosophila* Stock Center | BDSC:38286; FlyBase: FBst0038286; | |
| Genetic reagent (*D. melanogaster*) | *D. melanogaster.* Expresses dsRNA for RNAi of crb (FBgn0259685) under UAS control in the VALIUM20 vector.: y[1] v[1]; P{y[+t7.7] v[+t1.8]=TRiP.7.7 v[+t1.8]=TRiP. HMS01842}attP40 | Bloomington *Drosophila* Stock Center | BDSC:38373; FlyBase: FBst0038373; | |
| Genetic reagent (*D. melanogaster*) | *D. melanogaster.* Expresses dsRNA for RNAi of tws (FBgn0004889) under UAS control in the VALIUM22 vector.: y[1] v[1]; P{y[+t7.7] v[+t1.8]=TRiP .GL00670}attP40/CyO | Bloomington *Drosophila* Stock Center | BDSC:38899; FlyBase: FBst0038899; | |
| Genetic reagent (*D. melanogaster*) | *D. melanogaster.* Expresses dsRNA for RNAi of S (FBgn0003310) under UAS control in the VALIUM22 vector.: y[1] sc[*] v[1] sev[21]; P{y[+t7.7] v[+t1.8]=TRiP .GL00686}attP2 | Bloomington *Drosophila* Stock Center | BDSC:38914; FlyBase: FBst0038914; | |
| Genetic reagent (*D. melanogaster*) | *D. melanogaster.* Expresses dsRNA for RNAi of Pvf3 (FBgn0085407) under UAS control in the VALIUM20 vector.: y[1] v[1]; P{y[+t7.7] v[+t1.8]=TRiP.7.7 v[+t1.8]=TRiP. HMS01876}attP40/CyO | Bloomington *Drosophila* Stock Center | BDSC:38962; FlyBase: FBst0038962; | |
| Genetic reagent (*D. melanogaster*) | *D. melanogaster.* Expresses dsRNA for RNAi of CanB2 (FBgn0015614) under UAS control in the VALIUM20 vector.: y[1] sc[*] v[1] sev[21]; P{y[+t7.7] v[+t1.8]=TRiP.HMS01886}attP2 | Bloomington *Drosophila* Stock Center | BDSC:38971; FlyBase: FBst0038971; | |
| Genetic reagent (*D. melanogaster*) | *D. melanogaster.* Expresses dsRNA for RNAi of Pi3K21B (FBgn0020622) under UAS control in the VALIUM20 vector.: y[1] sc[*] v[1] sev[21]; P{y[+t7.7] v[+t1.8]=TRiP.HMS01907}attP40 | Bloomington *Drosophila* Stock Center | BDSC:38991; FlyBase: FBst0038991; | |
| Genetic reagent (*D. melanogaster*) | *D. melanogaster.* Expresses dsRNA for RNAi of par-6 (FBgn0026192) under UAS control in the VALIUM20 vector.: y[1] sc[*] v[1] sev[21]; P{y[+t7.7] v[+t1.8]=TRiP.HMS01928}attP40 | Bloomington *Drosophila* Stock Center | BDSC:39010; FlyBase: FBst0039010; | |

*Continued*

| Reagent type (species) or resource | Designation | Source or reference | Identifiers | Additional information |
|---|---|---|---|---|
| Genetic reagent (*D. melanogaster*) | *D. melanogaster.* Expresses dsRNA for RNAi of Pvf1 (FBgn0030964) under UAS control in the VALIUM20 vector.: y[1] sc[*] v[1] sev[21]; P{y[+t7.7] v[+t1.8]=TRiP.HMS01958}attP40 | Bloomington *Drosophila* Stock Center | BDSC:39038; FlyBase: FBst0039038; | |
| Genetic reagent (*D. melanogaster*) | *D. melanogaster.* Expresses dsRNA for RNAi of Pka-C3 (FBgn0000489) under UAS control in the VALIUM20 vector.: y[1] sc[*] v[1] sev[21]; P{y[+t7.7] v[+t1.8]=TRiP.HMS01970}attP2 | Bloomington *Drosophila* Stock Center | BDSC:39050; FlyBase: FBst0039050; | |
| Genetic reagent (*D. melanogaster*) | *D. melanogaster.* Expresses dsRNA for RNAi of DAAM (FBgn0025641) under UAS control in the VALIUM20 vector.: y[1] sc[*] v[1] sev[21]; P{y[+t7.7] v[+t1.8]=TRiP.HMS01978}attP2 | Bloomington *Drosophila* Stock Center | BDSC:39058; FlyBase: FBst0039058; | |
| Genetic reagent (*D. melanogaster*) | *D. melanogaster.* Expresses dsRNA for RNAi of scrib (FBgn0263289) under UAS control in the VALIUM20 vector.: y[1] sc[*] v[1] sev[21]; P{y[+t7.7] v[+t1.8]=TRiP. HMS01993}attP40/CyO | Bloomington *Drosophila* Stock Center | BDSC:39073; FlyBase: FBst0039073; | |
| Genetic reagent (*D. melanogaster*) | *D. melanogaster.* Expresses dsRNA for RNAi of sina (FBgn0003410) under UAS control in the VALIUM20 vector.: y[1] v[1]; P{y[+t7.7] v[+t1.8]=TRiP.7.7 v[+t1.8]=TRiP. HMS02008}attP40 | Bloomington *Drosophila* Stock Center | BDSC:40842; FlyBase: FBst0040842; | |
| Genetic reagent (*D. melanogaster*) | *D. melanogaster.* Expresses dsRNA for RNAi of btl (FBgn0285896) under UAS control in the VALIUM20 vector.: y[1] sc[*] v[1] sev[21]; P{y[+t7.7] v[+t1.8]=TRiP.HMS02038}attP2 | Bloomington *Drosophila* Stock Center | BDSC:40871; FlyBase: FBst0040871; | |
| Genetic reagent (*D. melanogaster*) | *D. melanogaster.* Expresses dsRNA for RNAi of Pp2B-14D (FBgn0011826) under UAS control in the VALIUM20 vector.: y[1] sc[*] v[1] sev[21]; P{y[+t7.7] v[+t1.8]=TRiP.HMS02039}attP2 | Bloomington *Drosophila* Stock Center | BDSC:40872; FlyBase: FBst0040872; | |
| Genetic reagent (*D. melanogaster*) | *D. melanogaster.* Expresses dsRNA for RNAi of Smurf (FBgn0029006) under UAS control in the VALIUM20 vector.: y[1] sc[*] v[1] sev[21]; P{y[+t7.7] v[+t1.8]=TRiP. HMS02153}attP40 | Bloomington *Drosophila* Stock Center | BDSC:40905; FlyBase: FBst0040905; | |
| Genetic reagent (*D. melanogaster*) | *D. melanogaster.* Expresses dsRNA for RNAi of CycB (FBgn0000405) under UAS control in the VALIUM20 vector.: y[1] sc[*] v[1] sev[21]; P{y[+t7.7] v[+t1.8]=TRiP. HMS02163}attP2/TM3, Sb[1] | Bloomington *Drosophila* Stock Center | BDSC:40915; FlyBase: FBst0040915; | |

*Continued*

| Reagent type (species) or resource | Designation | Source or reference | Identifiers | Additional information |
|---|---|---|---|---|
| Genetic reagent (*D. melanogaster*) | *D. melanogaster*. Expresses dsRNA for RNAi of fu (FBgn0001079) under UAS control in the VALIUM22 vector.: y[1] v[1]; P{y[+t7.7] v[+t1.8]=TRiP .GL00705}attP40 | Bloomington *Drosophila* Stock Center | BDSC:41588; FlyBase: FBst0041588; | |
| Genetic reagent (*D. melanogaster*) | *D. melanogaster*. Expresses dsRNA for RNAi of ru (FBgn0003295) under UAS control in the VALIUM22 vector.: y[1] sc[*] v[1] sev[21]; P{y[+t7.7] v[+t1.8]=TRiP .GL01129}attP2 | Bloomington *Drosophila* Stock Center | BDSC:41593; FlyBase: FBst0041593; | |
| Genetic reagent (*D. melanogaster*) | *D. melanogaster*. Expresses dsRNA for RNAi of ksr (FBgn0015402) under UAS control in the VALIUM22 vector.: y[1] v[1]; P{y[+t7.7] v[+t1.8]=TRiP .GL01134}attP2 | Bloomington *Drosophila* Stock Center | BDSC:41598; FlyBase: FBst0041598; | |
| Genetic reagent (*D. melanogaster*) | *D. melanogaster*. Expresses dsRNA for RNAi of slpr (FBgn0030018) under UAS control in the VALIUM22 vector.: y[1] v[1]; P{y[+t7.7] v[+t1.8]=TRiP .GL01187}attP2 | Bloomington *Drosophila* Stock Center | BDSC:41605; FlyBase: FBst0041605; | |
| Genetic reagent (*D. melanogaster*) | *D. melanogaster*. Expresses dsRNA for RNAi of p53 (FBgn0039044) under UAS control in the VALIUM22 vector.: y[1] v[1]; P{y[+t7.7] v[+t1.8]=TRiP .GL01220}attP40 | Bloomington *Drosophila* Stock Center | BDSC:41638; FlyBase: FBst0041638; | |
| Genetic reagent (*D. melanogaster*) | *D. melanogaster*. Expresses dsRNA for RNAi of Smox (FBgn0025800) under UAS control in the VALIUM20 vector.: y[1] sc[*] v[1] sev[21]; P{y[+t7.7] v[+t1.8]=TRiP.HMS02203}attP40 | Bloomington *Drosophila* Stock Center | BDSC:41670; FlyBase: FBst0041670; | |
| Genetic reagent (*D. melanogaster*) | *D. melanogaster*. Expresses dsRNA for RNAi of rho (FBgn0004635) under UAS control in the VALIUM20 vector.: y[1] v[1]; P{y[+t7.7] v[+t1.8]=TRiP.7.7 v[+t1.8]=TRiP. HMS02264}attP40 | Bloomington *Drosophila* Stock Center | BDSC:41699; FlyBase: FBst0041699; | |
| Genetic reagent (*D. melanogaster*) | *D. melanogaster*. Expresses dsRNA for RNAi of S6k (FBgn0283472) under UAS control in the VALIUM20 vector.: y[1] v[1]; P{y[+t7.7] v[+t1.8]=TRiP.7.7 v[+t1.8]=TRiP. HMS02267}attP2 | Bloomington *Drosophila* Stock Center | BDSC:41702; FlyBase: FBst0041702; | |
| Genetic reagent (*D. melanogaster*) | *D. melanogaster*. Expresses dsRNA for RNAi of snk (FBgn0003450) under UAS control in the VALIUM20 vector.: y[1] sc[*] v[1] sev[21]; P{y[+t7.7] v[+t1.8]=TRiP.HMS02289}attP2 | Bloomington *Drosophila* Stock Center | BDSC:41723; FlyBase: FBst0041723; | |
| Genetic reagent (*D. melanogaster*) | *D. melanogaster*. Expresses dsRNA for RNAi of Ufd1-like (FBgn0036136) under UAS control in the VALIUM22 vector.: y[1] v[1]; P{y[+t7.7] v[+t1.8]=TRiP .GL01251}attP2 | Bloomington *Drosophila* Stock Center | BDSC:41823; FlyBase: FBst0041823; | |

*Continued on next page*

Continued

| Reagent type (species) or resource | Designation | Source or reference | Identifiers | Additional information |
|---|---|---|---|---|
| Genetic reagent (*D. melanogaster*) | *D. melanogaster.* Expresses dsRNA for RNAi of RasGAP1 (FBgn0004390) under UAS control in the VALIUM22 vector.: y[1] v[1]; P{y[+t7.7] v[+t1.8]=TRiP .GL01258}attP2 | Bloomington *Drosophila* Stock Center | BDSC:41830; FlyBase: FBst0041830; | |
| Genetic reagent (*D. melanogaster*) | *D. melanogaster.* Expresses dsRNA for RNAi of 14-3-3zeta (FBgn0004907) under UAS control in the VALIUM22 vector.: y[1] sc[*] v[1] sev[21]; P{y[+t7.7] v[+t1.8]=TRiP .GL01310}attP40 | Bloomington *Drosophila* Stock Center | BDSC:41878; FlyBase: FBst0041878; | |
| Genetic reagent (*D. melanogaster*) | *D. melanogaster.* Expresses dsRNA for RNAi of tkv (FBgn0003716) under UAS control in the VALIUM22 vector.: y[1] sc[*] v[1] sev[21]; P{y[+t7.7] v[+t1.8]=TRiP .GL01338}attP2 | Bloomington *Drosophila* Stock Center | BDSC:41904; FlyBase: FBst0041904; | |
| Genetic reagent (*D. melanogaster*) | *D. melanogaster.* Expresses dsRNA for RNAi of homer (FBgn0025777) under UAS control in the VALIUM20 vector.: y[1] sc[*] v[1] sev[21]; P{y[+t7.7] v[+t1.8]=TRiP.HMS02301}attP2 | Bloomington *Drosophila* Stock Center | BDSC:41908; FlyBase: FBst0041908; | |
| Genetic reagent (*D. melanogaster*) | *D. melanogaster.* Expresses dsRNA for RNAi of pll (FBgn0010441) under UAS control in the VALIUM20 vector.: y[1] v[1]; P{y[+t7.7] v[+t1.8]=TRiP.7.7 v[+t1.8]=TRiP. HMS02332}attP40 | Bloomington *Drosophila* Stock Center | BDSC:41935; FlyBase: FBst0041935; | |
| Genetic reagent (*D. melanogaster*) | *D. melanogaster.* Expresses dsRNA for RNAi of elB (FBgn0004858) under UAS control in the VALIUM20 vector.: y[1] sc[*] v[1] sev[21]; P{y[+t7.7] v[+t1.8]=TRiP.HMS02357}attP2 | Bloomington *Drosophila* Stock Center | BDSC:41960; FlyBase: FBst0041960; | |
| Genetic reagent (*D. melanogaster*) | *D. melanogaster.* Expresses dsRNA for RNAi of ea (FBgn0000533) under UAS control in the VALIUM20 vector.: y[1] sc[*] v[1] sev[21]; P{y[+t7.7] v[+t1.8]=TRiP.HMS02358}attP2 | Bloomington *Drosophila* Stock Center | BDSC:41961; FlyBase: FBst0041961; | |
| Genetic reagent (*D. melanogaster*) | *D. melanogaster.* Expresses dsRNA for RNAi of CycB3 (FBgn0015625) under UAS control in the VALIUM20 vector.: y[1] sc[*] v[1] sev[21]; P{y[+t7.7] v[+t1.8]=TRiP.HMS02377}attP2 | Bloomington *Drosophila* Stock Center | BDSC:41979; FlyBase: FBst0041979; | |
| Genetic reagent (*D. melanogaster*) | *D. melanogaster.* Expresses dsRNA for RNAi of lgs (FBgn0039907) under UAS control in the VALIUM20 vector.: y[1] sc[*] v[1] sev[21]; P{y[+t7.7] v[+t1.8]=TRiP.HMS02381}attP2 | Bloomington *Drosophila* Stock Center | BDSC:41983; FlyBase: FBst0041983; | |
| Genetic reagent (*D. melanogaster*) | *D. melanogaster.* Expresses dsRNA for RNAi of Actbeta (FBgn0024913) under UAS control in the VALIUM20 vector.: y[1] v[1]; P{y[+t7.7] v[+t1.8]=TRiP.7.7 v[+t1.8]=TRiP. HMJ02057}attP40 | Bloomington *Drosophila* Stock Center | BDSC:42493; FlyBase: FBst0042493; | |

*Continued*

| Reagent type (species) or resource | Designation | Source or reference | Identifiers | Additional information |
|---|---|---|---|---|
| Genetic reagent (*D. melanogaster*) | *D. melanogaster.* Expresses dsRNA for RNAi of CG5059 (FBgn0037007) under UAS control in the VALIUM20 vector.: y[1] v[1]; P{y[+t7.7] v[+t1.8]=TRiP. HMJ02058}attP40 | Bloomington *Drosophila* Stock Center | BDSC:42494; FlyBase: FBst0042494; | |
| Genetic reagent (*D. melanogaster*) | *D. melanogaster.* Expresses dsRNA for RNAi of msn (FBgn0010909) under UAS control in the VALIUM20 vector.: y[1] v[1]; P{y[+t7.7] v[+t1.8]=TRiP.7.7 v[+t1.8]=TRiP. HMJ02084}attP40 | Bloomington *Drosophila* Stock Center | BDSC:42518; FlyBase: FBst0042518; | |
| Genetic reagent (*D. melanogaster*) | *D. melanogaster.* Expresses dsRNA for RNAi of Rassf (FBgn0039055) under UAS control in the VALIUM20 vector.: y[1] v[1]; P{y[+t7.7] v[+t1.8]=TRiP.7.7 v[+t1.8]=TRiP. HMJ02102}attP40 | Bloomington *Drosophila* Stock Center | BDSC:42534; FlyBase: FBst0042534; | |
| Genetic reagent (*D. melanogaster*) | *D. melanogaster.* Expresses dsRNA for RNAi of sax (FBgn0003317) under UAS control in the VALIUM20 vector.: y[1] v[1]; P{y[+t7.7] v[+t1.8]=TRiP.7.7 v[+t1.8]=TRiP. HMJ02118}attP40/CyO | Bloomington *Drosophila* Stock Center | BDSC:42546; FlyBase: FBst0042546; | |
| Genetic reagent (*D. melanogaster*) | *D. melanogaster.* Expresses dsRNA for RNAi of et (FBgn0031055) under UAS control in the VALIUM20 vector.: y[1] v[1]; P{y[+t7.7] v[+t1.8]=TRiP.7.7 v[+t1.8]=TRiP. HMJ02213}attP40 | Bloomington *Drosophila* Stock Center | BDSC:42557; FlyBase: FBst0042557; | |
| Genetic reagent (*D. melanogaster*) | *D. melanogaster.* Expresses dsRNA for RNAi of nmo (FBgn0011817) under UAS control in the VALIUM20 vector.: y[1] v[1]; P{y[+t7.7] v[+t1.8]=TRiP.7.7 v[+t1.8]=TRiP. HMJ02229}attP40 | Bloomington *Drosophila* Stock Center | BDSC:42570; FlyBase: FBst0042570; | |
| Genetic reagent (*D. melanogaster*) | *D. melanogaster.* Expresses dsRNA for RNAi of CG12147 (FBgn0037325) under UAS control in the VALIUM20 vector.: y[1] sc[*] v[1] sev[21]; P{y[+t7.7] v[+t1.8]=TRiP.HMS02447}attP2 | Bloomington *Drosophila* Stock Center | BDSC:42612; FlyBase: FBst0042612; | |
| Genetic reagent (*D. melanogaster*) | *D. melanogaster.* Expresses dsRNA for RNAi of Act5C (FBgn0000042) under UAS control in the VALIUM20 vector.: y[1] sc[*] v[1] sev[21]; P{y[+t7.7] v[+t1.8]=TRiP. HMS02487}attP2 | Bloomington *Drosophila* Stock Center | BDSC:42651; FlyBase: FBst0042651; | |
| Genetic reagent (*D. melanogaster*) | *D. melanogaster.* Expresses dsRNA for RNAi of Act87E (FBgn0000046) under UAS control in the VALIUM20 vector.: y[1] sc[*] v[1] sev[21]; P{y[+t7.7] v[+t1.8]=TRiP. HMS02488}attP2/TM3, Sb[1] | Bloomington *Drosophila* Stock Center | BDSC:42652; FlyBase: FBst0042652; | |

*Continued on next page*

*Continued*

| Reagent type (species) or resource | Designation | Source or reference | Identifiers | Additional information |
|---|---|---|---|---|
| Genetic reagent (*D. melanogaster*) | *D. melanogaster.* Expresses dsRNA for RNAi of emc (FBgn0000575) under UAS control in the VALIUM22 vector.: y[1] v[1]; P{y[+t7.7] v[+t1.8]=TRiP .GL00724}attP2 | Bloomington *Drosophila* Stock Center | BDSC:42768; FlyBase: FBst0042768; | |
| Genetic reagent (*D. melanogaster*) | *D. melanogaster.* Expresses dsRNA for RNAi of the Stellate gene family (FBgn0003523) plus Ste12DOR and SteXh:CG42398 (FBgn0044817 and FBgn0259817) under UAS control in the VALIUM22 vector.: y[1] v[1]; P{y[+t7.7] v[+t1.8]=TRiP .GL01156}attP2 | Bloomington *Drosophila* Stock Center | BDSC:42786; FlyBase: FBst0042786; | |
| Genetic reagent (*D. melanogaster*) | *D. melanogaster.* Expresses dsRNA for RNAi of Socs44A (FBgn0033266) under UAS control in the VALIUM20 vector.: y[1] sc[*] v[1] sev[21]; P{y[+t7.7] v[+t1.8]=TRiP.HMS02515}attP2 | Bloomington *Drosophila* Stock Center | BDSC:42830; FlyBase: FBst0042830; | |
| Genetic reagent (*D. melanogaster*) | *D. melanogaster.* Expresses dsRNA for RNAi of gcm (FBgn0014179) under UAS control in the VALIUM20 vector.: y[1] sc[*] v[1] sev[21]; P{y[+t7.7] v[+t1.8]=TRiP. HMS02582}attP40 | Bloomington *Drosophila* Stock Center | BDSC:42889; FlyBase: FBst0042889; | |
| Genetic reagent (*D. melanogaster*) | *D. melanogaster.* Expresses dsRNA for RNAi of CkIIbeta (FBgn0000259) under UAS control in the VALIUM20 vector.: y[1] sc[*] v[1] sev[21]; P{y[+t7.7] v[+t1.8]=TRiP. HMS02636}attP40 | Bloomington *Drosophila* Stock Center | BDSC:42943; FlyBase: FBst0042943; | |
| Genetic reagent (*D. melanogaster*) | *D. melanogaster.* Expresses dsRNA for RNAi of smo (FBgn0003444) under UAS control in the VALIUM22 vector.: y[1] v[1]; P{y[+t7.7] v[+t1.8]=TRiP .GL01472}attP2 | Bloomington *Drosophila* Stock Center | BDSC:43134; FlyBase: FBst0043134; | |
| Genetic reagent (*D. melanogaster*) | *D. melanogaster.* Expresses dsRNA for RNAi of dock (FBgn0010583) under UAS control in the VALIUM22 vector.: y[1] v[1]; P{y[+t7.7] v[+t1.8]=TRiP .GL01519}attP2/TM3, Sb[1] | Bloomington *Drosophila* Stock Center | BDSC:43176; FlyBase: FBst0043176; | |
| Genetic reagent (*D. melanogaster*) | *D. melanogaster.* Expresses dsRNA for RNAi of Mad (FBgn0011648) under UAS control in the VALIUM22 vector.: y[1] sc[*] v[1] sev[21]; P{y[+t7.7] v[+t1.8]=TRiP .GL01527}attP40 | Bloomington *Drosophila* Stock Center | BDSC:43183; FlyBase: FBst0043183; | |
| Genetic reagent (*D. melanogaster*) | *D. melanogaster.* Expresses dsRNA for RNAi of Pp2A-29B (FBgn0260439) under UAS control in the VALIUM20 vector.: y[1] sc[*] v[1] sev[21]; P{y[+t7.7] v[+t1.8]=TRiP. HMS01921}attP2/TM3, Sb[1] | Bloomington *Drosophila* Stock Center | BDSC:43283; FlyBase: FBst0043283; | |

*Continued on next page*

*Continued*

| Reagent type (species) or resource | Designation | Source or reference | Identifiers | Additional information |
|---|---|---|---|---|
| Genetic reagent (*D. melanogaster*) | *D. melanogaster.* Expresses dsRNA for RNAi of CG11658 (FBgn0036196) under UAS control in the VALIUM20 vector.: y[1] sc[*] v[1] sev[21]; P{y[+t7.7] v[+t1.8]=TRiP. HMS02671}attP40 | Bloomington *Drosophila* Stock Center | BDSC:43298; FlyBase: FBst0043298; | |
| Genetic reagent (*D. melanogaster*) | *D. melanogaster.* Expresses dsRNA for RNAi of Myc (FBgn0262656) under UAS control in the VALIUM22 vector.: y[1] sc[*] v[1] sev[21]; P{y[+t7.7] v[+t1.8]=TRiP .GL01314}attP40 | Bloomington *Drosophila* Stock Center | BDSC:43962; FlyBase: FBst0043962; | |
| Genetic reagent (*D. melanogaster*) | *D. melanogaster.* Expresses dsRNA for RNAi of pav (FBgn0011692) under UAS control in the VALIUM22 vector.: y[1] sc[*] v[1] sev[21]; P{y[+t7.7] v[+t1.8]=TRiP .GL01316}attP40 | Bloomington *Drosophila* Stock Center | BDSC:43963; FlyBase: FBst0043963; | |
| Genetic reagent (*D. melanogaster*) | *D. melanogaster.* Expresses dsRNA for RNAi of eIF4EHP (FBgn0053100) under UAS control in the VALIUM20 vector.: y[1] sc[*] v[1] sev[21]; P{y[+t7.7] v[+t1.8]=TRiP.HMS02703}attP40 | Bloomington *Drosophila* Stock Center | BDSC:43990; FlyBase: FBst0043990; | |
| Genetic reagent (*D. melanogaster*) | *D. melanogaster.* Expresses dsRNA for RNAi of Atg1 (FBgn0260945) under UAS control in the VALIUM20 vector.: y[1] v[1]; P{y[+t7.7] v[+t1.8]=TRiP.7.7 v[+t1.8]=TRiP. HMS02750}attP40 | Bloomington *Drosophila* Stock Center | BDSC:44034; FlyBase: FBst0044034; | |
| Genetic reagent (*D. melanogaster*) | *D. melanogaster.* Expresses dsRNA for RNAi of tefu (FBgn0045035) under UAS control in the VALIUM20 vector.: y[1] sc[*] v[1] sev[21]; P{y[+t7.7] v[+t1.8]=TRiP. HMS02790}attP40 | Bloomington *Drosophila* Stock Center | BDSC:44073; FlyBase: FBst0044073; | |
| Genetic reagent (*D. melanogaster*) | *D. melanogaster.* Expresses dsRNA for RNAi of bi (FBgn0000179) under UAS control in the VALIUM20 vector.: y[1] sc[*] v[1] sev[21]; P{y[+t7.7] v[+t1.8]=TRiP. HMS02815}attP2 | Bloomington *Drosophila* Stock Center | BDSC:44095; FlyBase: FBst0044095; | |
| Genetic reagent (*D. melanogaster*) | *D. melanogaster.* Expresses dsRNA for RNAi of mad2 (FBgn0035640) under UAS control in the VALIUM22 vector.: y[1] sc[*] v[1] sev[21]; P{y[+t7.7] v[+t1.8]=TRiP.GLC01381}attP2 | Bloomington *Drosophila* Stock Center | BDSC:44430; FlyBase: FBst0044430; | |
| Genetic reagent (*D. melanogaster*) | *D. melanogaster.* Expresses dsRNA for RNAi of ebi (FBgn0263933) under UAS control in the VALIUM22 vector.: y[1] v[1]; P{y[+t7.7] v[+t1.8]=TRiP. GLC01413}attP40 | Bloomington *Drosophila* Stock Center | BDSC:44443; FlyBase: FBst0044443; | |

*Continued on next page*

*Continued*

| Reagent type (species) or resource | Designation | Source or reference | Identifiers | Additional information |
|---|---|---|---|---|
| Genetic reagent (*D. melanogaster*) | *D. melanogaster.* Expresses dsRNA for RNAi of dx (FBgn0000524) under UAS control in the VALIUM22 vector.: y[1] sc[*] v[1] sev[21]; P{y[+t7.7] v[+t1.8]=TRiP.GLC01607}attP2 | Bloomington *Drosophila* Stock Center | BDSC:44455; FlyBase: FBst0044455; | |
| Genetic reagent (*D. melanogaster*) | *D. melanogaster.* Expresses dsRNA for RNAi of fz3 (FBgn0027343) under UAS control in the VALIUM22 vector.: y[1] sc[*] v[1] sev[21]; P{y[+t7.7] v[+t1.8]=TRiP. GLC01626}attP2 | Bloomington *Drosophila* Stock Center | BDSC:44468; FlyBase: FBst0044468; | |
| Genetic reagent (*D. melanogaster*) | *D. melanogaster.* Expresses dsRNA for RNAi of cos (FBgn0000352) under UAS control in the VALIUM20 vector.: y[1] v[1]; P{y[+t7.7] v[+t1.8]=TRiP.7.7 v[+t1.8]=TRiP. HMC02347}attP2 | Bloomington *Drosophila* Stock Center | BDSC:44472; FlyBase: FBst0044472; | |
| Genetic reagent (*D. melanogaster*) | *D. melanogaster.* Expresses dsRNA for RNAi of disp (FBgn0029088) under UAS control in the VALIUM20 vector.: y[1] sc[*] v[1] sev[21]; P{y[+t7.7] v[+t1.8]=TRiP. HMS02877}attP2/TM3, Sb[1] | Bloomington *Drosophila* Stock Center | BDSC:44633; FlyBase: FBst0044633; | |
| Genetic reagent (*D. melanogaster*) | *D. melanogaster.* Expresses dsRNA for RNAi of spen (FBgn0016977) under UAS control in the VALIUM22 vector.: y[1] v[1]; P{y[+t7.7] v[+t1.8]=TRiP. GLC01647}attP40 | Bloomington *Drosophila* Stock Center | BDSC:50529; FlyBase: FBst0050529; | |
| Genetic reagent (*D. melanogaster*) | *D. melanogaster.* Expresses dsRNA for RNAi of dlp (FBgn0041604) under UAS control in the VALIUM22 vector.: y[1] v[1]; P{y[+t7.7] v[+t1.8]=TRiP. GLC01658}attP40 | Bloomington *Drosophila* Stock Center | BDSC:50540; FlyBase: FBst0050540; | |
| Genetic reagent (*D. melanogaster*) | *D. melanogaster.* Expresses dsRNA for RNAi of wgn (FBgn0030941) under UAS control in the VALIUM22 vector.: y[1] sc[*] v[1] sev[21]; P{y[+t7.7] v[+t1.8]=TRiP.GLC01716}attP2 | Bloomington *Drosophila* Stock Center | BDSC:50594; FlyBase: FBst0050594; | |
| Genetic reagent (*D. melanogaster*) | *D. melanogaster.* Expresses dsRNA for RNAi of RpL8 (FBgn0261602) under UAS control in the VALIUM20 vector.: y[1] v[1]; P{y[+t7.7] v[+t1.8]=TRiP.7.7 v[+t1.8]=TRiP. HMC02977}attP2/TM3, Sb[1] | Bloomington *Drosophila* Stock Center | BDSC:50610; FlyBase: FBst0050610; | |
| Genetic reagent (*D. melanogaster*) | *D. melanogaster.* Expresses dsRNA for RNAi of Act42A (FBgn0000043) under UAS control in the VALIUM20 vector.: y[1] v[1]; P{y[+t7.7] v[+t1.8]=TRiP.7.7 v[+t1.8]=TRiP. HMC02992}attP2/TM3, Sb[1] | Bloomington *Drosophila* Stock Center | BDSC:50625; FlyBase: FBst0050625; | |

*Continued on next page*

*Continued*

| Reagent type (species) or resource | Designation | Source or reference | Identifiers | Additional information |
|---|---|---|---|---|
| Genetic reagent (*D. melanogaster*) | *D. melanogaster.* Expresses dsRNA for RNAi of tin (FBgn0004110) under UAS control in the VALIUM20 vector.: y[1] v[1]; P{y[+t7.7] v[+t1.8]=TRiP.7.7 v[+t1.8]=TRiP.HMC03064}attP2 | Bloomington *Drosophila* Stock Center | BDSC:50663; FlyBase: FBst0050663; | |
| Genetic reagent (*D. melanogaster*) | *D. melanogaster.* Expresses dsRNA for RNAi of EcR (FBgn0000546) under UAS control in the VALIUM20 vector.: y[1] v[1]; P{y[+t7.7] v[+t1.8]=TRiP.7.7 v[+t1.8]=TRiP.HMC03114}attP2/TM3, Sb[1] | Bloomington *Drosophila* Stock Center | BDSC:50712; FlyBase: FBst0050712; | |
| Genetic reagent (*D. melanogaster*) | *D. melanogaster.* Expresses dsRNA for RNAi of aru (FBgn0029095) under UAS control in the VALIUM20 vector.: y[1] sc[*] v[1] sev[21]; P{y[+t7.7] v[+t1.8]=TRiP.HMS02966}attP2 | Bloomington *Drosophila* Stock Center | BDSC:50730; FlyBase: FBst0050730; | |
| Genetic reagent (*D. melanogaster*) | *D. melanogaster.* Expresses dsRNA for RNAi of eIF4E4 (FBgn0035709) under UAS control in the VALIUM20 vector.: y[1] v[1]; P{y[+t7.7] v[+t1.8]=TRiP.7.7 v[+t1.8]=TRiP.HMJ21052}attP40 | Bloomington *Drosophila* Stock Center | BDSC:50951; FlyBase: FBst0050951; | |
| Genetic reagent (*D. melanogaster*) | *D. melanogaster.* Expresses dsRNA for RNAi of pont (FBgn0040078) under UAS control in the VALIUM20 vector.: y[1] v[1]; P{y[+t7.7] v[+t1.8]=TRiP.7.7 v[+t1.8]=TRiP.HMJ21078}attP40 | Bloomington *Drosophila* Stock Center | BDSC:50972; FlyBase: FBst0050972; | |
| Genetic reagent (*D. melanogaster*) | *D. melanogaster.* Expresses dsRNA for RNAi of dpn (FBgn0010109) under UAS control in the VALIUM20 vector.: y[1] v[1]; P{y[+t7.7] v[+t1.8]=TRiP.7.7 v[+t1.8]=TRiP. HMC03154}attP2 | Bloomington *Drosophila* Stock Center | BDSC:51440; FlyBase: FBst0051440; | |
| Genetic reagent (*D. melanogaster*) | *D. melanogaster.* Expresses dsRNA for RNAi of Spn27A (FBgn0028990) under UAS control in the VALIUM20 vector.: y[1] sc[*] v[1] sev[21]; P{y[+t7.7] v[+t1.8]=TRiP.HMC03159}attP2 | Bloomington *Drosophila* Stock Center | BDSC:51445; FlyBase: FBst0051445; | |
| Genetic reagent (*D. melanogaster*) | *D. melanogaster.* Expresses dsRNA for RNAi of ttv (FBgn0265974) under UAS control in the VALIUM20 vector.: y[1] v[1]; P{y[+t7.7] v[+t1.8]=TRiP.7.7 v[+t1.8]=TRiP.HMC03225}attP40 | Bloomington *Drosophila* Stock Center | BDSC:51480; FlyBase: FBst0051480; | |
| Genetic reagent (*D. melanogaster*) | *D. melanogaster.* Expresses dsRNA for RNAi of Mipp2 (FBgn0026060) under UAS control in the VALIUM20 vector.: y[1] v[1]; P{y[+t7.7] v[+t1.8]=TRiP.7.7 v[+t1.8]=TRiP.HMC03229}attP40/CyO | Bloomington *Drosophila* Stock Center | BDSC:51482; FlyBase: FBst0051482; | |

*Continued on next page*

*Continued*

| Reagent type (species) or resource | Designation | Source or reference | Identifiers | Additional information |
|---|---|---|---|---|
| Genetic reagent (*D. melanogaster*) | *D. melanogaster.* Expresses dsRNA for RNAi of kibra (FBgn0262127) under UAS control in the VALIUM20 vector.: y[1] sc[*] v[1] sev[21]; P{y[+t7.7] v[+t1.8]=TRiP.HMC03256}attP2 | Bloomington *Drosophila* Stock Center | BDSC:51499; FlyBase: FBst0051499; | |
| Genetic reagent (*D. melanogaster*) | *D. melanogaster.* Expresses dsRNA for RNAi of tld (FBgn0003719) under UAS control in the VALIUM20 vector.: y[1] sc[*] v[1] sev[21]; P{y[+t7.7] v[+t1.8]=TRiP.HMC03275}attP2 | Bloomington *Drosophila* Stock Center | BDSC:51507; FlyBase: FBst0051507; | |
| Genetic reagent (*D. melanogaster*) | *D. melanogaster.* Expresses dsRNA for RNAi of S6kII (FBgn0262866) under UAS control in the VALIUM20 vector.: y[1] sc[*] v[1] sev[21]; P{y[+t7.7] v[+t1.8]=TRiP. HMC03140}attP40 | Bloomington *Drosophila* Stock Center | BDSC:51694; FlyBase: FBst0051694; | |
| Genetic reagent (*D. melanogaster*) | *D. melanogaster.* Expresses dsRNA for RNAi of ast (FBgn0015905) under UAS control in the VALIUM20 vector.: y[1] v[1]; P{y[+t7.7] v[+t1.8]=TRiP.7.7 v[+t1.8]=TRiP. HMC03173}attP2 | Bloomington *Drosophila* Stock Center | BDSC:51700; FlyBase: FBst0051700; | |
| Genetic reagent (*D. melanogaster*) | *D. melanogaster.* Expresses dsRNA for RNAi of ras (FBgn0003204) under UAS control in the VALIUM20 vector.: y[1] sc[*] v[1] sev[21]; P{y[+t7.7] v[+t1.8]=TRiP.HMC03250}attP2 | Bloomington *Drosophila* Stock Center | BDSC:51717; FlyBase: FBst0051717; | |
| Genetic reagent (*D. melanogaster*) | *D. melanogaster.* Expresses dsRNA for RNAi of E(spl)mgamma-HLH (FBgn0002735) under UAS control in the VALIUM20 vector.: y[1] sc[*] v[1] sev[21]; P{y[+t7.7] v[+t1.8]=TRiP. HMC03315}attP2 | Bloomington *Drosophila* Stock Center | BDSC:51762; FlyBase: FBst0051762; | |
| Genetic reagent (*D. melanogaster*) | *D. melanogaster.* Expresses dsRNA for RNAi of Src64B (FBgn0262733) under UAS control in the VALIUM20 vector.: y[1] v[1]; P{y[+t7.7] v[+t1.8]=TRiP.7.7 v[+t1.8]=TRiP. HMC03327}attP40 | Bloomington *Drosophila* Stock Center | BDSC:51772; FlyBase: FBst0051772; | |
| Genetic reagent (*D. melanogaster*) | *D. melanogaster.* Expresses dsRNA for RNAi of brk (FBgn0024250) under UAS control in the VALIUM20 vector.: y[1] sc[*] v[1] sev[21]; P{y[+t7.7] v[+t1.8]=TRiP. HMC03345}attP2 | Bloomington *Drosophila* Stock Center | BDSC:51789; FlyBase: FBst0051789; | |
| Genetic reagent (*D. melanogaster*) | *D. melanogaster.* Expresses dsRNA for RNAi of Ext2 (FBgn0029175) under UAS control in the VALIUM20 vector.: y[1] sc[*] v[1] sev[21]; P{y[+t7.7] v[+t1.8]=TRiP. HMC03621}attP40 | Bloomington *Drosophila* Stock Center | BDSC:52883; FlyBase: FBst0052883; | |

*Continued on next page*

*Continued*

| Reagent type (species) or resource | Designation | Source or reference | Identifiers | Additional information |
|---|---|---|---|---|
| Genetic reagent (*D. melanogaster*) | *D. melanogaster*. Expresses dsRNA for RNAi of pen-2 (FBgn0053198) under UAS control in the VALIUM20 vector.: y[1] sc[*] v[1] sev[21]; P{y[+t7.7] v[+t1.8]=TRiP.HMC03648}attP40/CyO | Bloomington *Drosophila* Stock Center | BDSC:52908; FlyBase: FBst0052908; | |
| Genetic reagent (*D. melanogaster*) | *D. melanogaster*. Expresses dsRNA for RNAi of Tsc1 (FBgn0026317) under UAS control in the VALIUM20 vector.: y[1] sc[*] v[1] sev[21]; P{y[+t7.7] v[+t1.8]=TRiP.HMC03672}attP40 | Bloomington *Drosophila* Stock Center | BDSC:52931; FlyBase: FBst0052931; | |
| Genetic reagent (*D. melanogaster*) | *D. melanogaster*. Expresses dsRNA for RNAi of NT1 (FBgn0261526) under UAS control in the VALIUM20 vector.: y[1] v[1]; P{y[+t7.7] v[+t1.8]=TRiP.7.7 v[+t1.8]=TRiP.HMJ21720}attP40 | Bloomington *Drosophila* Stock Center | BDSC:53003; FlyBase: FBst0053003; | |
| Genetic reagent (*D. melanogaster*) | *D. melanogaster*. Expresses dsRNA for RNAi of arr (FBgn0000119) under UAS control in the VALIUM20 vector.: y[1] v[1]; P{y[+t7.7] v[+t1.8]=TRiP.7.7 v[+t1.8]=TRiP.HMC03571}attP40 | Bloomington *Drosophila* Stock Center | BDSC:53342; FlyBase: FBst0053342; | |
| Genetic reagent (*D. melanogaster*) | *D. melanogaster*. Expresses dsRNA for RNAi of kermit (FBgn0010504) under UAS control in the VALIUM20 vector.: y[1] sc[*] v[1] sev[21]; P{y[+t7.7] v[+t1.8]=TRiP.HMC03578}attP40 | Bloomington *Drosophila* Stock Center | BDSC:53349; FlyBase: FBst0053349; | |
| Genetic reagent (*D. melanogaster*) | *D. melanogaster*. Expresses dsRNA for RNAi of Sirt1 (FBgn0024291) under UAS control in the VALIUM20 vector.: y[1] v[1]; P{y[+t7.7] v[+t1.8]=TRiP.7.7 v[+t1.8]=TRiP.HMJ21708}attP40 | Bloomington *Drosophila* Stock Center | BDSC:53697; FlyBase: FBst0053697; | |
| Genetic reagent (*D. melanogaster*) | *D. melanogaster*. Expresses dsRNA for RNAi of tow (FBgn0035719) under UAS control in the VALIUM20 vector.: y[1] v[1]; P{y[+t7.7] v[+t1.8]=TRiP.7.7 v[+t1.8]=TRiP.HMJ21747}attP40 | Bloomington *Drosophila* Stock Center | BDSC:53704; FlyBase: FBst0053704; | |
| Genetic reagent (*D. melanogaster*) | *D. melanogaster*. Expresses dsRNA for RNAi of eIF4E3 (FBgn0265089) under UAS control in the VALIUM20 vector.: y[1] v[1]; P{y[+t7.7] v[+t1.8]=TRiP.7.7 v[+t1.8]=TRiP.HMJ21195}attP40 | Bloomington *Drosophila* Stock Center | BDSC:53880; FlyBase: FBst0053880; | |
| Genetic reagent (*D. melanogaster*) | *D. melanogaster*. Expresses dsRNA for RNAi of hppy (FBgn0263395) under UAS control in the VALIUM20 vector.: y[1] v[1]; P{y[+t7.7] v[+t1.8]=TRiP.7.7 v[+t1.8]=TRiP.HMJ21199}attP40 | Bloomington *Drosophila* Stock Center | BDSC:53884; FlyBase: FBst0053884; | |

*Continued on next page*

*Continued*

| Reagent type (species) or resource | Designation | Source or reference | Identifiers | Additional information |
|---|---|---|---|---|
| Genetic reagent (*D. melanogaster*) | *D. melanogaster*. Expresses dsRNA for RNAi of dome (FBgn0043903) under UAS control in the VALIUM20 vector.: y[1] v[1]; P{y[+t7.7] v[+t1.8]=TRiP.7.7 v[+t1.8]=TRiP. HMJ21208}attP40 | Bloomington *Drosophila* Stock Center | BDSC:53890; FlyBase: FBst0053890; | |
| Genetic reagent (*D. melanogaster*) | *D. melanogaster*. Expresses dsRNA for RNAi of Ocho (FBgn0040296) under UAS control in the VALIUM20 vector.: y[1] v[1]; P{y[+t7.7] v[+t1.8]=TRiP.7.7 v[+t1.8]=TRiP. HMJ21588}attP40/CyO | Bloomington *Drosophila* Stock Center | BDSC:54851; FlyBase: FBst0054851; | |
| Genetic reagent (*D. melanogaster*) | *D. melanogaster*. Expresses dsRNA for RNAi of sqd (FBgn0263396) under UAS control in the VALIUM20 vector.: y[1] sc[*] v[1] sev[21]; P{y[+t7.7] v[+t1.8]=TRiP. HMC03848}attP40 | Bloomington *Drosophila* Stock Center | BDSC:55169; FlyBase: FBst0055169; | |
| Genetic reagent (*D. melanogaster*) | *D. melanogaster*. Expresses dsRNA for RNAi of mago (FBgn0002736) under UAS control in the VALIUM20 vector.: y[1] sc[*] v[1] sev[21]; P{y[+t7.7] v[+t1.8]=TRiP. HMC03947}attP40 | Bloomington *Drosophila* Stock Center | BDSC:55260; FlyBase: FBst0055260; | |
| Genetic reagent (*D. melanogaster*) | *D. melanogaster*. Expresses dsRNA for RNAi of egr (FBgn0033483) under UAS control in the VALIUM20 vector.: y[1] sc[*] v[1] sev[21]; P{y[+t7.7] v[+t1.8]=TRiP. HMC03963}attP40 | Bloomington *Drosophila* Stock Center | BDSC:55276; FlyBase: FBst0055276; | |
| Genetic reagent (*D. melanogaster*) | *D. melanogaster*. Expresses dsRNA for RNAi of E(spl)m3-HLH (FBgn0002609) under UAS control in the VALIUM20 vector.: y[1] v[1]; P{y[+t7.7] v[+t1.8]=TRiP.7.7 v[+t1.8]=TRiP. HMC03989}attP2 | Bloomington *Drosophila* Stock Center | BDSC:55302; FlyBase: FBst0055302; | |
| Genetic reagent (*D. melanogaster*) | *D. melanogaster*. Expresses dsRNA for RNAi of Coprox (FBgn0021944) under UAS control in the VALIUM20 vector.: y[1] v[1]; P{y[+t7.7] v[+t1.8]=TRiP.7.7 v[+t1.8]=TRiP. HMC04005}attP40/CyO | Bloomington *Drosophila* Stock Center | BDSC:55318; FlyBase: FBst0055318; | |
| Genetic reagent (*D. melanogaster*) | *D. melanogaster*. Expresses dsRNA for RNAi of Rab23 (FBgn0037364) under UAS control in the VALIUM20 vector.: y[1] sc[*] v[1] sev[21]; P{y[+t7.7] v[+t1.8]=TRiP. HMC04039}attP40 | Bloomington *Drosophila* Stock Center | BDSC:55352; FlyBase: FBst0055352; | |

*Continued on next page*

*Continued*

| Reagent type (species) or resource | Designation | Source or reference | Identifiers | Additional information |
|---|---|---|---|---|
| Genetic reagent (*D. melanogaster*) | *D. melanogaster*. Expresses dsRNA for RNAi of tsu (FBgn0033378) under UAS control in the VALIUM20 vector.: y[1] sc[*] v[1] sev[21]; P{y[+t7.7] v[+t1.8]=TRiP.HMC04055}attP40 | Bloomington *Drosophila* Stock Center | BDSC:55367; FlyBase: FBst0055367; | |
| Genetic reagent (*D. melanogaster*) | *D. melanogaster*. Expresses dsRNA for RNAi of Notum (FBgn0044028) under UAS control in the VALIUM20 vector.: y[1] sc[*] v[1] sev[21]; P{y[+t7.7] v[+t1.8]=TRiP.HMC04067}attP40 | Bloomington *Drosophila* Stock Center | BDSC:55379; FlyBase: FBst0055379; | |
| Genetic reagent (*D. melanogaster*) | *D. melanogaster*. Expresses dsRNA for RNAi of sd (FBgn0003345) under UAS control in the VALIUM20 vector.: y[1] sc[*] v[1] sev[21]; P{y[+t7.7] v[+t1.8]=TRiP.HMC04092}attP40 | Bloomington *Drosophila* Stock Center | BDSC:55404; FlyBase: FBst0055404; | |
| Genetic reagent (*D. melanogaster*) | *D. melanogaster*. Expresses dsRNA for RNAi of Raf (FBgn0003079) under UAS control in the VALIUM20 vector.: y[1] sc[*] v[1] sev[21]; P{y[+t7.7] v[+t1.8]=TRiP.HMC03854}attP2 | Bloomington *Drosophila* Stock Center | BDSC:55679; FlyBase: FBst0055679; | |
| Genetic reagent (*D. melanogaster*) | *D. melanogaster*. Expresses dsRNA for RNAi of Mo25 (FBgn0017572) under UAS control in the VALIUM20 vector.: y[1] sc[*] v[1] sev[21]; P{y[+t7.7] v[+t1.8]=TRiP.HMC03865}attP2 | Bloomington *Drosophila* Stock Center | BDSC:55681; FlyBase: FBst0055681; | |
| Genetic reagent (*D. melanogaster*) | *D. melanogaster*. Expresses dsRNA for RNAi of ptc (FBgn0003892) under UAS control in the VALIUM20 vector.: y[1] sc[*] v[1] sev[21]; P{y[+t7.7] v[+t1.8]=TRiP.HMC03872}attP40 | Bloomington *Drosophila* Stock Center | BDSC:55686; FlyBase: FBst0055686; | |
| Genetic reagent (*D. melanogaster*) | *D. melanogaster*. Expresses dsRNA for RNAi of Pka-C2 (FBgn0000274) under UAS control in the VALIUM20 vector.: y[1] sc[*] v[1] sev[21]; P{y[+t7.7] v[+t1.8]=TRiP.HMC04129}attP2 | Bloomington *Drosophila* Stock Center | BDSC:55859; FlyBase: FBst0055859; | |
| Genetic reagent (*D. melanogaster*) | *D. melanogaster*. Expresses dsRNA for RNAi of sev (FBgn0003366) under UAS control in the VALIUM20 vector.: y[1] v[1]; P{y[+t7.7] v[+t1.8]=TRiP.7.7 v[+t1.8]=TRiP.HMC04136}attP2 | Bloomington *Drosophila* Stock Center | BDSC:55866; FlyBase: FBst0055866; | |
| Genetic reagent (*D. melanogaster*) | *D. melanogaster*. Expresses dsRNA for RNAi of shf (FBgn0003390) under UAS control in the VALIUM20 vector.: y[1] sc[*] v[1] sev[21]; P{y[+t7.7] v[+t1.8]=TRiP.HMC04137}attP2 | Bloomington *Drosophila* Stock Center | BDSC:55867; FlyBase: FBst0055867; | |

*Continued on next page*

*Continued*

| Reagent type (species) or resource | Designation | Source or reference | Identifiers | Additional information |
|---|---|---|---|---|
| Genetic reagent (*D. melanogaster*) | *D. melanogaster.* Expresses dsRNA for RNAi of babo (FBgn0011300) under UAS control in the VALIUM20 vector.: y[1] sc[*] v[1] sev[21]; P{y[+t7.7] v[+t1.8]=TRiP.HMC04142}attP2/TM3, Sb[1] | Bloomington *Drosophila* Stock Center | BDSC:55871; FlyBase: FBst0055871; | |
| Genetic reagent (*D. melanogaster*) | *D. melanogaster.* Expresses dsRNA for RNAi of Shark (FBgn0015295) under UAS control in the VALIUM20 vector.: y[1] sc[*] v[1] sev[21]; P{y[+t7.7] v[+t1.8]=TRiP.HMC04146}attP2 | Bloomington *Drosophila* Stock Center | BDSC:55874; FlyBase: FBst0055874; | |
| Genetic reagent (*D. melanogaster*) | *D. melanogaster.* Expresses dsRNA for RNAi of Drl-2 (FBgn0033791) under UAS control in the VALIUM20 vector.: y[1] sc[*] v[1] sev[21]; P{y[+t7.7] v[+t1.8]=TRiP.HMC04172}attP2 | Bloomington *Drosophila* Stock Center | BDSC:55893; FlyBase: FBst0055893; | |
| Genetic reagent (*D. melanogaster*) | *D. melanogaster.* Expresses dsRNA for RNAi of Takl2 (FBgn0039015) under UAS control in the VALIUM20 vector.: y[1] v[1]; P{y[+t7.7] v[+t1.8]=TRiP.7.7 v[+t1.8]=TRiP.HMC04181}attP2 | Bloomington *Drosophila* Stock Center | BDSC:55899; FlyBase: FBst0055899; | |
| Genetic reagent (*D. melanogaster*) | *D. melanogaster.* Expresses dsRNA for RNAi of Takl1 (FBgn0046689) under UAS control in the VALIUM20 vector.: y[1] sc[*] v[1] sev[21]; P{y[+t7.7] v[+t1.8]=TRiP.HMC04186}attP2 | Bloomington *Drosophila* Stock Center | BDSC:55903; FlyBase: FBst0055903; | |
| Genetic reagent (*D. melanogaster*) | *D. melanogaster.* Expresses dsRNA for RNAi of Doa (FBgn0265998) under UAS control in the VALIUM20 vector.: y[1] v[1]; P{y[+t7.7] v[+t1.8]=TRiP.7.7 v[+t1.8]=TRiP.HMC04193}attP2 | Bloomington *Drosophila* Stock Center | BDSC:55908; FlyBase: FBst0055908; | |
| Genetic reagent (*D. melanogaster*) | *D. melanogaster.* Expresses dsRNA for RNAi of fus (FBgn0023441) under UAS control in the VALIUM20 vector.: y[1] sc[*] v[1] sev[21]; P{y[+t7.7] v[+t1.8]=TRiP.HMC04208}attP40 | Bloomington *Drosophila* Stock Center | BDSC:55921; FlyBase: FBst0055921; | |
| Genetic reagent (*D. melanogaster*) | *D. melanogaster.* Expresses dsRNA for RNAi of grk (FBgn0001137) under UAS control in the VALIUM20 vector.: y[1] sc[*] v[1] sev[21]; P{y[+t7.7] v[+t1.8]=TRiP.HMC04213}attP40 | Bloomington *Drosophila* Stock Center | BDSC:55926; FlyBase: FBst0055926; | |
| Genetic reagent (*D. melanogaster*) | *D. melanogaster.* Expresses dsRNA for RNAi of PpD3 (FBgn0005777) under UAS control in the VALIUM20 vector.: y[1] sc[*] v[1] sev[21]; P{y[+t7.7] v[+t1.8]=TRiP.HMS04508}attP40 | Bloomington *Drosophila* Stock Center | BDSC:57307; FlyBase: FBst0057307; | |

*Continued on next page*

*Continued*

| Reagent type (species) or resource | Designation | Source or reference | Identifiers | Additional information |
|---|---|---|---|---|
| Genetic reagent (*D. melanogaster*) | *D. melanogaster.* Expresses dsRNA for RNAi of Nct (FBgn0039234) under UAS control in the VALIUM20 vector.: y[1] sc[*] v[1] sev[21]; P{y[+t7.7] v[+t1.8]=TRiP.HMC04812}attP40 | Bloomington *Drosophila* Stock Center | BDSC:57497; FlyBase: FBst0057497; | |
| Genetic reagent (*D. melanogaster*) | *D. melanogaster.* Expresses dsRNA for RNAi of CG15436 (FBgn0031610) under UAS control in the VALIUM20 vector.: y[1] sc[*] v[1] sev[21]; P{y[+t7.7] v[+t1.8]=TRiP.HMC04637}attP40 | Bloomington *Drosophila* Stock Center | BDSC:57867; FlyBase: FBst0057867; | |
| Genetic reagent (*D. melanogaster*) | *D. melanogaster.* Expresses dsRNA for RNAi of Su(var)2–10 (FBgn0003612) under UAS control in the VALIUM20 vector.: y[1] v[1]; P{y[+t7.7] v[+t1.8]=TRiP.7.7 v[+t1.8]=TRiP.HMJ21959}attP40 | Bloomington *Drosophila* Stock Center | BDSC:58067; FlyBase: FBst0058067; | |
| Genetic reagent (*D. melanogaster*) | *D. melanogaster.* Expresses dsRNA for RNAi of Atg8a (FBgn0052672) under UAS control in the VALIUM20 vector.: y[1] v[1]; P{y[+t7.7] v[+t1.8]=TRiP.7.7 v[+t1.8]=TRiP.HMJ22416}attP40 | Bloomington *Drosophila* Stock Center | BDSC:58309; FlyBase: FBst0058309; | |
| Genetic reagent (*D. melanogaster*) | *D. melanogaster.* Expresses dsRNA for RNAi of spz (FBgn0003495) under UAS control in the VALIUM20 vector.: y[1] v[1]; P{y[+t7.7] v[+t1.8]=TRiP.7.7 v[+t1.8]=TRiP.HMJ22258}attP40 | Bloomington *Drosophila* Stock Center | BDSC:58499; FlyBase: FBst0058499; | |
| Genetic reagent (*D. melanogaster*) | *D. melanogaster.* Expresses dsRNA for RNAi of AstC-R1 (FBgn0036790) under UAS control in the VALIUM20 vector.: y[1] v[1]; P{y[+t7.7] v[+t1.8]=TRiP.7.7 v[+t1.8]=TRiP.HMJ23767}attP40/CyO | Bloomington *Drosophila* Stock Center | BDSC:62372; FlyBase: FBst0062372; | |
| Genetic reagent (*D. melanogaster*) | *D. melanogaster.* Expresses dsRNA for RNAi of Ku80 (FBgn0041627) under UAS control in the VALIUM20 vector.: y[1] v[1]; P{y[+t7.7] v[+t1.8]=TRiP.7.7 v[+t1.8]=TRiP.HMJ24057}attP40/CyO | Bloomington *Drosophila* Stock Center | BDSC:62513; FlyBase: FBst0062513; | |
| Genetic reagent (*D. melanogaster*) | *D. melanogaster.* Expresses dsRNA for RNAi of CG2199 (FBgn0035213) under UAS control in the VALIUM20 vector.: y[1] v[1]; P{y[+t7.7] v[+t1.8]=TRiP.7.7 v[+t1.8]=TRiP.HMJ30228}attP40 | Bloomington *Drosophila* Stock Center | BDSC:63661; FlyBase: FBst0063661; | |
| Genetic reagent (*D. melanogaster*) | *D. melanogaster.* Expresses dsRNA for RNAi of tsr (FBgn0011726) under UAS control in the VALIUM20 vector.: y[1] sc[*] v[1] sev[21]; P{y[+t7.7] v[+t1.8]=TRiP.HMS00534}attP2 | Bloomington *Drosophila* Stock Center | BDSC:65055; FlyBase: FBst0065055; | |

*Continued on next page*

*Continued*

| Reagent type (species) or resource | Designation | Source or reference | Identifiers | Additional information |
|---|---|---|---|---|
| Genetic reagent (*D. melanogaster*) | *D. melanogaster.* Expresses dsRNA for RNAi of CG3630 (FBgn0023540) under UAS control in the VALIUM20 vector.: y[1] sc[*] v[1] sev[21]; P{y[+t7.7] v[+t1.8]=TRiP. HMC06220}attP2 | Bloomington *Drosophila* Stock Center | BDSC:65945; FlyBase: FBst0065945; | |
| Genetic reagent (*D. melanogaster*) | *D. melanogaster.* Expresses dsRNA for RNAi of tub (FBgn0003882) under UAS control in the VALIUM20 vector.: y[1] sc[*] v[1] sev[21]; P{y[+t7.7] v[+t1.8]=TRiP. HMS05426}attP40 | Bloomington *Drosophila* Stock Center | BDSC:66960; FlyBase: FBst0066960; | |
| Genetic reagent (*D. melanogaster*) | *D. melanogaster.* Expresses dsRNA for RNAi of eIF4E-6 (FBgn0039622) under UAS control. | Vienna *Drosophila* Resource Center | VDRC:v17580 | |
| Genetic reagent (*D. melanogaster*) | *D. melanogaster.* Expresses dsRNA for RNAi of gd (FBgn0000808) under UAS control. | Vienna *Drosophila* Resource Center | VDRC:v14892 | |
| Genetic reagent (*D. melanogaster*) | *D. melanogaster.* Expresses dsRNA for RNAi of Act88F (FBgn0000047) under UAS control. | Vienna *Drosophila* Resource Center | VDRC:v9780 | |
| Genetic reagent (*D. melanogaster*) | *D. melanogaster.* Expresses dsRNA for RNAi of eRF1 (FBgn0036974) under UAS control. | Vienna *Drosophila* Resource Center | VDRC:v45027 | |
| Genetic reagent (*D. melanogaster*) | *D. melanogaster.* Expresses dsRNA for RNAi of E(spl)m2-BFM (FBgn0002592) under UAS control. | Vienna *Drosophila* Resource Center | VDRC:v30115 | |
| Genetic reagent (*D. melanogaster*) | *D. melanogaster.* Expresses dsRNA for RNAi of tsl (FBgn0003867) under UAS control. | Vienna *Drosophila* Resource Center | VDRC:v14430 | |
| Genetic reagent (*D. melanogaster*) | *D. melanogaster.* Expresses dsRNA for RNAi of wbl (FBgn0004003) under UAS control. | Vienna *Drosophila* Resource Center | VDRC:v13864 | |
| Genetic reagent (*D. melanogaster*) | *D. melanogaster.* Expresses dsRNA for RNAi of boss (FBgn0000206) under UAS control. | Vienna *Drosophila* Resource Center | VDRC:v4365 | |
| Genetic reagent (*D. melanogaster*) | *D. melanogaster.* Expresses dsRNA for RNAi of CG32396 (FBgn0020251) under UAS control. | Vienna *Drosophila* Resource Center | VDRC:v41896 | |
| Genetic reagent (*D. melanogaster*) | *D. melanogaster.* Expresses dsRNA for RNAi of Rac2 (FBgn0014011) under UAS control. | Vienna *Drosophila* Resource Center | VDRC:v28926 | |
| Genetic reagent (*D. melanogaster*) | *D. melanogaster.* Expresses dsRNA for RNAi of ihog (FBgn0031872) under UAS control. | Vienna *Drosophila* Resource Center | VDRC:v29898 | |

*Continued on next page*

*Continued*

| Reagent type (species) or resource | Designation | Source or reference | Identifiers | Additional information |
|---|---|---|---|---|
| Genetic reagent (*D. melanogaster*) | *D. melanogaster.* Expresses dsRNA for RNAi of sog (FBgn0003463) under UAS control. | Vienna *Drosophila* Resource Center | VDRC:v37405 | |
| Genetic reagent (*D. melanogaster*) | *D. melanogaster.* Expresses dsRNA for RNAi of CG9314 (FBgn0032061) under UAS control. | Vienna *Drosophila* Resource Center | VDRC:v44647 | |
| Genetic reagent (*D. melanogaster*) | *D. melanogaster.* Expresses dsRNA for RNAi of sgl (FBgn0261445) under UAS control. | Vienna *Drosophila* Resource Center | VDRC:v29434 | |
| Genetic reagent (*D. melanogaster*) | *D. melanogaster.* Expresses dsRNA for RNAi of mirr (FBgn0014343) under UAS control. | Vienna *Drosophila* Resource Center | VDRC:v50134 | |
| Genetic reagent (*D. melanogaster*) | *D. melanogaster.* Expresses dsRNA for RNAi of eIF-4B (FBgn0020660) under UAS control. | Vienna *Drosophila* Resource Center | VDRC:v31364 | |
| Genetic reagent (*D. melanogaster*) | *D. melanogaster.* Expresses dsRNA for RNAi of rasp (FBgn0024194) under UAS control. | Vienna *Drosophila* Resource Center | VDRC:v6459 | |
| Genetic reagent (*D. melanogaster*) | *D. melanogaster.* Expresses dsRNA for RNAi of PGRP-SA (FBgn0030310) under UAS control. | Vienna *Drosophila* Resource Center | VDRC:v5594 | |
| Genetic reagent (*D. melanogaster*) | *D. melanogaster.* Expresses dsRNA for RNAi of sinah (FBgn0259794) under UAS control. | Vienna *Drosophila* Resource Center | VDRC:v17118 | |
| Genetic reagent (*D. melanogaster*) | *D. melanogaster.* Expresses dsRNA for RNAi of lft (FBgn0032230) under UAS control. | Vienna *Drosophila* Resource Center | VDRC:v32146 | |
| Genetic reagent (*D. melanogaster*) | *D. melanogaster.* Expresses dsRNA for RNAi of pyr (FBgn0033649) under UAS control. | Vienna *Drosophila* Resource Center | VDRC:v36524 | |
| Genetic reagent (*D. melanogaster*) | *D. melanogaster.* Expresses dsRNA for RNAi of Ssl (FBgn0015300) under UAS control. | Vienna *Drosophila* Resource Center | VDRC:v17282 | |
| Genetic reagent (*D. melanogaster*) | *D. melanogaster.* Expresses dsRNA for RNAi of Pvf2 (FBgn0031888) under UAS control. | Vienna *Drosophila* Resource Center | VDRC:v7628 | |
| Genetic reagent (*D. melanogaster*) | *D. melanogaster.* Expresses dsRNA for RNAi of spz4 (FBgn0032362) under UAS control. | Vienna *Drosophila* Resource Center | VDRC:v7679 | |
| Genetic reagent (*D. melanogaster*) | *D. melanogaster.* Expresses dsRNA for RNAi of IM23 (FBgn0034328) under UAS control. | Vienna *Drosophila* Resource Center | VDRC:v15384 | |
| Genetic reagent (*D. melanogaster*) | *D. melanogaster.* Expresses dsRNA for RNAi of botv (FBgn0027535) under UAS control. | Vienna *Drosophila* Resource Center | VDRC:v37186 | |

*Continued on next page*

*Continued*

| Reagent type (species) or resource | Designation | Source or reference | Identifiers | Additional information |
|---|---|---|---|---|
| Genetic reagent (*D. melanogaster*) | *D. melanogaster.* Expresses dsRNA for RNAi of G6P (FBgn0031463) under UAS control. | Vienna *Drosophila* Resource Center | VDRC:v7261 | |
| Genetic reagent (*D. melanogaster*) | *D. melanogaster.* Expresses dsRNA for RNAi of stet (FBgn0020248) under UAS control. | Vienna *Drosophila* Resource Center | VDRC:v7434 | |
| Genetic reagent (*D. melanogaster*) | *D. melanogaster.* Expresses dsRNA for RNAi of RanBPM (FBgn0262114) under UAS control. | Vienna *Drosophila* Resource Center | VDRC:v45981 | |
| Genetic reagent (*D. melanogaster*) | *D. melanogaster.* Expresses dsRNA for RNAi of Src42A (FBgn0264959) under UAS control. | Vienna *Drosophila* Resource Center | VDRC:v26019 | |
| Genetic reagent (*D. melanogaster*) | *D. melanogaster.* Expresses dsRNA for RNAi of nkd (FBgn0002945) under UAS control. | Vienna *Drosophila* Resource Center | VDRC:v3004 | |
| Genetic reagent (*D. melanogaster*) | *D. melanogaster.* Expresses dsRNA for RNAi of boi (FBgn0040388) under UAS control. | Vienna *Drosophila* Resource Center | VDRC:v3060 | |
| Genetic reagent (*D. melanogaster*) | *D. melanogaster.* Expresses dsRNA for RNAi of CR45683; Tehao (FBgn0026760) under UAS control. | Vienna *Drosophila* Resource Center | VDRC:v17903 | |
| Genetic reagent (*D. melanogaster*) | *D. melanogaster.* Expresses dsRNA for RNAi of CG45087; Pepck (FBgn0003067) under UAS control. | Vienna *Drosophila* Resource Center | VDRC:v20529 | |
| Genetic reagent (*D. melanogaster*) | *D. melanogaster.* Expresses dsRNA for RNAi of RpL13A (FBgn0037351) under UAS control. | Vienna *Drosophila* Resource Center | VDRC:v101369 | |
| Genetic reagent (*D. melanogaster*) | *D. melanogaster.* Expresses dsRNA for RNAi of Usp7 (FBgn0030366) under UAS control. | Vienna *Drosophila* Resource Center | VDRC:v110324 | |
| Genetic reagent (*D. melanogaster*) | *D. melanogaster.* Expresses dsRNA for RNAi of tsg (FBgn0003865) under UAS control. | Vienna *Drosophila* Resource Center | VDRC:v108750 | |
| Genetic reagent (*D. melanogaster*) | *D. melanogaster.* Expresses dsRNA for RNAi of nec (FBgn0002930) under UAS control. | Vienna *Drosophila* Resource Center | VDRC:v108366 | |
| Genetic reagent (*D. melanogaster*) | *D. melanogaster.* Expresses dsRNA for RNAi of Nle (FBgn0021874) under UAS control. | Vienna *Drosophila* Resource Center | VDRC:v110728 | |
| Genetic reagent (*D. melanogaster*) | *D. melanogaster.* Expresses dsRNA for RNAi of Brd (FBgn0000216) under UAS control. | Vienna *Drosophila* Resource Center | VDRC:v107929 | |

*Continued*

| Reagent type (species) or resource | Designation | Source or reference | Identifiers | Additional information |
|---|---|---|---|---|
| Genetic reagent (*D. melanogaster*) | *D. melanogaster*. Expresses dsRNA for RNAi of Shc (FBgn0015296) under UAS control. | Vienna *Drosophila* Resource Center | VDRC:v103906 | |
| Genetic reagent (*D. melanogaster*) | *D. melanogaster*. Expresses dsRNA for RNAi of Hs6st (FBgn0038755) under UAS control. | Vienna *Drosophila* Resource Center | VDRC:v110424 | |
| Genetic reagent (*D. melanogaster*) | *D. melanogaster*. Expresses dsRNA for RNAi of fz4 (FBgn0027342) under UAS control. | Vienna *Drosophila* Resource Center | VDRC:v102339 | |
| Genetic reagent (*D. melanogaster*) | *D. melanogaster*. Expresses dsRNA for RNAi of bib (FBgn0000180) under UAS control. | Vienna *Drosophila* Resource Center | VDRC:v103327 | |
| Genetic reagent (*D. melanogaster*) | *D. melanogaster*. Expresses dsRNA for RNAi of Wnt10 (FBgn0031903) under UAS control. | Vienna *Drosophila* Resource Center | VDRC:v100867 | |
| Genetic reagent (*D. melanogaster*) | *D. melanogaster*. Expresses dsRNA for RNAi of Tom (FBgn0026320) under UAS control. | Vienna *Drosophila* Resource Center | VDRC:v101652 | |
| Genetic reagent (*D. melanogaster*) | *D. melanogaster*. Expresses dsRNA for RNAi of Pli (FBgn0025574) under UAS control. | Vienna *Drosophila* Resource Center | VDRC:v106776 | |
| Genetic reagent (*D. melanogaster*) | *D. melanogaster*. Expresses dsRNA for RNAi of drk (FBgn0004638) under UAS control. | Vienna *Drosophila* Resource Center | VDRC:v105498 | |
| Genetic reagent (*D. melanogaster*) | *D. melanogaster*. Expresses dsRNA for RNAi of por (FBgn0004957) under UAS control. | Vienna *Drosophila* Resource Center | VDRC:v100780 | |
| Genetic reagent (*D. melanogaster*) | *D. melanogaster*. Expresses dsRNA for RNAi of wls (FBgn0036141) under UAS control. | Vienna *Drosophila* Resource Center | VDRC:v103812 | |
| Genetic reagent (*D. melanogaster*) | *D. melanogaster*. Expresses dsRNA for RNAi of CG6843 (FBgn0036827) under UAS control. | Vienna *Drosophila* Resource Center | VDRC:v109411 | |
| Genetic reagent (*D. melanogaster*) | *D. melanogaster*. Expresses dsRNA for RNAi of spz3 (FBgn0031959) under UAS control. | Vienna *Drosophila* Resource Center | VDRC:v102871 | |
| Genetic reagent (*D. melanogaster*) | *D. melanogaster*. Expresses dsRNA for RNAi of kuz (FBgn0259984) under UAS control. | Vienna *Drosophila* Resource Center | VDRC:v107036 | |
| Genetic reagent (*D. melanogaster*) | *D. melanogaster*. Expresses dsRNA for RNAi of Hs3st-B (FBgn0031005) under UAS control. | Vienna *Drosophila* Resource Center | VDRC:v110601 | |
| Genetic reagent (*D. melanogaster*) | *D. melanogaster*. Expresses dsRNA for RNAi of Tace (FBgn0039734) under UAS control. | Vienna *Drosophila* Resource Center | VDRC:v106335 | |

*Continued on next page*

*Continued*

| Reagent type (species) or resource | Designation | Source or reference | Identifiers | Additional information |
|---|---|---|---|---|
| Genetic reagent (*D. melanogaster*) | *D. melanogaster.* Expresses dsRNA for RNAi of eIF-1A (FBgn0026250) under UAS control. | Vienna *Drosophila* Resource Center | VDRC:v100611 | |
| Genetic reagent (*D. melanogaster*) | *D. melanogaster.* Expresses dsRNA for RNAi of Socs16D (FBgn0030869) under UAS control. | Vienna *Drosophila* Resource Center | VDRC:v100568 | |
| Genetic reagent (*D. melanogaster*) | *D. melanogaster.* Expresses dsRNA for RNAi of E(spl)malpha-BFM (FBgn0002732) under UAS control. | Vienna *Drosophila* Resource Center | VDRC:v109384 | |
| Genetic reagent (*D. melanogaster*) | *D. melanogaster.* Expresses dsRNA for RNAi of E(spl)m6-BFM (FBgn0002632) under UAS control. | Vienna *Drosophila* Resource Center | VDRC:v101965 | |
| Genetic reagent (*D. melanogaster*) | *D. melanogaster.* Expresses dsRNA for RNAi of VhaM8.9 (FBgn0037671) under UAS control. | Vienna *Drosophila* Resource Center | VDRC:v105281 | |
| Genetic reagent (*D. melanogaster*) | *D. melanogaster.* Expresses dsRNA for RNAi of melt (FBgn0023001) under UAS control. | Vienna *Drosophila* Resource Center | VDRC:v105110 | |
| Genetic reagent (*D. melanogaster*) | *D. melanogaster.* Expresses dsRNA for RNAi of SkpB (FBgn0026176) under UAS control. | Vienna *Drosophila* Resource Center | VDRC:v106521 | |
| Genetic reagent (*D. melanogaster*) | *D. melanogaster.* Expresses dsRNA for RNAi of CkIIbeta2 (FBgn0026136) under UAS control. | Vienna *Drosophila* Resource Center | VDRC:v102633 | |
| Genetic reagent (*D. melanogaster*) | *D. melanogaster.* Expresses dsRNA for RNAi of spz6 (FBgn0035056) under UAS control. | Vienna *Drosophila* Resource Center | VDRC:v100897 | |
| Genetic reagent (*D. melanogaster*) | *D. melanogaster.* Expresses dsRNA for RNAi of Rok (FBgn0026181) under UAS control. | Vienna *Drosophila* Resource Center | VDRC:v104675 | |
| Genetic reagent (*D. melanogaster*) | *D. melanogaster.* Expresses dsRNA for RNAi of CG9962 (FBgn0031441) under UAS control. | Vienna *Drosophila* Resource Center | VDRC:v108721 | |
| Genetic reagent (*D. melanogaster*) | *D. melanogaster.* Expresses dsRNA for RNAi of spz5 (FBgn0035379) under UAS control. | Vienna *Drosophila* Resource Center | VDRC:v102389 | |
| Genetic reagent (*D. melanogaster*) | *D. melanogaster.* Expresses dsRNA for RNAi of Act57B (FBgn0000044) under UAS control. | Vienna *Drosophila* Resource Center | VDRC:v102129 | |
| Genetic reagent (*D. melanogaster*) | *D. melanogaster.* Expresses dsRNA for RNAi of ndl (FBgn0002926) under UAS control. | Vienna *Drosophila* Resource Center | VDRC:v102818 | |

*Continued on next page*

Continued

| Reagent type (species) or resource | Designation | Source or reference | Identifiers | Additional information |
|---|---|---|---|---|
| Genetic reagent (*D. melanogaster*) | *D. melanogaster.* Expresses dsRNA for RNAi of vn (FBgn0003984) under UAS control. | Vienna *Drosophila* Resource Center | VDRC:v109437 | |
| Genetic reagent (*D. melanogaster*) | *D. melanogaster.* Expresses dsRNA for RNAi of ECSIT (FBgn0028436) under UAS control. | Vienna *Drosophila* Resource Center | VDRC:v106141 | |
| Genetic reagent (*D. melanogaster*) | *D. melanogaster.* Expresses dsRNA for RNAi of SkpE (FBgn0031074) under UAS control. | Vienna *Drosophila* Resource Center | VDRC:v109539 | |
| Genetic reagent (*D. melanogaster*) | *D. melanogaster.* Expresses dsRNA for RNAi of SkpF (FBgn0034863) under UAS control. | Vienna *Drosophila* Resource Center | VDRC:v106572 | |
| Genetic reagent (*D. melanogaster*) | *D. melanogaster.* Expresses dsRNA for RNAi of kek1 (FBgn0015399) under UAS control. | Vienna *Drosophila* Resource Center | VDRC:v101166 | |
| Genetic reagent (*D. melanogaster*) | *D. melanogaster.* Expresses dsRNA for RNAi of ths (FBgn0033652) under UAS control. | Vienna *Drosophila* Resource Center | VDRC:v102441 | |
| Genetic reagent (*D. melanogaster*) | *D. melanogaster.* Expresses dsRNA for RNAi of Ilp8 (FBgn0036690) under UAS control. | Vienna *Drosophila* Resource Center | VDRC:v102604 | |
| Genetic reagent (*D. melanogaster*) | *D. melanogaster.* Expresses dsRNA for RNAi of CG15800 (FBgn0034904) under UAS control. | Vienna *Drosophila* Resource Center | VDRC:v110049 | |
| Genetic reagent (*D. melanogaster*) | *D. melanogaster.* Expresses dsRNA for RNAi of IM3 (FBgn0040736) under UAS control. | Vienna *Drosophila* Resource Center | VDRC:v104908 | |
| Genetic reagent (*D. melanogaster*) | *D. melanogaster.* Expresses dsRNA for RNAi of Roc1a (FBgn0025638) under UAS control. | Vienna *Drosophila* Resource Center | VDRC:v106315 | |
| Genetic reagent (*D. melanogaster*) | *D. melanogaster.* Expresses dsRNA for RNAi of dod (FBgn0015379) under UAS control. | Vienna *Drosophila* Resource Center | VDRC:v110593 | |
| Genetic reagent (*D. melanogaster*) | *D. melanogaster.* Expresses dsRNA for RNAi of hipk (FBgn0035142) under UAS control. | Vienna *Drosophila* Resource Center | VDRC:v108254 | |
| Genetic reagent (*D. melanogaster*) | *D. melanogaster.* Expresses dsRNA for RNAi of ave (FBgn0050476) under UAS control. | Vienna *Drosophila* Resource Center | VDRC:v101471 | |
| Genetic reagent (*D. melanogaster*) | *D. melanogaster.* Expresses dsRNA for RNAi of boca (FBgn0004132) under UAS control. | Vienna *Drosophila* Resource Center | VDRC:v108406 | |
| Genetic reagent (*D. melanogaster*) | *D. melanogaster.* Expresses dsRNA for RNAi of gskt (FBgn0046332) under UAS control. | Vienna *Drosophila* Resource Center | VDRC:v107429 | |

*Continued*

| Reagent type (species) or resource | Designation | Source or reference | Identifiers | Additional information |
|---|---|---|---|---|
| Genetic reagent (*D. melanogaster*) | *D. melanogaster.* Expresses dsRNA for RNAi of stumps (FBgn0020299) under UAS control. | Vienna *Drosophila* Resource Center | VDRC:v105603 | |
| Genetic reagent (*D. melanogaster*) | *D. melanogaster.* Expresses dsRNA for RNAi of CG31431 (FBgn0051431) under UAS control. | Vienna *Drosophila* Resource Center | VDRC:v104697 | |
| Genetic reagent (*D. melanogaster*) | *D. melanogaster.* Expresses dsRNA for RNAi of scw (FBgn0005590) under UAS control. | Vienna *Drosophila* Resource Center | VDRC:v105303 | |
| Genetic reagent (*D. melanogaster*) | *D. melanogaster.* Expresses dsRNA for RNAi of fry (FBgn0016081) under UAS control. | Vienna *Drosophila* Resource Center | VDRC:v103569 | |
| Genetic reagent (*D. melanogaster*) | *D. melanogaster.* Expresses dsRNA for RNAi of Krn (FBgn0052179) under UAS control. | Vienna *Drosophila* Resource Center | VDRC:v104299 | |
| Genetic reagent (*D. melanogaster*) | *D. melanogaster.* Expresses dsRNA for RNAi of pxb (FBgn0053207) under UAS control. | Vienna *Drosophila* Resource Center | VDRC:v102240 | |
| Genetic reagent (*D. melanogaster*) | *D. melanogaster.* Expresses dsRNA for RNAi of cv-c (FBgn0285955) under UAS control. | Vienna *Drosophila* Resource Center | VDRC:v105435 | |
| Genetic reagent (*D. melanogaster*) | *D. melanogaster.* Expresses dsRNA for RNAi of cic (FBgn0262582) under UAS control. | Vienna *Drosophila* Resource Center | VDRC:v103805 | |
| Genetic reagent (*D. melanogaster*) | *D. melanogaster.* Expresses dsRNA for RNAi of dia (FBgn0011202) under UAS control. | Vienna *Drosophila* Resource Center | VDRC:v103914 | |
| Genetic reagent (*D. melanogaster*) | *D. melanogaster.* Expresses dsRNA for RNAi of SkpD;SkpC (FBgn0026174) under UAS control. | Vienna *Drosophila* Resource Center | VDRC:v109181 | |
| Genetic reagent (*D. melanogaster*) | *D. melanogaster.* Expresses dsRNA for RNAi of drk (FBgn0004638) under UAS control. | Vienna *Drosophila* Resource Center | VDRC:v105498 | |
| Genetic reagent (*D. melanogaster*) | *D. melanogaster.* Expresses dsRNA for RNAi of botv (FBgn0027535) under UAS control. | Vienna *Drosophila* Resource Center | VDRC:v37186 | |

## Lead contact and materials availability

This study did not generate new unique reagents. This study generated new python3 code available on GitHub: https://github.com/extavourlab/hpo_ova_eggL_screen; *Kumar, 2020* (copy archived at https://github.com/elifesciences-publications/hpo_ova_eggL_screen).

Further information and requests for resources and reagents should be directed to and will be fulfilled by the Lead Contact, Cassandra G. Extavour (extavour@oeb.harvard.edu).

## Experimental model and subject details

Wild type and mutant lines of *Drosophila melanogaster* were obtained from publicly accessible stock centers and maintained as described in 'Fly Stocks' below. Genotypes and provenance are provided in the Key Resource Table. Candidate genes were randomly assigned to batches for screening (see the *Supplementary file 1* for which genes were in each batch). F1 animals from the same cross were randomly assigned to experimental groups for phenotyping in all screens.

## Method details

### Fly stocks

Flies were reared at 25°C at 60% humidity with standard *Drosophila* food (*Sarikaya et al., 2012*) containing yeast and in uncrowded conditions as previously defined (*Sarikaya and Extavour, 2015*). RNAi lines were obtained from the TRiP RNAi collection at the Bloomington *Drosophila* Stock Centre (BDSC) and from the Vienna *Drosophila* Resource Centre (VDRC). See Key Resources Table for complete list of stocks used in this study. Oregon R was used as a wild-type strain. The genotype of the *traffic jam:Gal4* line used in the screen was *y w; P{w[+mW.hs]=GawB}NP1624* (Kyoto Stock Center, K104–055; abbreviated hereafter as *tj:Gal4*). The *hippo* RNAi line used in the screen was *y[1] v [1]; P{y[+t7.7]v[+t1.8]=TRiP. HMS00006}attP2* (BDSC:33614; abbreviated hereafter as *hpo[RNAi]*).

### Egg and ovariole number counts

Adult egg laying was quantified by crossing three virgin females of the desired genotype (see '*Screen design*' below) with two males in a vial containing standard food and yeast granules (day one) and then transferring them into a fresh food vial without yeast granules for a 24 hr period. Eggs from vials were then counted by visual inspection of the surface of the food in the vial. Males and females were transferred to fresh food vials without yeast granules, every day thereafter until day six. All egg-laying measurements reported and analysed in the paper are the sum of the eggs laid by three adult female flies over the five days of this assay (days two through six without yeast granules). Data from any vial in which either a female or male died, during the course of the experiment, were not included in the analysis.

Ovariole number was quantified by mating ten virgin adult females with five virgin adult Oregon R males for three days post-eclosion in vials with yeast at 25°C and 60% humidity. After this three-day mating period, all 20 adult ovaries from the mated females were dissected in 1X PBS with 0.1% Triton-X-100 and stained with 1 μg/ml Hoechst 33321 (1:10,000 of a 10 mg/ml stock solution). Ovarioles were separated from each other with No. 5 forceps (Fine Science Tools) and counted by counting the number of germaria under a ZEISS Stemi 305 compact stereo microscope with a NIGHTSEA stereo microscope UV Fluorescence adaptor.

### Screen design

In the primary screen (*Figure 1a*: *hpo[RNAi]* Egg Laying), 463 candidate genes (*Supplementary file 1*) were screened for the effect of an RNAi-induced loss of gene function in a *hpo[RNAi]* background on the number of eggs laid in the first five days of mating (see '*Egg and ovariole number counts*' above) by adult females. These females were the F1 offspring of UAS:*candidate gene* RNAi males crossed to *P{w[+mW.hs]=GawB}NP1624; P{y[+t7.7] v[+t1.8]=TRiP.HMS00006}attP2* (*tj:Gal4; UAS: hpo[RNAi]*) virgin adult females (*Figure 1a*: *hpo[RNAi]* Egg Laying). All genes that yielded an egg laying count with a $|Z_{gene}| > 1$ (see '*Gene selection based on Z score and batch standardisation*' below) were selected to undergo two secondary screenings (n = 273, *Table 2*; *Figure 1d*). First, these genes were screened for effects on the egg laying of mated adult female offspring from a cross of *UAS:candidate gene[RNAi]* males and *tj:Gal4* virgin adult females (*Figure 1b*: Egg Laying). Secondly, these genes were screened for effects on ovariole number in a *hpo[RNAi]* background. All 20 ovaries from ten adult female F1 offspring of a cross between *UAS:candidate gene[RNAi]* males to *P{w[+mW.hs]=GawB}NP1624; P{y[+t7.7] v[+t1.8]=TRiP.HMS00006}attP2* (*tj:Gal4; UAS:hpo[RNAi]*) virgin adult females were scored for ovariole number (see '*Egg and ovariole number counts*' above). (*Figure 1c*: *hpo[RNAi]* Ovariole Number).

## Gene selection based on Z score and batch standardisation

Candidate genes were screened in batches with an average size of 50 genes. For each batch, control flies were the female F1 offspring of Oregon R males crossed to *P{w[+mW.hs]=GawB}NP1624; P{y [+t7.7] v[+t1.8]=TRiP.HMS00006}attP2* (*tj:Gal4; UAS:hpo[RNAi]*) virgin adult females. Because the control group in each batch had slightly different distributions of egg laying and ovariole number values (*Figure 1—figure supplement 1*), it was inappropriate to compare absolute mean values between genes that were scored in different batches. Instead, comparisons of the Z score of each candidate ($Z_{gene}$) to its batch control group was used as a discriminant. This approach standardises for batch effects and allows the comparison of all genotypes within and across the primary and secondary screens with a single metric ($Z_{gene}$).

Firstly, the mean and standard deviation of the eggs laid by the control genotype for a batch were calculated as $\mu_b$ and $\sigma_b$ respectively. Then, using the number of eggs laid by adult females of a candidate gene RNAi ($x_{gene}$) of the same batch, the Z score for the egg laying count of that gene ($Z_{gene}$) was calculated as $Z_{gene} = \frac{x_{gene} - \mu_b}{\sigma_b}$. The same standardisation protocol was applied to both egg laying and ovariole number counts of every gene and its corresponding batch control.

Ovariole numbers were derived from counts of the number of ovarioles per ovary for 20 ovaries per candidate gene, and a threshold of $|Z_{gene}| > 2$ was applied for ovariole number phenotype. Egg laying counts were derived from measurements of three females in a single vial per gene. We therefore chose to be more conservative in our Z score comparisons for the egg laying phenotype, than for ovariole number phenotype, and applied a stringent threshold of $|Z_{gene}| > 5$ to select genes of interest. All genes with $|Z_{gene}|$ values above these thresholds are referred to throughout the study as 'positive candidates'. (See Ipython notebooks 02_Z_score_calculation.ipynb and 02.2_Z_score_calculation_prediction.ipynb for code implementation and calculation of Z scores, and 06_Screen Analysis.ipynb for batch effects.)

## Signalling pathway enrichment analysis

To study the enrichment of a particular signalling pathway in a group of candidate genes that had similar phenotypic effects revealed by the screen, custom scripts (see 07_Signaling_pathway_analysis.ipynb for code implementation) were generated to implement two different methods (*Figure 3a and b*; *Figure 3—figure supplement 1*; *Figure 5—figure supplement 6a,b*).

The first method is a numerical method that uses random sampling to calculate the null distribution of the number of members (M) of a signalling pathway (S) that would be expected at random in a set of genes of size (N). The script randomly sampled N genes from among the 463 tested *D. melanogaster* signalling genes 10,000 times, and counted the number of genes (M) that were members of the signalling pathway S. Positive candidates in each of the three screens were sorted by their presence in signalling pathways and counted. The Z score was then calculated by comparing the experimentally observed number of positive candidates in each signalling pathway against the randomly sampled null distribution.

The second method used the hypergeometric p-value to calculate the probability of M members of a signalling pathway being in a group of N genes, given a starting population of 463 tested *D. melanogaster* signalling genes, and the known attribution to a pathway S of each gene.

## Protein-Protein Interaction Network (PIN) building

There is no standard complete Protein-Protein Interaction network (PIN) available for *Drosophila melanogaster*. However, there exist many smaller networks from different screens, as well as literature extractions. We therefore combined data from these sources and then created a PIN for use in the present study, as follows:

## Step 1

Several screens assessing protein-protein interactions have been centralised in a database called DroID: http://www.droidb.org. The version DroID_v2018_08 was used. All available datasets were first downloaded from that database using this link: http://www.droidb.org/Downloads.jsp. The description of all of these datasets can be found here: http://www.droidb.org/DBdescription.jsp.

## Step 2

We used the datasets from all screens that assessed direct protein-protein interactions and did not use the interolog database (predicted protein interaction based on mouse human and yeast PPIs). These direct assessment screens were seven in total, as follows:

Finley Yeast Two-Hybrid Data (size 2.0 MB; 3610 Nodes and 9007 Edges)
Curagen Yeast Two-Hybrid Data (size 4.6 MB; 6678 Nodes and 19506 Edges)
Hybrigenics Yeast Two-Hybrid Data (size 381 KB; 1269 Nodes & 1842 Edges)
Perrimon co-AP complex (size 108 KB; 252 Nodes and 384 Edges)
DPiM co-AP complex (size 6.3 MB; 3732 Nodes and 17652 Edges)
PPI from other databases (size 16.2 MB; 7524 Nodes and 47471 Edges)
PPI curated by FlyBase (size 7.4 MB; 5125 Nodes and 31491 Edges)

We did not consider self-loop edges from proteins predicted to interact with themselves (homotypic or self-interactions). An important element to note is that the PPIs curated by FlyBase is a literature-based PPIs. FlyBase protein-protein interactions are experimentally derived physical interactions curated from the literature by FlyBase and does not include FlyBase-curated genetic interactions.

## Step 3

We concatenated the seven datasets listed above into a single unique database. A custom python script was created that downloads and reads each of the above seven unique PPI tables, and generates a single PIN (see 01_PIN_builder.ipynb). From this concatenation, a single-edge undirected network was created and saved. This network is hereafter referred to as the PIN (see 01_PIN_builder.ipynb). The PIN contains 10,632 proteins (nodes) and 85,019 interactions (edges), giving a network density of 0.0015.

## Network metric computations

The centrality of a node is often used as a measure of a node's importance in a network. Within a PIN, the centrality of a gene reflects the number of interactions in which the gene directly or indirectly participates. Four different centrality metrics were computed for all genes in the PIN using the networkx python library:

1. Betweenness reflects the number of shortest paths passing through a gene.
2. Eigenvector is a measure of the influence of a gene in the network.
3. Closeness measures the sum of shortest distance of a gene to all the other genes.
4. Degree centrality corresponds to the normalised number of edges of a gene in the network.

While there exist more centrality measures, these four are commonly used to assess biological networks. These computed centrality parameters of the genes measured in the screen were computed with 03_ROC_curve_analysis_of_network_metrics.ipynb, and are reported in the *Supplementary file 1* (see 09_Making_the_database_table.ipynb).

## Receiver Operating Characteristic (ROC) curves

To check whether the centrality of a gene in the network could predict the phenotypic effect produced by RNAi against that gene, ROC curves were plotted for the four aforementioned centrality measures of each gene in each screen. A ROC analysis is used to measure the correlation between a continuous variable (centrality) and a binary outcome (above or below Z score threshold). Therefore, for each screen, measured genes were rank ordered from high centrality to low centrality, and plotted against the binary outcome of $|Z_{gene}|$ being above or below the appropriate |Z score| threshold (>5 for egg laying and >2 for ovariole number). The Area Under the Curve (AUC) measures the extent of correlation between centrality and effect of a gene on measured phenotype. AUC above or below 0.5 indicates a positive or negative correlation respectively, while an AUC of 0.5 indicates no correlation of the parameters. The scikit-learn python package was used to calculate the AUC of each ROC curve plotted (see 03_ROC_curve_analysis_of_network_metrics.ipynb).

## Building degree-controlled randomised networks

We assessed the modularity of the networks by comparing the network metrics of each sub-network to a degree-controlled randomly sampled network. To generate this degree controlled random network, we applied a previously developed method (*Guney et al., 2016*). In short, nodes in the PPI are binned by degree with the minimum size of each bin being set at 100 nodes. Bins are constructed iteratively from the lowest degree to the highest degree in the network. To sample a set of nodes, the sub-network degree distribution is computed, using the bin cut-off, from the PPI. Then, nodes are randomly selected from each bin to match this degree distribution (see 05.2_Degree_-Controlled_Testing.ipynb for code implementation).

## Assessing the utility of the Seed Connector Algorithm in building network modules

Network modules were assessed using the previously published Seed Connector Algorithm (SCA) (*Wang et al., 2017*; *Wang and Loscalzo, 2018*), implemented here in python (see 04_Seed-Connector.ipynb) and illustrated in *Figure 5a*. Creating a module using the SCA requires a list of seed genes and a PIN. From each of the three screens, we selected the genes whose $|Z_{gene}|$ value was above the threshold and created three seed lists, respectively (*Figure 4c*: Egg laying, *hpo[RNAi]* egg laying and *hpo[RNAi]* ovariole 'seed' list). A fourth list consisting of the intersection of the aforementioned seed lists was also collated and called the core 'seed' list (*Figure 4b*). Genes were assigned in the core list if they passed the Z threshold in all three screens. The SCA was then executed on each of these seed lists using the PIN. Not all genes in the four seed lists were found in the PIN (specifically, CG12147 in the *hpo[RNAi]* Egg Laying seed list and CG6104 in the *hpo[RNAi]* Ovariole number seed list were absent from the PIN) and were therefore eliminated from further network analysis. The removal of these two genes accounts for the variation in the number of positive candidates in *Table 2* and the number of seed genes in the module. Modules were obtained for each seed list (*Figure 5b*; *Figure 5—figure supplements 1–3*) consisting of the seed genes (circles in *Figure 5b* and *Figure 5—figure supplements 1–3*) and previously untested genes added by the SCA (squares in *Figure 5b* and *Figure 5—figure supplements 1–3*) to increase the LCC size that we refer to as connector genes (see 04_Seed-Connector.ipynb). The results of the algorithm are summarised in the *Supplementary file 1*.

The modularity of the sub-networks was then assessed using four network metrics namely Largest Connected Component (LCC), number of edges, network density and average shortest path in the LCC. Each metric for each module was assessed using distance of the network metric to a null distribution. Initially, the null distribution was calculated by taking 1000 samples of 463 genes randomly selected from the PIN and calculating the above metrics. We found that the 463 genes selected in the signalling screen were already more connected than the null distribution of sets of 463 genes randomly selected from the PIN (*Figure 5—figure supplement 4a*). Therefore, to avoid a false positive detection of modularity, the four experimentally obtained sub-networks were compared to null distributions obtained by randomly sampling an equal number of genes from the 463 signalling candidate genes selected for our screen. For each of the four modules, comparison of the metrics was performed on the seed lists and the sub-network after the SCA. Most metrics were enriched in the seed group when compared to the null distribution with the exception of the Average shortest path (*Figure 5—figure supplement 4b*, light red line). The sub-networks obtained from the SCA further increased all four metrics suggesting the modularity of the four sub-networks (*Figure 5—figure supplement 4b*, dark red line; see 05_Network_Module_testing.ipynb for code implementation).

## Meta network

To build the meta network, the genes from all four sub-networks were concatenated into one network. This network was then visually sorted in an approach akin to projecting the network onto a Venn Diagram. The meta network was sorted by which of the three screens the gene was positive in. The intersections were genes whose $|Z_{gene}|$ value was above the threshold in more than one and possibly all three of the screening paradigms. For example, if a gene was found in the *hpo[RNAi]* Ovariole Number and Egg Laying sub-networks it is then assigned to the dual positive group *hpo[RNAi]* Ovariole Number/Egg Laying (*Figure 6a*, sub-network VI). After applying this grouping strategy, the connectivity across the groups was studied by calculating the edge density between all groups

($density = \frac{Edges_{1:2}}{Nodes_1 * Nodes_2}$). Finally, the proportion of each signalling candidate in each of those groups was calculated by taking the number of members of a signalling pathway divided by the total members of a group (see Ipython notebook 08_MetaModule_Analysis.ipynb). A single gene, *sloppy paired 1*, was a seed in the Egg Laying sub-network and also a connector in the *hpo[RNAi]* Egg Laying sub-network; it fell within sub-network VII in the meta network, and is marked as a seed (grey) in *Figure 6a*.

## Quantification and statistical analysis

### Number of samples
The number of samples across the different screens were as follows:

> *Hpo[RNAi]* Egg Laying and Egg Laying screens
> > Controls: five vials of three females and two males
> > Sample: one vial of three females and two males
>
> *Hpo[RNAi]* Ovariole number screen
> > Controls: 20 flies, two ovaries per fly considered as independent measurements
> > Sample: 10 flies, two ovaries per fly considered as independent measurements

### Correction of batch effect
Despite best efforts to maintain the exact same condition between each experiment, some variation was measured between the batches. Control flies showed variations in both measured phenotypes, ovariole number and egg laying (*Figure 1—figure supplement 1*). In order to compare the values measured across different batches, each sample was standardised by calculating its Z score ($Z_{gene}$) to the control distribution. For each batch, the measurements for controls were pooled into a distribution, and the mean and standard deviation was computed. Then each sample was compared to its respective batch and its Z score computed (see '*Gene selection based on Z score and batch standardisation*' for formula).

### Statistical analysis
All statistical analyses were performed using the scipy stats module (https://www.scipy.org/) and scikit-learn (https://scikit-learn.org/) python package. Statistical tests and p-values are reported in the figure legends. All statistical tests can be found in the Ipython notebooks mentioned below.

## Data and code availability
This study generated a series of python3 Ipython notebook files that perform the entire analysis presented in this study. All the results presented in this paper, including the figures with the exception of the network visualisations, which were created using Cytoscape3 (https://cytoscape.org/) can be reproduced by running the aforementioned python3 code. The raw data, calculations made with these data, and code used for calculations and analyses (Ipython notebooks) are available as supplementary information. For ease of access, legibility and reproducibility, the code and datasets have been deposited in a GitHub repository available at https://github.com/extavourlab/hpo_ova_eggL_screen.

## Additional information

### Funding

| Funder | Grant reference number | Author |
| --- | --- | --- |
| National Institutes of Health | 1R01-HD073499 | Tarun Kumar<br>Cassandra G Extavour |

The funders had no role in study design, data collection and interpretation, or the decision to submit the work for publication.

## Author contributions

Tarun Kumar, Conceptualization, Data curation, Formal analysis, Supervision, Validation, Investigation, Visualization, Methodology, Writing - original draft, Project administration, Writing - review and editing; Leo Blondel, Conceptualization, Data curation, Software, Formal analysis, Validation, Visualization, Methodology, Writing - original draft, Writing - review and editing; Cassandra G Extavour, Conceptualization, Supervision, Funding acquisition, Methodology, Project administration, Writing - review and editing

## Author ORCIDs

Tarun Kumar ⬡ https://orcid.org/0000-0003-4071-4342
Leo Blondel ⬡ https://orcid.org/0000-0003-2276-4821
Cassandra G Extavour ⬡ https://orcid.org/0000-0003-2922-5855

## Decision letter and Author response

Decision letter https://doi.org/10.7554/eLife.54082.sa1
Author response https://doi.org/10.7554/eLife.54082.sa2

---

## Additional files

### Supplementary files

• Supplementary file 1. Tabulation of raw data and analysis for every gene in the screen. This table contains a summary representation of the data generated by the three screens as well as results from the analysis. Each line corresponds to an independent measurement of a particular RNAi line. Some genes which did not pass the first filter of $|Z_{gene}| > 1$ in the *hpo[RNAi]* Egg Laying screen where then predicted as connectors, therefore they have two entries as they have been independently measured again. The Z scores have been rounded up to 4 significant digits in this table and the Centrality metrics rounded up to 10 significant digits due to their low values, but the full values for both are available in the raw data files provided in the supplementary files in Data/Screens for the Z scores and Results for the centrality values. Moreover, this is a summary table and does not contain values for controls as well as batch numbers, all are available in the supplementary files in Data/Screens. - **FbID**: FlyBase ID of the tested gene; - **CG number**: CG Number of the tested gene; - **NAME**: Common name (as per FlyBase nomenclature) of the gene if existing, else it is a **"-"**; - **SYMBOL**: Symbol (as per FlyBase nomenclature) of the gene if it exists, else it is its CG number; - **[*ScreenName*]_[*Variable*]_(*Metric*)_Count**: Within the screen [*ScreenName*], the count of the measured variable [*Variable*]. Optional: (*metric*) will indicate if a particular operation was done over the data, such as sum, mean or standard deviation. e.g. [HippoRNAi_EggL]_[Day_4_Egg]_Count is the count of eggs, on day 4, of the *hpo[RNAi]* Egg Laying screen; - **[*ScreenName*]_[*Variable*]_(*Metric*)_Zscore**: Within the screen [*ScreenName*], the Z score of the measured variable [Variable] as calculated to batch control. Optional: (*metric*) will indicate if a particular operation was done over the data, such as sum, mean or standard deviation. e.g. [EggL]_[All_Days_Egg]_(Sum)_Zscore is the Z score of the sum of eggs count, of the Egg Laying screen; - **PIN_[*Metric*]_centrality**: Within the PIN used in this study, the calculated centrality value for the metric [*Metric*]; - **[*SubNetworkName*]_Network**: Presence of absence of a gene in the sub-network [*SubNetworkName*]. If the gene is in the module, this value is True, if it is absent it is False. (An exception is made for the Meta Network displayed in Figure 6 where instead of True/False, the group assignment I-VII is written); - **[*SubNetworkName*]_Connector**: Status of a gene in the sub-network [*SubNetworkName*] as a connector. If True, the gene is a connector, else if False, the gene is not a connector; - **[*PathwayName*]_Pathway**: Participation of a gene to the signalling pathway [*PathwayName*]. If the gene participates in the pathway the value is 1, else it is 0.

• Transparent reporting form

## Data availability

This study did not generate new unique reagents. This study generated new python3 code available on GitHub: https://github.com/extavourlab/hpo_ova_eggL_screen (copy archived at https://github.com/elifesciences-publications/hpo_ova_eggL_screen).

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
