## [Decision Letter]

**Acceptance summary:**

This paper is amongst the first to show how computational network analyses based on protein-protein interactions can be successfully used to augment genetic screens to identify genes involved in essential developmental processes.

**Decision letter after peer review:**

Thank you for submitting your article "Topology-driven protein-protein interaction network analysis detects genetic modules regulating reproductive capacity" for consideration by *eLife*. Your article has been reviewed by three peer reviewers, and the evaluation has been overseen by Michael Eisen as the Senior and Reviewing Editor The following individuals involved in review of your submission have agreed to reveal their identity: Felipe Aguilera (Reviewer #2); Luke Lambourne (Reviewer #3).

The reviewers have discussed the reviews with one another and the Reviewing Editor has drafted this decision to help you prepare a revised submission.

We would like to draw your attention to changes in our revision policy that we have made in response to COVID-19 (https://elifesciences.org/articles/57162). Specifically, we are asking editors to proceed without manuscripts, like yours, that they judge can stand as *eLife* papers without additional data, even if they feel that they would make the manuscript stronger. Thus the revisions requested below only address clarity and presentation.

This work aims to predict protein-protein interaction in a developmental context with a combination of approaches drawn from developmental genetics and system biology. This work represents an elegant study that combines computational biology and molecular developmental biology to explain how organ systems are formed.

Although it is our usual practice to combine the reviews into a single consensus, in this case I think the independent reviews offer distinct takes on the manuscript and suggestions for revisions and stand better on their own, although to expedite your revisions, I will summarize the main issues to address here:

Analysis:

Several issues were raised by reviewer 3 and during discussion:

1) Additional analysis is needed to support claim of non-random modules

In particular, the results of Figure 4 show that the positive pairs in the screen on average, tend to have higher degree in the network, which could be responsible for the higher connectivity of the seed genes, rather than them forming modules. This should be addressed with some form of control analysis.

2) Module separation

As a general rule, it is not surprising that the connectivity is higher within the output of the SCA algorithm, which returns a subnetwork of connected nodes, to the connectivity between two different SCA produced sub-networks. This is expected to happen with random lists of seed genes, and does not prove that functionally separate sub-networks have been identified. The claim is confusing given that there are shared subnetworks (V, VI, VII) which seems to contradict the idea of separate modules. Reviewer 3 offers several suggestions for how to make this analysis more convincing.

Writing and presentation:

1) In addition to suggestions about presentation in the reviews below, during discussion all three reviewers highlighted the value of the overall approach and felt that the manuscript merits publication in *eLife*. However there is a sense that the manuscripts lacks a clear take home message. This is not required for publication per se, but we all feel the manuscript could be made stronger in revisions by attempting to distill a take home message for readers.

2) The experimental validation results are not presented clearly. That section could benefit from more clarity in what specific aspects of the predictions are being tested and how the specific tests address them.

Reviewer #1:

In this study, the authors investigate the signaling pathways that regulate *Drosophila* ovary development and the rate of egg production. An earlier study from this lab identified a role for hippo signaling in these processes. Here, they performed an RNAi-based screen through 463 candidate genes involved in signal transduction to identify genetic modifiers of the hippo[RNAi] phenotype as well as genes that regulate the rate of egg laying through a potentially hippo-independent mechanism. The screens were well-designed and produced quantitative data that the authors could use to rank-order genes according to the strength and direction of the phenotype. Their strategy for assessing the rate of egg laying is in line with standard approaches in the field and their method of counting ovariole number is entirely appropriate. These screens identified a long list of genes, and they took a systems biology approach to assess the individual and combinatorial contributions of these genes to ovary development and function. These types of analyses are outside my area of expertise so I could not assess whether the appropriate computational methods were chosen and applied correctly. However, the writing of this section was clear to a non-expert and, taking their claims at face value, the data seem to support their interpretations.

In the final section of the manuscript they describe how their network analysis identified additional genes not uncovered by their screens and show that RNAi knockdown of these additional genes do, indeed, cause phenotypes in egg laying rates and ovariole number. This is a nice validation of the approach. As a minor point, the data presented in Figure 7 only summarize how many genes in the list showed a phenotype in each category but did not indicate what the genes are, which ones are associated with which phenotype, or what the z-scores were in each case (I could not find this information in the supplemental material either, though perhaps I missed it). If it has not been included already, the authors should provide this for completeness and to allow comparison to the other data in the study.

Overall, I think this study will be a useful though perhaps somewhat limited contribution to the field. The results of the screens provide a large amount of data for the field to build on and I appreciate their holistic approach to describing the interplay between genes and pathways in the regulation of processes as complex as development and oogenesis. At the same time, I struggled a bit to extract the most important take-away messages that change how I think about these processes. For example, they state in the Discussion that the core module consists of housekeeping genes and hedgehog pathway genes, but this could have largely been predicted from previous publications on this topic. Likewise, terminal filament formation is known to involve cell migration and, consistent with this, they find that several genes involved in cytoskeletal dynamics are important for proper specification of ovariole number. Though some of these genes have not been studied in this context, their involvement here is not too surprising. Nonetheless, it may be that the main innovation of this study is in the systems biology approach they have used to analyze the data, so I look forward to the online discussion with the other reviewers who have more expertise in this area.

Reviewer #2:

A general assessment of the work

This manuscript aims to predict protein-protein interaction in a developmental context. By using tissue-specific RNAi screening and systems biology approaches, the authors found known and unknown genes involved in ovarian development and function. A plethora of unknown genes was functionally tested, giving support to the topology-driven network analysis conducted, and the developmental regulatory modules found. This work represents an elegant study that combines computational biology and molecular developmental biology to explain how organ systems are formed.

Numbered summary of any substantive concerns

This article is generally well-structured, the data analysis is performed in a proper fashion, and the results are nicely presented, giving support to the major conclusions. I have no major concerns with regard to this manuscript, only some minor edits.

1) I am wondering if authors can give more detail regarding the paragraph starting "To choose candidates.…" I might be wrong, but have the feeling that this approach might produce bias in the results obtained by the authors.

2) In Figure 4—figure supplement 2, the authors show a comparison of Z_gene_ scores of positive candidate genes sorted by centrality metrics. However, I wonder why the authors did not calculate p-values (significance level) between 1st and 5th quantile comparisons of Z_gene_ scores shown in this figure? Significance levels in these comparisons might give strong support to the weak prediction of a gene that would affect a specific phenotype.

3) In Figure 6, for the sake of clarity, it is necessary to show the sloppy paired 1 gene in the meta-network analysis (Figure 6A). As is now shown in this figure, readers cannot see easily the importance of this gene within sub-network VII, which I think is one point that the authors want to highlight.

4) In step 3 in the protein-protein interaction network (PPI) building section (“Method Details” subsection), authors indicate that a custom python script was created to download and reads each of the PPI tables from DroID database, but I could not find this python script in the Github repository in the paper. Please, can authors upload this script in the repository?

Reviewer #3:

The authors present a novel study integrating the results of performing screens for regulators of reproduction in *Drosophila* with protein-protein interaction networks. Highly interesting, interdisciplinary work lying at the intersection of developmental and network biology. They identify genes regulating ovariole number and egg laying and analyze those genes in the context of PPI networks, predicting and experimentally testing additional novel reproductive regulatory genes with some success. However there are serious issues with the analysis supporting key claims in the manuscript related to network topology.

1) The claim of non-random modules is not supported by the analysis.

– The claim in the section title "Genes regulating egg laying and ovariole number regulation form non-random interaction modules" (L260-L261) is supported by Figure 5—figure supplement 4. In that figure both the seed genes and results from the SCA algorithm are compared to the subnetworks formed by randomly selected genes from the initial screen. This is a sensible comparison for the seed genes but not for the output of the SCA algorithm, since that algorithm builds a LCC (see Figure 5A) and so will obviously have a much larger LCC and be much more connected than randomly selected genes from the network.

– The results of Figure 4 show that the positive pairs in the screen on average, tend to have higher degree in the network, which could be responsible for the higher connectivity of the seed genes, rather than them forming modules. To account for this, the seed genes could be compared to a null distribution of degree-controlled randomized networks.

– The text reads "decreased average shorted path" (L331) but it is increased in 7 of 8 cases in the figure.

2) The claim of separate modules is not supported by the analysis.

– Similarly to point 1, it is not surprising that the connectivity is higher within the output of the SCA algorithm, which returns a subnetwork of connected nodes, to the connectivity between two different SCA produced sub-networks, that will happen with random lists of seed genes, it does not prove that functionally separate sub-networks have been identified.

– There are sizable shared subnetworks (V, VI, VII) which seems to contradict the idea of separate modules.

– The heat-map Figure 6B, shows very low edge density within the groups (diagonal squares), in contradiction to what is written in the text. This appears to be due to a miscalculation in the file 08_MetaModule_Analysis.ipynb in the Github repository: within the group the number of edges should be divided by (x^2^ – x) / 2 where x is the number of nodes in the group (assuming self-interactions are not considered), the edge density calculations between the different groups are correct.

– An alternative approach to this analysis could be to look at the connectivity of only the seed genes and not the connector genes in the different groups and comparing to a random assignment of group.

– Identifying modules of genes which affect one phenotype and not another is presumably made more difficult by the decision to filter candidate genes for two of the phenotypic screens based on the results of the first screen.

– The prediction results do not appear to be consistent with separate modules (see point 3 below).

3) The results of experimentally testing the predictions are presented in an unclear way, making it difficult for the reader to assess their accuracy.

The results in Figure 7A/B/C appear to show that the predictions from the modules are just as good for the phenotypes not associated with the module as for those associated with the module, e.g. for the EL phenotype there is one correct prediction from both the Core and EL modules and 9 correct predictions from the hpo[RNAi] ON/EL modules, with a similar number of predictions. So it would seem that these results suggest that, although there is positive predictive value overall relative to the initial signaling genes list, that the separate modules predicting their associated phenotype separately from the others is not supported by the data. This is not discussed in the text.

– From what I could understand, the fractions for the initial signaling gene lists for the EL and hpo[RNAi] ON phenotypes are not exactly consistent with the compared fractions of the predictions, due to the filtering applied on the hpo[RNAi] of |Z| > 1, for the initial screens. Perhaps the fractions for the 2 affected bars in Figure 7C could also be calculated with this threshold applied to remove genes?

– There are no p-values calculated, comparing the predictions to the original screens and no error bars shown in Figure 7.

– The title of Figure 7C could be changed to make it clearer that this contains predictions from all modules.

– There is no “All screens” column in either Figure 7C or Figure 7D for comparison.

---

## [Author Response]

1) Although it is our usual practice to combine the reviews into a single consensus, in this case I think the independent reviews offer distinct takes on the manuscript and suggestions for revisions and stand better on their own, although to expedite your revisions, I will summarize the main issues to address here:2) Analysis:Several issues were raised by reviewer 3 and during discussion:(2.1) Additional analysis is needed to support claim of non-random modulesIn particular, the results of Figure 4 show that the positive pairs in the screen on average, tend to have higher degree in the network, which could be responsible for the higher connectivity of the seed genes, rather than them forming modules. This should be addressed with some form of control analysis.

We understand the concerns of the reviewer and we have modified our analysis to better support the claim of putative non-random modules. Firstly, we performed a new analysis, comparing the network metrics of the seed genes, to two different control groups: (a) a random selection of the same number of genes, chosen from among the signalling candidates, and (b) a degree-controlled random network of the same number of genes (new Figure 4—figure supplement 3). Our results indicate that in most cases, the seed gene lists display higher network metrics than both null distributions. We therefore interpret this as evidence that the seed genes may constitute a non-random putative interaction module, independent of application of the SCA.

On application of the SCA, we find that there is reduction of the difference in network metrics when compared to the two null distributions described above, of similarly sized seed lists selected from either (a) the signalling genes (light gray curve in new Figure 5—figure supplement 5) or (b) a degree-controlled seed list (dark grey curve in new Figure 5—figure supplement 5). We interpret this result to mean that though the SCA is able to predict genes that may function in the network, its application does not increase the modularity of the sub-network in this setting. We believe that this may also explain why genes predicted by the SCA are more likely to affect any of the three phenotypes measured, rather than the specific subnetwork in which they were predicted. We present and discuss these new control analyses in the revised manuscript paragraph six subsection “Genes regulating egg laying and ovariole number regulation form non-random gene interaction networks” and subsection “Building degree-controlled randomized networks”, and in the new figure supplements Figure 4—figure supplement 3 and Figure 5—figure supplement 5. We also qualify our description of the possible relationship between genes in the sub-networks under study, by calling them “putative modules” only when we describe them as predicted by the SCA, and subsequently as “sub-networks” throughout the revised manuscript. This comment is addressed in more detail in response to reviewer 3’s comments 2.1 and 2.2.

(2.2) Module separationAs a general rule, it is not surprising that the connectivity is higher within the output of the SCA algorithm, which returns a subnetwork of connected nodes, to the connectivity between two different SCA produced sub-networks. This is expected to happen with random lists of seed genes, and does not prove that functionally separate sub-networks have been identified. The claim is confusing given that there are shared subnetworks (V, VI, VII) which seems to contradict the idea of separate modules. Reviewer 3 offers several suggestions for how to make this analysis more convincing.

We understand the reviewers' point, and have addressed this in two major ways. First, we performed the suggested control analyses as described in response to comment 1 above, and in light of those results, are more conservative in our discussion and interpretation of the “connectedness” of the genes in phenotypically linked sub-networks. Second, we have revised the text to better convey what we meant in the original manuscript by “separate modules,” and now explain that one interpretation of the data is that genes in such sub-networks (as we now call them) may potentially function rather independently of other sub-networks, in regulating a given phenotype, or pairs of phenotypes We have more detailed explanations of this point in response to reviewer 3's comment 3.1, 3.2 and 3.6.

(3) Writing and presentation:(3.1) In addition to suggestions about presentation in the reviews below, during discussion all three reviewers highlighted the value of the overall approach and felt that the manuscript merits publication in eLife. However there is a sense that the manuscripts lacks a clear take home message. This is not required for publication per se, but we all feel the manuscript could be made stronger in revisions by attempting to distill a take home message for readers.

We take this point, and in response have modified the Discussion section to better present what we believe are the most important take home messages of the paper. We have addressed this in further detail in response to reviewer 1’s comment 4.

(3.2) The experimental validation results are not presented clearly. That section could benefit from more clarity in what specific aspects of the predictions are being tested and how the specific tests address them.

To address this concern, we have made significant modifications to the presentation of experimental validation results. We have modified original Figure 7C as per the advice of the reviewer, and merged it with original Figure 7D to create a new Figure 7C, which we believe provides a better understanding of our results (discussed in detail in response to reviewer 3’s comment 4.2 ). We have also modified the description of Figure 7 in the revised results (subsection “Network analysis predicts novel genes involved in egg laying and ovariole number”; discussed in detail in response to reviewer 3’s comment 4.4) and Discussion (subsection “Network analysis as a tool in developmental biology”; discussed in detail in response to reviewer 3’s comment 4.1).

Reviewer #1:(1) In this study, the authors investigate the signaling pathways that regulate *DrosophilaDrosophila* ovary development and the rate of egg production. An earlier study from this lab identified a role for hippo signaling in these processes. Here, they performed an RNAi-based screen through 463 candidate genes involved in signal transduction to identify genetic modifiers of the hippo[RNAi] phenotype as well as genes that regulate the rate of egg laying through a potentially hippo-independent mechanism. The screens were well-designed and produced quantitative data that the authors could use to rank-order genes according to the strength and direction of the phenotype. Their strategy for assessing the rate of egg laying is in line with standard approaches in the field and their method of counting ovariole number is entirely appropriate. These screens identified a long list of genes, and they took a systems biology approach to assess the individual and combinatorial contributions of these genes to ovary development and function. These types of analyses are outside my area of expertise so I could not assess whether the appropriate computational methods were chosen and applied correctly. However, the writing of this section was clear to a non-expert and, taking their claims at face value, the data seem to support their interpretations.(2) In the final section of the manuscript they describe how their network analysis identified additional genes not uncovered by their screens and show that RNAi knockdown of these additional genes do, indeed, cause phenotypes in egg laying rates and ovariole number. This is a nice validation of the approach. As a minor point, the data presented in Figure 7 only summarize how many genes in the list showed a phenotype in each category but did not indicate what the genes are, which ones are associated with which phenotype, or what the z-scores were in each case (I could not find this information in the supplemental material either, though perhaps I missed it). If it has not been included already, the authors should provide this for completeness and to allow comparison to the other data in the study.

We appreciate the comments of the reviewer, which made us realize that we should emphasize more clearly where to find this information within the manuscript. The novel genes (i.e. connector genes) tabulated in original Figures 7, Figure 7—figure supplement 1 and Figure 7—figure supplement 2, are indicated as triangles with the gene names shown beside them, in each of the four putative modules generated by the SCA, shown in original Figure 5B and Figure 5—figure supplement 1-3. The genes are also listed in Table 4 and can be identified under the column header [ModuleName]_Connector. To highlight this for the reader, we have added new text to the revised legend of Figure 7 as follows: “The connector genes are listed in Table 4, and can be identified under the column header [ModuleName]_Connector. The raw data for egg laying and ovariole number for each of the connector genes can be found within Table 4.”.

(3) Overall, I think this study will be a useful though perhaps somewhat limited contribution to the field. The results of the screens provide a large amount of data for the field to build on and I appreciate their holistic approach to describing the interplay between genes and pathways in the regulation of processes as complex as development and oogenesis.

We thank the reviewer for this comment. The aim of the project was to indeed identify more genes that affect the development of the ovaries, both to advance the field of ovary development and also to provide an example of how such large datasets may be better used to enhance our understanding of complex developmental processes.

(4) At the same time, I struggled a bit to extract the most important take-away messages that change how I think about these processes. For example, they state in the discussion that the core module consists of housekeeping genes and hedgehog pathway genes, but this could have largely been predicted from previous publications on this topic. Likewise, terminal filament formation is known to involve cell migration and, consistent with this, they find that several genes involved in cytoskeletal dynamics are important for proper specification of ovariole number. Though some of these genes have not been studied in this context, their involvement here is not too surprising. Nonetheless, it may be that the main innovation of this study is in the systems biology approach they have used to analyze the data, so I look forward to the online discussion with the other reviewers who have more expertise in this area.

We appreciate this point, and in response have added the following text to the beginning of the revised Discussion to help clarify the take home messages of the work, as follows: “In this study, we have identified many novel genes that regulate either or both of egg laying and ovariole number. Though the development of the insect ovary has been studied for over 100 years, our understanding of the genetic mechanisms that regulate the development of the ovary is sparse. The female reproductive system and its ability to produce eggs are one of the key determinants for the survival of a species in an ecological niche. The genes we have uncovered here are possible targets for the regulation of the construction and function of the reproductive system in *Drosophila melanogaster* , and potentially in other species of insects as well. Understanding the gene regulatory networks that regulate egg laying and ovariole development could provide a framework to understand the key regulatory steps during this process that may be modified over evolutionary time, to yield the wide diversity of ovariole numbers and fecundities displayed by extant insects. We suggest that, given our success in applying a network approach to the results of a traditional forward genetic screen, the field of developmental genetics should find it fruitful to apply network analyses to the interpretation of large scale transcriptomic and proteomic data.”

Reviewer #2:[…](2) Numbered summary of any substantive concernsThis article is generally well-structured, the data analysis is performed in a proper fashion, and the results are nicely presented, giving support to the major conclusions. I have no major concerns with regard to this manuscript, only some minor edits.(2.1) I am wondering if authors can give more detail regarding the paragraph starting "To choose candidates.…" I might be wrong, but have the feeling that this approach might produce bias in the results obtained by the authors.

We acknowledge the concerns of the reviewer in this section, where we state our reasons for the setting of thresholds of phenotypic significance in the screens. We considered running our egg laying screen with multiple vials containing three flies each, but as we endeavoured to test many genes for multiple phenotypes with as high throughput but accurate a method as possible, we settled on using only a single vial of three flies to measure egg laying. Aiming to mitigate this bias, we did, however, set our thresholds for positive candidates in the egg laying at a higher value of |5| when compared to an ovariole number threshold of |2| to account for the difference in statistical power. We also address this concern in the Materials and methods section.

(2.2) In Figure 4—figure supplement 2, the authors show a comparison of Z_gene_ scores of positive candidate genes sorted by centrality metrics. However, I wonder why the authors did not calculate p-values (significance level) between 1st and 5th quantile comparisons of Z_gene_ scores shown in this figure? Significance levels in these comparisons might give strong support to the weak prediction of a gene that would affect a specific phenotype.

We have taken this suggestion and calculated the significance for the differences of Z_gene_ scores between the 1st and 5th quintile. We have modified Figure 4—figure supplement 2 by adding asterisks to indicate significant differences reflected by these p-values, which are shown under each comparison. Significance thresholds were set at 0.05 and are indicated in the revised figure legend, which now contains new text as follows: “significant differences (p-value<0.05 Mann Whitney U test) are indicated by asterisks. p-values are displayed below every bar plot.” We also revised the text in subsection “Centrality of genes in the ovarian protein-protein interaction networks can predict the likelihood of loss of function phenotypic effects” that describes the relationship between the centrality of genes and the strength of phenotypic effect to include the relevant pvalue < 0.05 in Figure 4—figure supplement 2”

(2.3) In Figure 6, for the sake of clarity, it is necessary to show the sloppy paired 1 gene in the meta-network analysis (Figure 6A). As is now shown in this figure, readers cannot see easily the importance of this gene within sub-network VII, which I think is one point that the authors want to highlight.

This comment by the reviewer made us realize that we needed to clarify that our intention with the reference to slp1 in the legend of Figure 6A was simply to explain that it is a gene that is present both as a seed gene in the Egg Laying sub-network as well as a novel predicted gene (connector) in the hpo[RNAi] Egg Laying sub-network. For improved clarity, we have removed all mention of slp1 from the revised legend of Figure 6A and have instead explained its presence in two sub-networks within the revised “Meta network” section of the Materials and methods.

(2.4) In step 3 in the protein-protein interaction network (PPI) building section (“Method Details” subsection), authors indicate that a custom python script was created to download and reads each of the PPI tables from DroID database, but I could not find this python script in the Github repository in the paper. Please, can authors upload this script in the repository?

We thank the reviewer for pointing this out, and have both confirmed that the script is in the repository (called 01.1_PIN_builder.ipynb) and also added a reference to this script in subsection “Protein-Protein Interaction Network (PIN) building”.

Reviewer #3:(1) The authors present a novel study integrating the results of performing screens for regulators of reproduction in *Drosophila* with protein-protein interaction networks. Highly interesting, interdisciplinary work lying at the intersection of developmental and network biology. They identify genes regulating ovariole number and egg laying and analyze those genes in the context of PPI networks, predicting and experimentally testing additional novel reproductive regulatory genes with some success. However there are serious issues with the analysis supporting key claims in the manuscript related to network topology.

We thank the reviewer for their very careful and thorough analysis of our work. We are also grateful for their appreciation of the key aspects and novelty of our work. We understand their reservations on some aspects of the manuscript and have attempted to address them with more analyses or through clearer explanations.

(2) The claim of non-random modules is not supported by the analysis.

We have both modified our analyses and performed new ones to more carefully assess this claim, and to more conservatively interpret the results of these assessments . We have a more detailed explanation below in our response to comment 2.1.

(2.1) The claim in the section title "Genes regulating egg laying and ovariole number regulation form non-random interaction modules" (L260-L261) is supported by Figure 5—figure supplement 4. In that figure both the seed genes and results from the SCA algorithm are compared to the subnetworks formed by randomly selected genes from the initial screen. This is a sensible comparison for the seed genes but not for the output of the SCA algorithm, since that algorithm builds a LCC (see Figure 5A) and so will obviously have a much larger LCC and be much more connected than randomly selected genes from the network.

We understand the concerns of the reviewer, and in response we have modified our analysis to better address this interpretation. Firstly, we compared the network metrics of the seed genes both to a similarly sized random selection of genes from the signalling candidates, and to a degree-controlled random network (see reference to Guney et al., 2016 in our response to comment 2.2 by this reviewer). Our results, shown in new Figure 4—figure supplement 3, indicate that the seed gene lists display higher network metrics than both null distributions in most cases. We therefore conclude that, independent of application of the SCA, the groups of seed genes display many features of modularity by this assessment, and this analysis thus suggests that they may function as putative modules. We report this in subsection “Genes regulating egg laying and ovariole number regulation form non-random gene interaction networks” of the revised manuscript.

Secondly, to directly address this important point by the reviewer, we assessed these network metrics in the putative modules generated by application of the SCA. We found that there was an overall reduction of network metrics when compared to null distributions of similarly sized seed lists selected from either the signalling genes (light gray curve in new Figure 5—figure supplement 5) or a degree-controlled seed list (dark grey curve in Figure 5—figure supplement 5). We therefore conclude that though the SCA may be able to correctly predict genes that may function in the network (as we present in revised Figure 7 and its two supplements), it does not increase the “modularity” of the sub-networks. We therefore conservatively refer to these connected groups of genes as “sub-networks” throughout the remainder of the manuscript. These analyses and interpretations are explained and presented in paragraph six of subsection “Genes regulating egg laying and ovariole number regulation form non-264 random gene interaction networks”.

(2.2) The results of Figure 4 show that the positive pairs in the screen on average, tend to have higher degree in the network, which could be responsible for the higher connectivity of the seed genes, rather than them forming modules. To account for this, the seed genes could be compared to a null distribution of degree-controlled randomized networks.

We understand the concern of the reviewer, and have implemented this suggestion as described in response to this reviewer’s comment 2.1 above, using the method of Guney and colleagues (Guney et al., 2016).

The revised Materials and methods section now contains the following text explaining the methodology for obtaining the degree-controlled randomized network, and mentioning the sections in the python code where this is applied, as follows. “Building degree-controlled randomized networks. We assessed the modularity of the networks by comparing the network metrics of each sub-network to a degree-controlled randomly sampled network. To generate this degree controlled random network, we applied a previously developed method (Guney et al., 2016). In short, nodes in the PPI are binned by degree with the minimum size of each bin being set at 100 nodes. Bins are constructed iteratively from the lowest degree to the highest degree in the network. To sample a set of nodes, the sub-network degree distribution is computed, using the bin cut-off, from the PPI. Then, nodes are randomly selected from each bin to match this degree distribution (see 05.2_Degree_Controlled_Testing.ipynb for code implementation).”

(2.3) The text reads "decreased average shorted path" (L331) but it is increased in 7 of 8 cases in the figure.

Our new analyses with more appropriate control groups for the SCA-generated putative modules do indeed often display relatively increased paths. This new result is described in paragraph six of subsection “Genes regulating egg laying and ovariole number regulation form non-random gene interaction networks”, and shown in new Figure 5—figure supplement 5.

(3) The claim of separate modules is not supported by the analysis.

We believe that this comment refers to our interpretation of the data shown in original Figure 6, which is discussed in subsection “Low edge densities between sub-networks suggest genetically separable mechanisms of ovariole number and egg laying”. This comment from the reviewer helped us realize that, in addition to performing the additional control analyses suggested in this reviewer’s comment 2.1, which encouraged us to be more conservative with our description of these gene groups and refer to them simply as “sub-networks” in the revised manuscript, we should improve the clarity of our interpretation of the groups of genes we present in Figure 6. In the “meta network” shown in this figure, we first separated genes into the seven different groups based on the phenotype(s) affected in their loss of function condition as assessed in our initial screens (Figure 1). Of these seven groups, three groups included genes that affected only one of the three measured phenotypes (groups I, II and III in Figure 6A), three groups included genes that affect two of the three phenotypes (groups V, VI and VII in Figure 6A), and the seventh group included genes that affected all three phenotypes (group IV in Figure 6A). The differences in protein-protein interactions between these seven different phenotypically binned groups suggested the possibility that each phenotype could be largely regulated by genes that do not interact very much with genes regulating another phenotype. We have added new text to this section in the revised manuscript with the aim of improving the explanation of our thought process for suggesting this interpretation. We further address related points in our responses to this reviewer’s comments 3.1 and 3.2 below.

(3.1) Similarly to point 1, it is not surprising that the connectivity is higher within the output of the SCA algorithm, which returns a subnetwork of connected nodes, to the connectivity between two different SCA produced sub-networks, that will happen with random lists of seed genes, it does not prove that functionally separate sub-networks have been identified.

We take this point by the reviewer, and in response to this and other reviewer comments, are more conservative in our interpretation of the functions of these gene groups in the revised manuscript. The seed lists for the SCA included any gene that affected a phenotype, unlike the seven groups of the meta network, which included genes that affected only one, two, or all three phenotypes. We have more detailed explanations in our responses to this reviewer’s comments 3.2, 3.4 and 3.6.

(3.2) There are sizable shared subnetworks (V, VI, VII) which seems to contradict the idea of separate modules.

We believe that we have addressed this comment by the modifications described in our response to comment 3 above. We hope that our revised section on low edge densities clarifies that we intended to communicate an interpretation of separable putative genetic interaction groups affecting one or more phenotypes, rather than independent functional modules, combined with our more conservative interpretations of these groups as simply “sub-networks,” has clarified our logic and intent here.

(3.3) The heat-map Figure 6B, shows very low edge density within the groups (diagonal squares), in contradiction to what is written in the text. This appears to be due to a miscalculation in the file 08_MetaModule_Analysis.ipynb in the Github repository: within the group the number of edges should be divided by (x^2^ – x) / 2 where x is the number of nodes in the group (assuming self-interactions are not considered), the edge density calculations between the different groups are correct.

We thank the reviewer for the thoroughness of their review, which has allowed us to correct this inadvertent mistake. We have now corrected the error in the formula in the code (file: 08_MetaModule_Analysis.ipynb) and updated the figure with the new densities. We are here attaching the new version of the figure for simplicity. This new, corrected result does indicate higher edge densities within groups, than between groups (new Figure 6B).

(3.4) An alternative approach to this analysis could be to look at the connectivity of only the seed genes and not the connector genes in the different groups and comparing to a random assignment of group.

We thank the reviewer for this suggestion. We have implemented it by adding the results of a new analysis, shown in new Figure 6—figure supplement 1 and described in the revised Results section in subsection “Low edge densities between sub-networks suggest genetically separable mechanisms of ovariole number and egg laying”. In this analysis, we ask if the seven groups of genes that we bin by phenotype in Figure 6A – considering either the seed genes alone, or the seed genes plus the connector genes predicted by the SCA – show connectivity different from that expected from a random group of genes of the same size. We found that the connectivity of the seed genes alone was not greater than expected by chance, but that including connector genes in the group changed their connectivity to be either higher or lower than that of a control group of randomized genes. This analysis helped clarify the internal structure of the meta network.

(3.5) Identifying modules of genes which affect one phenotype and not another is presumably made more difficult by the decision to filter candidate genes for two of the phenotypic screens based on the results of the first screen.

We agree with the reviewer that this screening strategy is likely to lead to us missing some genes that affect the second phenotype, if they did not pass our threshold for analysis based on the first screen. For this reason, we had included this explanatory text in the original manuscript “ Once established during larval life, ovariole number in *Drosophila* remains unaltered through to and during adulthood, even if oogenesis within those ovarioles suffers congenital or age-related defects (King, 1970). Because previous work suggested that ovariole number in *Drosophila* could have at least some predictive relation to egg laying (Cohet and David, 1978; Klepsatel et al., 2013b; Sarikaya and Extavour, 2015), we reasoned that scoring the latter phenotype in a primary screen (Figure 1A) could be an effective way to uncover ovariole number regulators (Figure 1C). While our results showed that this was true in many cases, it was also clear that these two traits can vary independently (Figure 2), highlighting the fact that ovariole number is not the only determinant of egg laying. Egg-laying dynamics, even during the limited five day assay used in our study, are likely influenced not just by a single anatomical parameter such as ovariole number, but rather by many biological, biomechanical, hormonal and behavioural processes.”. We have left this text unchanged in the revised manuscript.

(3.6) The prediction results do not appear to be consistent with separate modules (see point 3 below).

Thanks to this comment and others by this reviewer, we realised through our new analyses that the SCA results indeed do not support a prediction of functionally separate modules. We have therefore modified the text of the manuscript throughout the revised manuscript to reflect our new, more conservative interpretation of our results, as described in our response to this reviewer’s comment 2.1.

(4) The results of experimentally testing the predictions are presented in an unclear way, making it difficult for the reader to assess their accuracy.

To address this concern of the reviewer we have made significant modifications to the presentation of experimental validation results. We have modified Figure 7C as per the advice of the reviewer, and merged it with Figure 7D for better comparison (discussed in further detail in our response to this reviewer’s comment 4.2). We have also modified the description of the new Figure 7 in the Results (subsection “Network analysis predicts novel genes involved in egg laying and ovariole number”; discussed also in detail in our response to this reviewer’s comment 4.4) and the Discussion on the consequence of this result (subsection “Network analysis as a tool in developmental biology”; also discussed in detail in our response to this reviewer’s comment 4.1).

(4.1) The results in Figure 7A/B/C appear to show that the predictions from the modules are just as good for the phenotypes not associated with the module as for those associated with the module, e.g. for the EL phenotype there is one correct prediction from both the Core and EL modules and 9 correct predictions from the hpo[RNAi] ON/EL modules, with a similar number of predictions. So it would seem that these results suggest that, although there is positive predictive value overall relative to the initial signaling genes list, that the separate modules predicting their associated phenotype separately from the others is not supported by the data. This is not discussed in the text.

We have incorporated this important observation by the reviewer into a revised section of the Discussion, and have differentiated between prediction rates within the module, and prediction rates for genes involved in any of the phenotypic classes.

(4.2) From what I could understand, the fractions for the initial signaling gene lists for the EL and hpo[RNAi] ON phenotypes are not exactly consistent with the compared fractions of the predictions, due to the filtering applied on the hpo[RNAi] of |Z| > 1, for the initial screens. Perhaps the fractions for the 2 affected bars in Figure 7C could also be calculated with this threshold applied to remove genes?

We understand the comment of the reviewer as follows in the original Figure 7C, we compare the number of connectors from any of the three modules that are above the |Z| gene threshold of the screen. The |Z| gene threshold for the connectors in the Egg Laying and hpo[RNAi]EL screens are |5| while the |Z| gene threshold for the connectors in hpo[RNAi]ON Screen is |2|. Given that we are comparing the results of the connectors with the original signalling candidate screen results, the reviewer suggests that we utilise the same thresholds that we used in the original screen. We agree that this is logical, and accordingly, modified the thresholds for the connectors in what has become revised Figure 7C. The genes from the connector list were first subject to a threshold of |1| and the genes that were above this threshold were then counted depending on whether they were above the |Z| gene threshold |5| in Egg Laying and hpo[RNAi]Egg Laying screens and |2| in the hpo[RNAi]Ovariole Number screen. Application of this new method does not change the number of genes in the hpo[RNAi]Egg Laying screen, but does reduce the number of genes in the Egg Laying and hpo[RNAi]Ovariole Number screen. The results of these calculations have been added to the manuscript as new Figure 7—figure supplement 2.

(4.3) There are no p-values calculated, comparing the predictions to the original screens and no error bars shown in Figure 7.

This comment by the reviewer made us realize that reorganizing the figure could help with its interpretation and avoid the confusion inadvertently caused by the original version. Figures 7A and 7B, which remain unchanged from the original version in this revised manuscript, show the percentage of connector genes within each phenotypic bin, and therefore do not have error bars or p-values. In modified Figure 7C, the prediction rate comparisons between the connector candidates and the signalling candidates now indicate calculated p-values with asterisks. We have modified and merged former Figures 7C and 7D into a single new Figure 7C, shown above in our response to this reviewer’s comment 4.2.

(4.4) The title of Figure 7C could be changed to make it clearer that this contains predictions from all modules.

We have taken this suggestion by modifying the title of Figure 7C to read “Positive prediction rates of all unique connectors compared to signalling candidates”. We have also modified the legend of Figure 7C to improve the clarity of this point. The modified legend now reads as follows: “Proportion of all unique connector genes (dark grey bars) predicted by all four sub-networks compared to the proportion of signaling candidate genes (light grey bars) with |Z_gene_| above the respective threshold in any of the three phenotypic screens (Figure 1A, 1B, 1C). Positive connector and signaling candidates that were above the |Z_gene_| threshold in all three phenotypic screens (Figure 1A, 1B, 1C) are indicated in an “All phenotypes” column. Statistical significance was computed using the binomial test, comparing the probability of a positive candidate amongst the connectors to the probability of a positive candidate amongst the signalling candidates (p-value is found below each bar).”

(4.5) There is no “All screens” column in either Figure 7C or Figure 7D for comparison.

To address this comment, we have added an “All screens” column to both revised Figure 7C and 7D, which are discussed in detail in our response to this reviewer’s comment 4.2.